# Tight Regret Bounds for Model-Based Reinforcement Learning with Greedy Policies

**Yonathan Efroni**[*]
Technion, Israel

**Nadav Merlis** [*]
Technion, Israel

**Mohammad Ghavamzadeh**
Facebook AI Research

**Shie Mannor**
Technion, Israel

## Abstract

State-of-the-art efficient model-based Reinforcement Learning (RL) algorithms typically act by iteratively solving empirical models, i.e., by performing *full-planning* on Markov Decision Processes (MDPs) built by the gathered experience. In this paper, we focus on model-based RL in the finite-state finite-horizon undiscounted MDP setting and establish that exploring with *greedy policies* – act by *1-step planning* – can achieve tight minimax performance in terms of regret, $\tilde{\mathcal{O}}(\sqrt{HSAT})$. Thus, full-planning in model-based RL can be avoided altogether without any performance degradation, and, by doing so, the computational complexity decreases by a factor of $S$. The results are based on a novel analysis of real-time dynamic programming, then extended to model-based RL. Specifically, we generalize existing algorithms that perform full-planning to act by 1-step planning. For these generalizations, we prove regret bounds with the same rate as their full-planning counterparts.

## 1 Introduction

Reinforcement learning (RL) [Sutton and Barto, 2018] is a field of machine learning that tackles the problem of learning how to act in an *unknown* dynamic environment. An agent interacts with the environment, and receives feedback on its actions in the form of a state-dependent reward signal. Using this experience, the agent's goal is then to find a policy that maximizes the long-term reward.

There are two main approaches for learning such a policy: model-based and model-free. The model-based approach estimates the system's model and uses it to assess the long-term effects of actions via *full-planning* (e.g., Jaksch et al. 2010). Model-based RL algorithms usually enjoy good performance guarantees in terms of the regret – the difference between the sum of rewards gained by playing an optimal policy and the sum of rewards that the agent accumulates [Jaksch et al., 2010, Bartlett and Tewari, 2009]. Nevertheless, model-based algorithms suffer from high space and computation complexity. The former is caused by the need for storing a model. The latter is due to the frequent full-planning, which requires a full solution of the estimated model. Alternatively, model-free RL algorithms directly estimate quantities that take into account the long-term effect of an action, thus, avoiding model estimation and planning operations altogether [Jin et al., 2018]. These algorithms usually enjoy better computational and space complexity, but seem to have worse performance guarantees.

In many applications, the high computational complexity of model-based RL makes them infeasible. Thus, practical model-based approaches alleviate this computational burden by using *short-term planning* e.g., Dyna [Sutton, 1991], instead of full-planning. To the best of our knowledge, there are no regret guarantees for such algorithms, even in the tabular setting. This raises the following question: *Can a model-based approach coupled with short-term planning enjoy the favorable performance of model-based RL?*

---

[*]equal contribution

| Algorithm | Regret | Time Complexity | Space Complexity |
|---|---|---|---|
| UCRL2[2] [Jaksch et al., 2010] | $\tilde{O}(\sqrt{H^2 S^2 AT})$ | $\tilde{\mathcal{O}}(\mathcal{N}SAH)$ | $\tilde{\mathcal{O}}(HS + \mathcal{N}SA)$ |
| UCBVI [Azar et al., 2017] | $\tilde{O}(\sqrt{HSAT} + \sqrt{H^2 T})$ | $\tilde{\mathcal{O}}(\mathcal{N}SAH)$ | $\tilde{\mathcal{O}}(HS + \mathcal{N}SA)$ |
| EULER [Zanette and Brunskill, 2019] | $\tilde{\mathcal{O}}(\sqrt{HSAT})$ | $\tilde{\mathcal{O}}(\mathcal{N}SAH)$ | $\tilde{\mathcal{O}}(HS + \mathcal{N}SA)$ |
| UCRL2-GP | $\tilde{O}(\sqrt{H^2 S^2 AT})$ | $\tilde{\mathcal{O}}(\mathcal{N}AH)$ | $\tilde{\mathcal{O}}(HS + \mathcal{N}SA)$ |
| EULER-GP | $\tilde{O}(\sqrt{HSAT})$ | $\tilde{\mathcal{O}}(\mathcal{N}AH)$ | $\tilde{\mathcal{O}}(HS + \mathcal{N}SA)$ |
| Q-v2 [Jin et al., 2018] | $\tilde{O}(\sqrt{H^3 SAT}$ | $\tilde{\mathcal{O}}(AH)$ | $\tilde{\mathcal{O}}(HSA)$ |
| Lower bounds | $\Omega\left(\sqrt{SAHT}\right)$ | – | – |

Table 1: Comparison of our bounds with several state-of-the-art bounds for RL in tabular finite-horizon MDPs. The time complexity of the algorithms is per episode; $S$ and $A$ are the sizes of the state and action sets, respectively; $H$ is the horizon of the MDP; $T$ is the total number of samples that the algorithm gathers; $\mathcal{N} \leq S$ is the maximum number of non-zero transition probabilities across the entire state-action pairs. The algorithms proposed in this paper are highlighted in gray.

In this work, we show that model-based algorithms that use 1-step planning can achieve the same performance as algorithms that perform full-planning, thus, answering affirmatively to the above question. To this end, we study Real-Time Dynamic-Programming (RTDP) [Barto et al., 1995] that finds the optimal policy of a *known* model by acting greedily based on 1-step planning, and establish new and sharper finite sample guarantees. We demonstrate how the new analysis of RTDP can be incorporated into two model-based RL algorithms, and prove that the regret of the resulting algorithms remains unchanged, while their computational complexity drastically decreases. As Table 1 shows, this reduces the computational complexity of model-based RL methods by a factor of $S$.

The contributions of our paper are as follows: we first prove regret bounds for RTDP when the model is known. To do so, we establish concentration results on Decreasing Bounded Processes, which are of independent interest. We then show that the regret bound translates into a Uniform Probably Approximately Correct (PAC) [Dann et al., 2017] bound for RTDP that greatly improves existing PAC results [Strehl et al., 2006]. Next, we move to the learning problem, where the model is unknown. Based on the analysis developed for RTDP we adapt UCRL2 [Jaksch et al., 2010] and EULER [Zanette and Brunskill, 2019], both act by full-planning, to UCRL2 with Greedy Policies (UCRL2-GP) and EULER with Greedy Policies (EULER-GP); model-based algorithms that act by 1-step planning. The adapted versions are shown to preserve the performance guarantees, while improve in terms of computational complexity.

## 2 Notations and Definitions

We consider finite-horizon MDPs with time-independent dynamics [Bertsekas and Tsitsiklis, 1996]. A finite-horizon MDP is defined by the tuple $\mathcal{M} = (\mathcal{S}, \mathcal{A}, R, p, H)$, where $\mathcal{S}$ and $\mathcal{A}$ are the state and action spaces with cardinalities $S$ and $A$, respectively. The immediate reward for taking an action $a$ at state $s$ is a random variable $R(s, a) \in [0, 1]$ with expectation $\mathbb{E}R(s, a) = r(s, a)$. The transition probability is $p(s' \mid s, a)$, the probability of transitioning to state $s'$ upon taking action $a$ at state $s$. Furthermore, $\mathcal{N} := \max_{s,a} |\{s' : p(s' \mid s, a) > 0\}|$ is the maximum number of non-zero transition probabilities across the entire state-action pairs. If this number is unknown to the designer of the algorithm in advanced, then we set $\mathcal{N} = S$. The initial state in each episode is arbitrarily chosen and $H \in \mathbb{N}$ is the *horizon*, i.e., the number of time-steps in each episode. We define $[N] := \{1, \ldots, N\}$, for all $N \in \mathbb{N}$, and throughout the paper use $t \in [H]$ and $k \in [K]$ to denote time-step inside an episode and the index of an episode, respectively.

A deterministic policy $\pi : \mathcal{S} \times [H] \to \mathcal{A}$ is a mapping from states and time-step indices to actions. We denote by $a_t := \pi(s_t, t)$, the action taken at time $t$ at state $s_t$ according to a policy $\pi$. The quality

of a policy $\pi$ from state $s$ at time $t$ is measured by its value function, which is defined as

$$V_t^\pi(s) := \mathbb{E}\left[\sum_{t'=t}^{H} r(s_{t'}, \pi(s_{t'}, t')) \mid s_t = s\right],$$

where the expectation is over the environment's randomness. An optimal policy maximizes this value for all states $s$ and time-steps $t$, and the corresponding optimal value is denoted by $V_t^*(s) := \max_\pi V_t^\pi(s)$, for all $t \in [H]$. The optimal value satisfies the optimal Bellman equation, i.e.,

$$V_t^*(s) = T^* V_{t+1}^*(s) := \max_a \left\{ r(s, a) + p(\cdot \mid s, a)^T V_{t+1}^* \right\}. \tag{1}$$

We consider an agent that repeatedly interacts with an MDP in a sequence of episodes $[K]$. The performance of the agent is measured by its *regret*, defined as $\mathrm{Regret}(K) := \sum_{k=1}^{K} \left( V_1^*(s_1^k) - V_1^{\pi_k}(s_1^k) \right)$. Throughout this work, the policy $\pi_k$ is computed by a 1-step planning operation with respect to the value function estimated by the algorithm at the end of episode $k - 1$, denoted by $\bar{V}^{k-1}$. We also call such policy a *greedy policy*. Moreover, $s_t^k$ and $a_t^k$ stand, respectively, for the state and the action taken at the $t^{th}$ time-step of the $k^{th}$ episode.

Next, we define the filtration $\mathcal{F}_k$ that includes all events (states, actions, and rewards) until the end of the $k^{th}$ episode, as well as the initial state of the episode $k + 1$. We denote by $T = KH$, the total number of time-steps (samples). Moreover, we denote by $n_k(s, a)$, the number of times that the agent has visited state-action pair $(s, a)$, and by $\hat{X}_k$, the empirical average of a random variable $X$. Both quantities are based on experience gathered until the end of the $k^{th}$ episode and are $\mathcal{F}_k$ measurable. We also define the probability to visit the state-action pair $(s, a)$ at the $k^{th}$ episode at time-step $t$ by $w_{tk}(s, a) = \mathrm{Pr}\left( s_t^k = s, a_t^k = a \mid s_0^k, \pi_k \right)$. We note that $\pi_k$ is $\mathcal{F}_{k-1}$ measurable, and thus, $w_{tk}(s, a) = \mathrm{Pr}\left( s_t^k = s, a_t^k = a \mid \mathcal{F}_{k-1} \right)$. Also denote $w_k(s, a) = \sum_{t=1}^{H} w_{tk}(s, a)$.

We use $\tilde{\mathcal{O}}(X)$ to refer to a quantity that depends on $X$ up to poly-log expression of a quantity at most polynomial in $S$, $A$, $T$, $K$, $H$, and $\frac{1}{\delta}$. Similarly, $\lesssim$ represents $\leq$ up to numerical constants or poly-log factors. We define $\|X\|_{2,p} := \sqrt{\mathbb{E}_p X^2}$, where $p$ is a probability distribution over the domain of $X$, and use $X \vee Y := \max\{X, Y\}$. Lastly, $\mathcal{P}(\mathcal{S})$ is the set of probability distributions over the state space $\mathcal{S}$.

## 3 Real-Time Dynamic Programming

---
**Algorithm 1** Real-Time Dynamic Programming

---
    Initialize: $\forall s \in \mathcal{S}$, $\forall t \in [H]$, $\bar{V}_t^0(s) = H - (t - 1)$.
    **for** $k = 1, 2, \ldots$ **do**
        Initialize $s_1^k$
        **for** $t = 1, \ldots, H$ **do**
            $a_t^k \in \arg\max_a r(s_t^k, a) + p(\cdot \mid s_t^k, a)^T \bar{V}_{t+1}^{k-1}$
            $\bar{V}_t^k(s_t^k) = r(s_t^k, a_t^k) + p(\cdot \mid s_t^k, a_t^k)^T \bar{V}_{t+1}^{k-1}$
            Act with $a_t^k$ and observe $s_{t+1}^k$.
        **end for**
    **end for**

---

RTDP [Barto et al., 1995] is a well-known algorithm that solves an MDP when a model of the environment is given. Unlike, e.g., Value Iteration (VI) [Bertsekas and Tsitsiklis, 1996] that solves an MDP by offline calculations, RTDP solves an MDP in a real-time manner. As mentioned in Barto et al. [1995], RTDP can be interpreted as an asynchronous VI adjusted to a real-time algorithm.

Algorithm 1 contains the pseudocode of RTDP for finite-horizon MDPs. The value function is initialized with an optimistic value, i.e., an upper bound of the optimal value. At each time-step $t$ and episode $k$, the agent acts from the current state $s_t^k$ greedily with respect to the current value at the next time step, $\bar{V}_{t+1}^{k-1}$. It then updates the value of $s_t^k$ according to the optimal Bellman operator. We denote by $\bar{V}$, the value function, and as we show in the following, it always upper bounds $V^*$. Note that since the action at a fixed state is chosen according to $\bar{V}^{k-1}$, then $\pi_k$ is $\mathcal{F}_{k-1}$ measurable.

Since RTDP is an online algorithm, i.e., it updates its value estimates through interactions with the environment, it is natural to measure its performance in terms of the regret. The rest of this section is devoted to supplying expected and high-probability bounds on the regret of RTDP, which will also lead to PAC bounds for this algorithm. In Section 4, based on the observations from this section, we will establish minimax regret bounds for 1-step greedy model-based RL.

We start by stating two basic properties of RTDP in the following lemma: the value is always optimistic and decreases in $k$ (see proof in Appendix B). Although the first property is known [Barto et al., 1995], to the best of our knowledge, the second one has not been proven in previous work.

**Lemma 1.** *For all $s$, $t$, and $k$, it holds that (i) $V_t^*(s) \leq \bar{V}_t^k(s)$ and (ii) $\bar{V}_t^k(s) \leq \bar{V}_t^{k-1}(s)$.*

The following lemma, that we believe is new, relates the difference between the optimistic value $\bar{V}_1^{k-1}(s_1^k)$ and the real value $V_1^{\pi_k}(s_1^k)$ to the *expected cumulative update* of the value function at the end of the $k^{th}$ episode (see proof in Appendix B).

**Lemma 2** (Value Update for Exact Model). *The expected cumulative value update of RTDP at the $k^{th}$ episode satisfies*

$$\bar{V}_1^{k-1}(s_1^k) - V_1^{\pi_k}(s_1^k) = \sum_{t=1}^H \mathbb{E}[\bar{V}_t^{k-1}(s_t^k) - \bar{V}_t^k(s_t^k) \mid \mathcal{F}_{k-1}].$$

The result relates the difference of the optimistic value $\bar{V}^{k-1}$ and the value of the greedy policy $V^{\pi_k}$ to the expected update along the trajectory, created by following $\pi_k$. Thus, for example, if the optimistic value is overestimated, then the value update throughout this episode is expected to be large.

## 3.1 Regret and PAC Analysis

Using Lemma 1, we observe that the sequence of values is decreasing and bounded from below. Thus, intuitively, the decrements of the values cannot be indefinitely large. Importantly, Lemma 2 states that when the expected decrements of the values are small, then $V_1^{\pi_k}(s_1^k)$ is close to $\bar{V}^{k-1}(s_1^k)$, and thus, to $V^*$, since $\bar{V}^{k-1}(s_1^k) \geq \bar{V}^*(s_1^k) \geq V_1^{\pi_k}(s_1^k)$.

Building on this reasoning, we are led to establish a general result on a decreasing process. This result will allow us to formally justify the aforementioned reasoning and derive regret bounds for RTDP. The proof utilizes self-normalized concentration bounds [de la Peña et al., 2007], applied on martingales, and can be found in Appendix A.

**Definition 1** (Decreasing Bounded Process). *We call a random process $\{X_k, \mathcal{F}_k\}_{k \geq 0}$, where $\{\mathcal{F}_k\}_{k \geq 0}$ is a filtration and $\{X_k\}_{k \geq 0}$ is adapted to this filtration, a Decreasing Bounded Process, if it satisfies the following properties:*

1. *$\{X_k\}_{k \geq 0}$ decreases, i.e., $X_{k+1} \leq X_k$ a.s. .*

2. *$X_0 = C \geq 0$, and for all $k$, $X_k \geq 0$ a.s. .*

**Theorem 3** (Regret Bound of a Decreasing Bounded Process). *Let $\{X_k, \mathcal{F}_k\}_{k \geq 0}$ be a Decreasing Bounded Process and $R_K = \sum_{k=1}^K X_{k-1} - \mathbb{E}[X_k \mid \mathcal{F}_{k-1}]$ be its $K$-round regret. Then,*

$$\Pr\left\{\exists K > 0 : R_K \geq C\left(1 + 2\sqrt{\ln(2/\delta)}\right)^2\right\} \leq \delta.$$

*Specifically, it holds that $\Pr\{\exists K > 0 : R_K \geq 9C\ln(3/\delta)\} \leq \delta$.*

We are now ready to prove the central result of this section, the expected and high-probability regret bounds on RTDP (see full proof in Appendix B).

**Theorem 4** (Regret Bounds for RTDP). *The following regret bounds hold for RTDP:*

1. *$\mathbb{E}[\text{Regret}(K)] \leq SH^2$.*

2. *For any $\delta > 0$, with probability $1 - \delta$, for all $K > 0$, $\text{Regret}(K) \leq 9SH^2\ln(3SH/\delta)$.*

*Proof Sketch.* We give a sketch of the proof of the second claim. Applying Lemmas 1 and then 2,

$$\text{Regret}(K) := \sum_{k=1}^{K} V_1^*(s_1^k) - V_1^{\pi_k}(s_1^k) \leq \sum_{k=1}^{K} \bar{V}_1^{k-1}(s_1^k) - V^{\pi_k}(s_1^k)$$

$$\leq \sum_{k=1}^{K} \sum_{t=1}^{H} \mathbb{E}[\bar{V}_t^{k-1}(s_t^k) - \bar{V}_t^k(s_t^k) \mid \mathcal{F}_{k-1}]. \tag{2}$$

We then establish (see Lemma 34) that RHS of (2) is, in fact, a sum of $SH$ Decreasing Bounded Processes, i.e.,

$$(2) = \sum_{t=1}^{H} \sum_{s \in \mathcal{S}} \sum_{k=1}^{K} \bar{V}_t^{k-1}(s) - \mathbb{E}[\bar{V}_t^k(s) \mid \mathcal{F}_{k-1}]. \tag{3}$$

Since for any fixed $s, t$, $\{\bar{V}_t^k(s)\}_{k \geq 0}$ is a decreasing process by Lemma 1, we can use Theorem 3, for a fixed $s, t$, and conclude the proof by applying the union bound on all $SH$ terms in (3). □

Theorem 4 exhibits a regret bound that does not depend on $T = KH$. While it is expected that RTDP, that has access to the exact model, would achieve better performance than an RL algorithm with no such access, a regret bound independent of $T$ is a noteworthy result. Indeed, it leads to the following Uniform PAC (see Dann et al. 2017 for the definition) and $(0, \delta)$ PAC guarantees for RTDP (see proofs in Appendix B). To the best of our knowledge, both are the first PAC guarantees for RTDP.[3]

**Corollary 5** (RTDP is Uniform PAC). *Let $\delta > 0$ and $N_\epsilon$ be the number of episodes in which RTDP outputs a policy with $V_1^*(s_1^k) - V_1^{\pi_k}(s_1^k) > \epsilon$. Then,*

$$\Pr\left\{\exists \epsilon > 0 : N_\epsilon \geq \frac{9SH^2 \ln(3SH/\delta)}{\epsilon}\right\} \leq \delta.$$

**Corollary 6** (RTDP is $(0, \delta)$ PAC). *Let $\delta > 0$ and $N$ be the number of episodes in which RTDP outputs a non optimal policy. Define the (unknown) gap of the MDP, $\Delta(\mathcal{M}) = \min_s \min_{\pi:V_1^\pi(s) \neq V_1^*(s)} V_1^*(s) - V_1^\pi(s) > 0$. Then,*

$$\Pr\left\{N \geq \frac{9SH^2 \ln(3SH/\delta)}{\Delta(\mathcal{M})}\right\} \leq \delta.$$

## 4 Exploration in Model-based RL: Greedy Policy Achieves Minimax Regret

We start this section by formulating a general optimistic RL scheme that acts by 1-step planning (see Algorithm 2). Then, we establish Lemma 7, which generalizes Lemma 2 to the case where a non-exact model is used for the value updates. Using this lemma, we offer a novel regret decomposition for algorithms which follow Algorithm 2. Based on the decomposition, we analyze generalizations of UCRL2 [Jaksch et al., 2010] (for finite horizon MDPs) and EULER [Zanette and Brunskill, 2019], that use greedy policies instead of solving an MDP (full planning) at the beginning of each episode. Surprisingly, we find that both generalized algorithms do not suffer from performance degradation, up to numerical constants and logarithmic factors. Thus, we conclude that there exists an RL algorithm that achieves the minimax regret bound, while acting according to greedy policies.

Consider the general RL scheme that explores by greedy policies as depicted in Algorithm 2. The value $\bar{V}$ is initialized optimistically and the algorithm interacts with the unknown environment in an episodic manner. At each time-step $t$, a greedy policy from the current state, $s_t^k$, is calculated optimistically based on the empirical model $(\hat{r}_{k-1}, \hat{p}_{k-1}, n_{k-1})$ and the current value at the next time-step $\bar{V}_{t+1}^{k-1}$. This is done in a subroutine called 'ModelBaseOptimisticQ'.[4] We further assume the optimistic $Q$-function has the form $\bar{Q}(s_t^k, a) = \tilde{r}_{k-1}(s_t^k, a) + \tilde{p}_{k-1}(\cdot \mid s_t^k, a)^T \bar{V}_{t+1}^{k-1}$ and refer to

**Algorithm 2** Model-based RL with Greedy Policies
___
1: Initialize: $\forall s \in \mathcal{S},\ \forall t \in [H],\ \bar{V}_t^0(s) = H - (t-1)$.
2: **for** $k = 1, 2, \dots$ **do**
3:     Initialize $s_1^k$
4:     **for** $t = 1, \dots, H$ **do**
5:         $\forall a,\ \bar{Q}(s_t^k, a) = \text{ModelBaseOptimisticQ}\big(\hat{r}_{k-1}, \hat{p}_{k-1}, n_{k-1}, \bar{V}_{t+1}^{k-1}\big)$
6:         $a_t^k \in \arg\max_a \bar{Q}(s_t^k, a)$
7:         $\bar{V}_t^k(s_t^k) = \min\big\{\bar{V}_t^{k-1}(s_t^k), \bar{Q}(s_t^k, a_t^k)\big\}$
8:         Act with $a_t^k$ and observe $s_{t+1}^k$.
9:     **end for**
10:    Update $\hat{r}_k, \hat{p}_k, n_k$ with all experience gathered in episode.
11: **end for**
___

$(\tilde{r}_{k-1}, \tilde{p}_{k-1})$ as the optimistic model. The agent interacts with the environment based on the greedy policy with respect to $\bar{Q}$ and uses the gathered experience to update the empirical model at the end of the episode.

By construction of the update rule (see Line 7), the value is a decreasing function of $k$, for all $(s,t) \in \mathcal{S} \times [H]$. Thus, property (ii) in Lemma 1 holds for Algorithm 2. Furthermore, the algorithms analyzed in this section will also be optimistic with high probability, i.e., property (i) in Lemma 1 also holds. Finally, since the value update uses the empirical quantities $\hat{r}_{k-1}$, $\hat{p}_{k-1}$, $n_{k-1}$ and $\bar{V}_{t+1}^{k-1}$ from the previous episode, policy $\pi_k$ is still $\mathcal{F}_{k-1}$ measurable.

The following lemma generalizes Lemma 2 to the case where, unlike in RTDP, the update rule does not use the exact model (see proof in Appendix C).

**Lemma 7** (Value Update for Optimistic Model)**.** *The expected cumulative value update of Algorithm 2 in the $k^{th}$ episode is bounded by*

$$\bar{V}_1^{k-1}(s_1^k) - V_1^{\pi_k}(s_1^k) \le \sum_{t=1}^{H} \mathbb{E}\big[\bar{V}_t^{k-1}(s_t^k) - \bar{V}_t^k(s_t^k) \mid \mathcal{F}_{k-1}\big]$$

$$+ \sum_{t=1}^{H} \mathbb{E}\big[(\tilde{r}_{k-1} - r)(s_t^k, a_t^k) + (\tilde{p}_{k-1} - p)(\cdot \mid s_t^k, a_t^k)^T \bar{V}_{t+1}^{k-1} \mid \mathcal{F}_{k-1}\big]\ .$$

In the rest of the section, we consider two instantiations of the subroutine 'ModelBaseOptimisticQ' in Algorithm 2. We use the bonus terms of UCRL2 and of EULER to acquire an optimistic $Q$-function, $\bar{Q}$. These two options then lead to UCRL2 with Greedy Policies (UCRL2-GP) and EULER with Greedy Policies (EULER-GP) algorithms.

## 4.1   UCRL2 with Greedy Policies for Finite-Horizon MDPs

**Algorithm 3** UCRL2 with Greedy Policies (UCRL2-GP)
___
1: $\tilde{r}_{k-1}(s_t^k, a) = \hat{r}_{k-1}(s_t^k, a) + \sqrt{\frac{2\ln\frac{8SAT}{\delta}}{n_{k-1}(s_t^k, a)\vee 1}}$

2: $CI(s_t^k, a) = \left\{ P' \in \mathcal{P}(\mathcal{S}) : \|P'(\cdot) - \hat{p}_{k-1}(\cdot \mid s_t^k, a)\|_1 \le \sqrt{\frac{4S\ln\frac{12SAT}{\delta}}{n_{k-1}(s_t^k, a)\vee 1}} \right\}$

3: $\tilde{p}_{k-1}(\cdot \mid s_t^k, a) = \arg\max_{P' \in CI(s_t^k, a)} P'(\cdot \mid s_t^k, a)^T \bar{V}_{t+1}^{k-1}$

4: $\bar{Q}(s_t^k, a) = \tilde{r}_{k-1}(s_t^k, a) + \tilde{p}_{k-1}(\cdot \mid s_t^k, a)^T \bar{V}_{t+1}^{k-1}$

5: **Return** $\bar{Q}(s_t^k, a)$
___

We form the optimistic local model based on the confidence set of UCRL2 [Jaksch et al., 2010]. This amounts to use Algorithm 3 as the subroutine 'ModelBaseOptimisticQ' in Algorithm 2. The maximization problem on Line 3 of Algorithm 3 is common, when using bonus based on an optimistic model [Jaksch et al., 2010], and it can be solved efficiently in $\tilde{\mathcal{O}}(\mathcal{N})$ operations (e.g., Strehl and Littman 2008, Section 3.1.5). A full version of the algorithm can be found in Appendix D.

Thus, Algorithm 3 performs $\mathcal{N}AH$ operations per episode. This saves the need to perform Extended Value Iteration [Jaksch et al., 2010], that costs $\mathcal{N}SAH$ operations per episode (an extra factor of $S$). Despite the significant improvement in terms of computational complexity, the regret of UCRL2-GP is similar to the one of UCRL2 [Jaksch et al., 2010] as the following theorem formalizes (see proof in Appendix D).

**Theorem 8** (Regret Bound of UCRL2-GP). *For any time $T \leq KH$, with probability at least $1 - \delta$, the regret of UCRL2-GP is bounded by $\tilde{\mathcal{O}}\left( HS\sqrt{AT} + H^2\sqrt{S}SA \right)$.*

*Proof Sketch.* Using the optimism of the value function (see Section D.2) and by applying Lemma 7, we bound the regret as follows:

$$
\begin{aligned}
\text{Regret}(K) = \sum_{k=1}^{K} V_1^*(s_1^k) - V_1^{\pi_k}(s_1^k) &\leq \sum_{k=1}^{K} \bar{V}_1^{k-1}(s_1^k) - V_1^{\pi_k}(s_1^k) \\
&\leq \sum_{k=1}^{K} \sum_{t=1}^{H} \mathbb{E}[\bar{V}_t^{k-1}(s_t^k) - \bar{V}_t^k(s_t^k) \mid \mathcal{F}_{k-1}] \\
&\quad + \sum_{k=1}^{K} \sum_{t=1}^{H} \mathbb{E}\left[ (\tilde{r}_{k-1} - r)(s_t^k, a_t^k) + (\tilde{p}_{k-1} - p)(\cdot \mid s_t^k, a_t^k)^T \bar{V}_{t+1}^{k-1} \mid \mathcal{F}_{k-1} \right]. \quad (4)
\end{aligned}
$$

Thus, the regret is upper bounded by two terms. As in Theorem 4, by applying Lemma 11 (Appendix A), the first term in (4) is a sum of $SH$ Decreasing Bounded Processes, and can thus be bounded by $\tilde{\mathcal{O}}(SH^2)$. The presence of the second term in (4) is common in recent regret analyses (e.g., Dann et al. 2017). Using standard techniques [Jaksch et al., 2010, Dann et al., 2017, Zanette and Brunskill, 2019], this term can be bounded (up to additive constant factors) with high probability by $\lesssim H\sqrt{S} \sum_{k=1}^{K} \sum_{t=1}^{H} \mathbb{E}\left[ \sqrt{\frac{1}{n_{k-1}(s_t^k, a_t^k)}} \mid \mathcal{F}_{k-1} \right] \leq \tilde{\mathcal{O}}(HS\sqrt{AT})$. □

### 4.2 EULER with Greedy Policies

In this section, we use bonus terms as in EULER [Zanette and Brunskill, 2019]. Similar to the previous section, this amounts to replacing the subroutine 'ModelBaseOptimisticQ' in Algorithm 2 with a subroutine based on the bonus terms from [Zanette and Brunskill, 2019]. Algorithm 5 in Appendix E contains the pseudocode of the algorithm. The bonus terms in EULER are based on the empirical Bernstein inequality and tracking both an upper bound $\bar{V}_t$ and a lower-bound $\underline{V}_t$ on $V_t^*$. Using these, EULER achieves both minimax optimal and problem dependent regret bounds.

EULER [Zanette and Brunskill, 2019] performs $\mathcal{O}(\mathcal{N}SAH)$ computations per episode (same as the VI algorithm), while EULER-GP requires only $\mathcal{O}(\mathcal{N}AH)$. Despite this advantage in computational complexity, EULER-GP exhibits similar minimax regret bounds to EULER (see proof in Appendix E), much like the equivalent performance of UCRL2 and UCRL2-GP proved in Section 4.1.

**Theorem 9** (Regret Bound of EULER-GP). *Let $\mathcal{G}$ be an upper bound on the total reward collected within an episode. Define $\mathbb{Q}^* := \max_{s,a,t}\left( \text{Var}R(s,a) + \text{Var}_{s' \sim p(\cdot|s,a)} V_{t+1}^*(s) \right)$ and $H_{\text{eff}} := \min\{\mathbb{Q}^*, \mathcal{G}^2/H\}$. With probability $1 - \delta$, for any time $T \leq KH$ jointly on all episodes $k \in [K]$, the regret of EULER-GP is bounded by $\tilde{\mathcal{O}}\left( \sqrt{H_{\text{eff}}SAT} + \sqrt{S}SAH^2(\sqrt{S} + \sqrt{H}) \right)$. Thus, it is also bounded by $\tilde{\mathcal{O}}\left( \sqrt{HSAT} + \sqrt{S}SAH^2(\sqrt{S} + \sqrt{H}) \right)$.*

Note that Theorem 9 exhibits similar problem-dependent regret-bounds as in Theorem 1 of [Zanette and Brunskill, 2019]. Thus, the same corollaries derived in [Zanette and Brunskill, 2019] for EULER can also be applied to EULER-GP.

## 5 Experiments

In this section, we present an empirical evaluation of both UCRL2 and EULER, and compare their performance to the proposed variants, which use greedy policy updates, UCRL2-GP and EULER-GP,

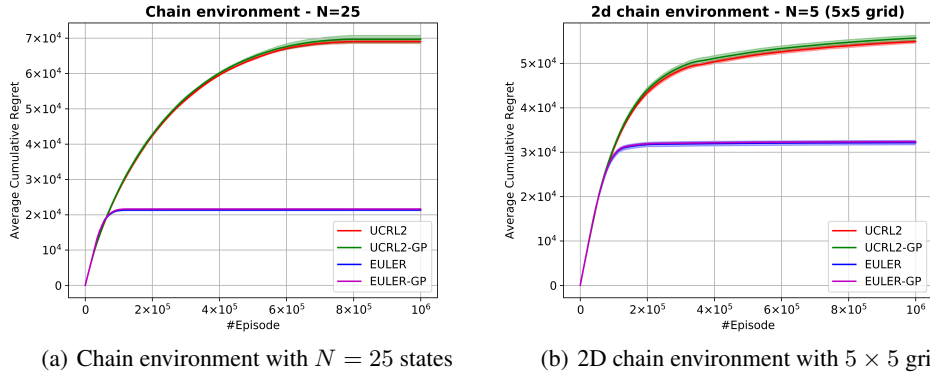

|                          |                          |
| ------------------------ | ------------------------ |
| (a) Chain environment with $N = 25$ states | (b) 2D chain environment with $5 \times 5$ grid |

Figure 1: A comparison UCRL2 and EULER with their greedy counterpart. Results are averaged over 5 random seeds and are shown alongside error bars ($\pm$3std).

respectively. We evaluated the algorithms on two environments. (i) **Chain environment** [Osband and Van Roy, 2017]: In this MDP, there are $N$ states, which are connected in a chain. The agent starts at the left side of the chain and can move either to the left or try moving to the right, which succeeds w.p. $1 - 1/N$, and results with movement to the left otherwise. The agent goal is to reach the right side of the chain and try moving to the right, which results with a reward $r \sim \mathcal{N}(1, 1)$. Moving backwards from the initials state also results with $r \sim \mathcal{N}(0, 1)$, and otherwise, the reward is $r = 0$. Furthermore, the horizon is set to $H = N$, so that the agent must always move to the right to have a chance to receive a reward. (ii) **2D chain**: A generalization of the chain environment, in which the agent starts at the upper-left corner of a $N \times N$ grid and aims to reach the lower-right corner and move towards this corner, in $H = 2N - 1$ steps. Similarly to the chain environment, there is a probability $1/H$ to move backwards (up or left), and the agent must always move toward the corner to observe a reward $r \sim \mathcal{N}(1, 1)$. Moving into the starting corner results with $r \sim \mathcal{N}(0, 1)$, and otherwise $r = 0$. This environment is more challenging for greedy updates, since there are many possible trajectories that lead to reward.

The simulation results can be found in Figure 1, and clearly indicate that using greedy planning leads to negligible degradation in the performance. Thus, the simulations verify our claim that greedy policy updates greatly improve the efficiency of the algorithm while maintaining the same performance.

## 6   Related Work

**Real-Time Dynamic Programming**: RTDP [Barto et al., 1995] has been extensively used and has many variants that exhibit superior empirical performance (e.g., [Bonet and Geffner, 2003, McMahan et al., 2005, Smith and Simmons, 2006]). For discounted MDPs, Strehl et al. [2006] proved $(\epsilon, \delta)$-PAC bounds of $\tilde{O}\left(SA/\epsilon^2(1 - \gamma)^4\right)$, for a modified version of RTDP in which the value updates occur only if the decrease in value is larger than $\epsilon(1 - \gamma)$. I.e., their algorithm explicitly use $\epsilon$ to mark states with accurate value estimate. We prove that RTDP converges in a rate of $\tilde{O}\left(SH^2/\epsilon\right)$ without knowing $\epsilon$. Indeed, Strehl et al. [2006] posed *whether the original RTDP is PAC* as an open problem. Furthermore, no regret bound for RTDP has been reported in the literature.

**Regret bounds for RL**: The most renowned algorithms with regret guarantees for undiscounted infinite-horizon MDPs are UCRL2 [Jaksch et al., 2010] and REGAL [Bartlett and Tewari, 2009], which have been extended throughout the years (e.g., by Fruit et al. 2018, Talebi and Maillard 2018). Recently, there is an increasing interest in regret bounds for MDPs with finite horizon $H$ and stationary dynamics. In this scenario, UCRL2 enjoys a regret bound of order $HS\sqrt{AT}$. Azar et al. [2017] proposed UCBVI, with improved regret bound of order $\sqrt{HSAT}$, which is also asymptotically tight [Osband and Van Roy, 2016]. Dann et al. [2018] presented ORLC that achieves tight regret bounds and (nearly) tight PAC guarantees for non-stationary MDPs. Finally, Zanette and Brunskill [2019] proposed EULER, an algorithm that enjoys tight minimax regret bounds and has additional

problem-dependent bounds that encapsulate the MDP's complexity. All of these algorithms are model-based and require full-planning. Model-free RL was analyzed by [Jin et al., 2018]. There, the authors exhibit regret bounds that are worse by a factor of $H$ relatively to the lower-bound. To the best of our knowledge, there are no model-based algorithms with regret guarantees that avoid full-planning. It is worth noting that while all the above algorithms, and the ones in this work, rely on the Optimism in the Face of Uncertainty principle [Lai and Robbins, 1985], Thompson Sampling model-based RL algorithms exist [Osband et al., 2013, Gopalan and Mannor, 2015, Agrawal and Jia, 2017, Osband and Van Roy, 2017]. There, a model is sampled from a distribution over models, on which full-planning takes place.

**Greedy policies in model-based RL:** By adjusting RTDP to the case where the model is unknown, Strehl et al. [2012] formulated model-based RL algorithms that act using a greedy policy. They proved a $\tilde{O}\left(S^2 A/\epsilon^3 (1-\gamma)^6\right)$ sample complexity bound for discounted MDPs. To the best of our knowledge, there are no regret bounds for model-based RL algorithms that act by greedy policies.

**Practical model-based RL**: Due to the high computational complexity of planning in model-based RL, most of the practical algorithms are model-free (e.g., Mnih et al. 2015). Algorithms that do use a model usually only take advantage of local information. For example, Dyna [Sutton, 1991, Peng et al., 2018] selects state-action pairs, either randomly or via prioritized sweeping [Moore and Atkeson, 1993, Van Seijen and Sutton, 2013], and updates them according to a local model. Other papers use the local model to plan for a short horizon from the current state [Tamar et al., 2016, Hafner et al., 2018]. The performance of such algorithms depends heavily on the planning horizon, that in turn dramatically increases the computational complexity.

# 7    Conclusions and Future Work

In this work, we established that tabular model-based RL algorithms can explore by 1-step planning instead of full-planning, without suffering from performance degradation. Specifically, exploring with model-based greedy policies can be minimax optimal in terms of regret. Differently put, the variance caused by exploring with greedy policies is smaller than the variance caused by learning a sufficiently good model. Indeed, the extra term which appears due to the greedy exploration is $\tilde{\mathcal{O}}(SH^2)$ (e.g., the first term in (4)); a constant term, smaller than the existing constant terms of UCRL2 and EULER.

This work raises and highlights some interesting research questions. The obvious ones are extensions to average and discounted MDPs, as well as to Thompson sampling based RL algorithms. Although these scenarios are harder or different in terms of analysis, we believe this work introduces the relevant approach to tackle this question. Another interesting question is the applicability of the results in large-scale problems, when tabular representation is infeasible and approximation must be used. There, algorithms that act using lookahead policies, instead of 1-step planning, are expected to yield better performance, as they are less sensitive to value approximation errors (e.g., Bertsekas and Tsitsiklis 1996, Jiang et al. 2018, Efroni et al. 2018b,a). Even then, full-planning, as opposed to using a short-horizon planning, might be unnecessary. Lastly, establishing whether the model-based approach is or is not provably better than the model-free approach, as the current state of the literature suggests, is yet an important and unsolved open problem.

# Acknowledgments

We thank Oren Louidor for illuminating discussions relating the Decreasing Bounded Process, and Esther Derman for the very helpful comments. This work was partially funded by the Israel Science Foundation under ISF grant number 1380/16.

## Footnotes

[2]Similarly to previous work in the finite horizon setting, we state the regret in terms of the horizon $H$. The regret in the infinite horizon setting is $DS\sqrt{AT}$, where $D$ is the diameter of the MDP.

[3]Existing PAC results on RTDP analyze variations of RTDP in which $\epsilon$ is an input parameter of the algorithm.

[4]We also allow the subroutine to use $\mathcal{O}(S)$ internal memory for auxiliary calculations, which does not change the overall space complexity.

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
