# A  Proofs on Decreasing Bounded Processes

In this section, we state and prove useful results on Decreasing Bounded Processes (see Definition 1). These results will be in use in proofs of the central theorems of this work.

**Theorem 3** (Regret Bound of a Decreasing Bounded Process). *Let $\{X_k, \mathcal{F}_k\}_{k \geq 0}$ be a Decreasing Bounded Process and $R_K = \sum_{k=1}^{K} X_{k-1} - \mathbb{E}[X_k \mid \mathcal{F}_{k-1}]$ be its $K$-round regret. Then,*

$$\Pr\left\{\exists K > 0 : R_K \geq C\left(1 + 2\sqrt{\ln(2/\delta)}\right)^2\right\} \leq \delta.$$

*Specifically, it holds that* $\Pr\{\exists K > 0 : R_K \geq 9C\ln(3/\delta)\} \leq \delta$.

*Proof.* Without loss of generality, assume $C > 0$, since otherwise the results are trivial. We start by remarking that $R_K$ is almost surely monotonically increasing, since $X_k \leq X_{k-1}$. Define the martingale difference process

$$\xi_k = X_k - \mathbb{E}[X_k \mid \mathcal{F}_{k-1}] = X_k - X_{k-1} - \mathbb{E}[X_k - X_{k-1} \mid \mathcal{F}_{k-1}]$$

and the martingale process $M_K = \sum_{k=1}^{K} \xi_k$. Since $X_k \geq 0$ almost surely, $R_K$ can be bounded by $R_K = M_K + X_0 - X_K \leq X_0 + M_K$. Also define the quadratic variations as $\langle M \rangle_K = \sum_{k=1}^{K} \mathbb{E}[\xi_k^2 \mid \mathcal{F}_{k-1}]$ and $[M]_K = \sum_{k=1}^{K} \xi_k^2$. Next, recall Theorem 2.7 of [de la Peña et al., 2007]:

**Theorem 10.** *Let $A$ and $B$ be two random variables, such that for all $\lambda \in \mathbb{R}$, we have*

$$\mathbb{E}\left[e^{\lambda A - \frac{\lambda^2 B^2}{2}}\right] \leq 1 \ . \tag{5}$$

*Then, $\forall x > 0$,*

$$\Pr\left\{\frac{|A|}{\sqrt{B^2 + \mathbb{E}[B^2]}} > x\right\} \leq \sqrt{2}e^{-x^2/4}. \tag{6}$$

Condition (5) holds for $A_K = M_K$ and $B_K^2 = \langle M \rangle_K + [M]_K$, due to Theorem 9.21 of [de la Peña et al., 2008]. $A_K$ can be easily bounded by $|A_K| \geq R_K - X_0 \geq R_K - C$. To bound $B_K^2$, we first calculate $\xi_k^2$ and $\mathbb{E}[\xi_k^2 \mid \mathcal{F}_{k-1}]$:

$$\xi_k^2 = (X_k - X_{k-1})^2 - 2(X_k - X_{k-1})\mathbb{E}[X_k - X_{k-1} \mid \mathcal{F}_{k-1}] + (\mathbb{E}[X_k - X_{k-1} \mid \mathcal{F}_{k-1}])^2,$$

$$\mathbb{E}[\xi_k^2 \mid \mathcal{F}_{k-1}] = \mathbb{E}\left[(X_k - X_{k-1})^2 \mid \mathcal{F}_{k-1}\right] - (\mathbb{E}[X_k - X_{k-1} \mid \mathcal{F}_{k-1}])^2.$$

Thus,

$$\xi_k^2 + \mathbb{E}[\xi_k^2 \mid \mathcal{F}_{k-1}]$$

$$= (X_k - X_{k-1})^2 + \mathbb{E}\left[(X_k - X_{k-1})^2 \mid \mathcal{F}_{k-1}\right] - 2(X_k - X_{k-1})\mathbb{E}[X_k - X_{k-1} \mid \mathcal{F}_{k-1}]$$

$$\overset{(*)}{\leq} (X_k - X_{k-1})^2 + \mathbb{E}\left[(X_k - X_{k-1})^2 \mid \mathcal{F}_{k-1}\right]$$

$$\overset{(**)}{\leq} (X_k - X_{k-1})^2 + C\mathbb{E}[X_{k-1} - X_k \mid \mathcal{F}_{k-1}] \ .$$

In $(*)$ we used the fact that $X_{k-1} - X_k \geq 0$ a.s., which allows us to conclude that the cross-term is non-positive. In $(**)$, we bounded $0 \leq X_{k-1} - X_k \leq C$. We can also bound $\sum_{k=1}^{K}(X_{k-1} - X_k)^2 \leq C^2$, since each of the summands is a.s. non-negative, and thus,

$$\sum_{k=1}^{K}(X_{k-1} - X_k)^2 \leq \left(\sum_{k=1}^{K} X_{k-1} - X_k\right)^2 = (X_K - X_0)^2 \leq C^2.$$

Combining all of the above bounds yields

$$B_K^2 \leq \sum_{k=1}^{K}\left((X_k - X_{k-1})^2 + C\mathbb{E}[X_{k-1} - X_k \mid \mathcal{F}_{k-1}]\right)$$

$$\leq C^2 + C\sum_{k=1}^{K}\mathbb{E}[X_{k-1} - X_k \mid \mathcal{F}_{k-1}] = C^2 + CR_K.$$

Finally, we can bound $\mathbb{E}\big[B_K^2\big]$ by

$$\mathbb{E}\big[B_K^2\big] = \sum_{k=1}^{K} \mathbb{E}\big[\xi_k^2 + \mathbb{E}[\xi_k^2 \mid \mathcal{F}_{k-1}]\big] = 2\sum_{k=1}^{K} \mathbb{E}\big[\mathbb{E}[\xi_k^2 \mid \mathcal{F}_{k-1}]\big]$$

$$= 2\sum_{k=1}^{K} \mathbb{E}\Big[\mathbb{E}\big[(X_k - X_{k-1})^2 \mid \mathcal{F}_{k-1}\big] - (\mathbb{E}[X_k - X_{k-1} \mid \mathcal{F}_{k-1}])^2\Big]$$

$$\leq 2\sum_{k=1}^{K} \mathbb{E}\Big[\mathbb{E}\big[(X_k - X_{k-1})^2 \mid \mathcal{F}_{k-1}\big]\Big] = 2\sum_{k=1}^{K}(X_k - X_{k-1})^2 \leq 2C^2.$$

Combining everything we obtain

$$\frac{|A|}{\sqrt{B^2 + \mathbb{E}[B^2]}} \geq \frac{R_K - C}{\sqrt{C^2 + CR_K + 2C^2}} = \frac{R_K - C}{\sqrt{3C^2 + CR_K}}.$$

Or, substituting in (6), we have

$$\Pr\bigg\{\frac{R_K - C}{\sqrt{3C^2 + CR_K}} > x\bigg\} \leq \Pr\bigg\{\frac{|A|}{\sqrt{B^2 + \mathbb{E}[B^2]}} > x\bigg\} \leq \sqrt{2}e^{-x^2/4}.$$

Next, notice that for $C > 0$, the function $f(y) = \frac{y-C}{\sqrt{3C^2+Cy}}$ is monotonically increasing for any $y > 0$:

$$f'(y) = \frac{\sqrt{3C^2 + Cy} - \frac{C(y-C)}{2\sqrt{3C^2+Cy}}}{3C^2 + Cy} = \frac{2(3C^2 + Cy) - Cy + C^2}{2(3C^2 + Cy)^{3/2}} = \frac{7C^2 + Cy}{2(3C^2 + Cy)^{3/2}} > 0$$

Moreover, for $y = C(1 + x)^2$,

$$f\big(C(1+x)^2\big) = \frac{C(1+x)^2 - C}{\sqrt{3C^2 + C^2(1+x)^2}} = \frac{Cx^2 + 2Cx}{\sqrt{4C^2 + 2C^2x + C^2x^2}}$$

$$> \frac{Cx^2 + 2Cx}{\sqrt{4C^2 + 4C^2x + C^2x^2}} = \frac{Cx^2 + 2Cx}{Cx + 2C} = x \ ,$$

where the inequality holds since $x > 0$. Thus, if $R_K \geq C(1 + x)^2$, then $f(R_K) > x$, and we can bound the probability that $R_K \geq C(1 + x)^2$ by

$$\Pr\big\{R_K \geq C(1+x)^2\big\} \leq \Pr\bigg\{\frac{R_K - C}{\sqrt{3C^2 + CR_K}} > x\bigg\} \leq \sqrt{2}e^{-x^2/4} \ ,$$

and setting $x = 2\sqrt{\ln\frac{2}{\delta}} > 0$, we obtain

$$\Pr\bigg\{R_K \geq C\bigg(1 + 2\sqrt{\ln\frac{2}{\delta}}\bigg)^2\bigg\} \leq \delta \ .$$

We remark that since $R_K$ is monotonically increasing a.s., this bound also implies that

$$\Pr\bigg\{\exists N : 1 \leq N \leq K, \ R_N \geq C\bigg(1 + 2\sqrt{\ln\frac{2}{\delta}}\bigg)^2\bigg\} \leq \delta.$$

To obtain a uniform bound, that is, bound that holds for all $K > 0$, note that the random sequence $Z_K = \mathbb{1}\left\{ \exists 1 \leq N \leq K : R_N \geq C\left(1 + 2\sqrt{\ln \frac{2}{\delta}}\right)^2 \right\}$ is monotonically increasing in $K$ and bounded. Thus, due to monotone convergence

$$\Pr\left\{ \exists K > 0 : R_K \geq C\left(1 + 2\sqrt{\ln \frac{2}{\delta}}\right)^2 \right\} = \mathbb{E}\left[ \lim_{K \to \infty} Z_K \right] = \lim_{K \to \infty} \mathbb{E}[Z_K]$$

$$= \lim_{K \to \infty} \Pr\left\{ \exists 1 \leq N \leq K : R_N \geq C\left(1 + 2\sqrt{\ln \frac{2}{\delta}}\right)^2 \right\} \leq \delta.$$

To conclude the proof, note that $\delta \leq 1$, and thus, $\ln \frac{3}{\delta} \geq 1$. Therefore, we can bound

$$C\left(1 + 2\sqrt{\ln \frac{2}{\delta}}\right)^2 \leq C\left(1 + 2\sqrt{\ln \frac{3}{\delta}}\right)^2 \leq C\left(3\sqrt{\ln \frac{3}{\delta}}\right)^2 = 9C \ln \frac{3}{\delta} \ ,$$

which yields the second bound. $\qquad\square$

**Lemma 11.** *Let $\{X_n^k\}_{k \geq 1}$ be a Bounded Decreasing Process in $[0, C]$ for any $n \in [N]$. The regret of the sum of processes is defined as $R(K) = \sum_{n=1}^N \sum_{k=1}^K X_n^{k-1} - \mathbb{E}[X_n^k \mid \mathcal{F}_{k-1}]$. Then, for any $\delta > 0$, we have*

$$\Pr\left\{ \exists K > 0 : R(K) \geq 9CN \ln \frac{3N}{\delta} \right\} \leq \delta.$$

*Proof.* We first remark that if $X_n^0 < C$, we can replace it to $X_n^0 = C$, which only increases the regret, so we assume w.l.o.g. that $X_n^0 = C$. Define

$$R_n(K) := \sum_{k=1}^K X_n^{k-1}(s) - \mathbb{E}[X_n^k(s) \mid \mathcal{F}_{k-1}].$$

Define the event $A_n := \left\{ \exists K > 0 : R_n(K) \geq 9CN \ln \frac{3N}{\delta} \right\}$. By applying Theorem 3, with probability $\frac{\delta}{N}$, it holds that for a fixed $n \in [N]$

$$\Pr\left\{ \exists K > 0 : R_n(K) \geq 9C \ln \frac{3N}{\delta} \right\} = \Pr\{A_n\} \leq \frac{\delta}{N}. \tag{7}$$

Finally, we obtain

$$\Pr\left\{ \exists K > 0 : R(K) \geq 9NC \ln \frac{3N}{\delta} \right\} = \Pr\left\{ \exists K > 0 : \sum_{n=1}^N R_n(K) \geq 9NC \ln \frac{3N}{\delta} \right\}$$

$$\overset{(1)}{\leq} \Pr\left\{ \bigcup_{n=1}^N A_n \right\} \overset{(2)}{\leq} \sum_{n=1}^N \Pr\{A_n\} \overset{(3)}{\leq} \delta.$$

Relation (1) holds since

$$\left\{ \exists K > 0 : \sum_{n=1}^N R_n(K) \geq 9NC \ln \frac{3N}{\delta} \right\} \subseteq \bigcup_{n=1}^N A_n.$$

In (2) we use the union bound and (3) holds by (7). $\qquad\square$

# B Proof of Real-Time Dynamic Programming Bounds

**Lemma 1.** *For all s, t, and k, it holds that (i) $V_t^*(s) \leq \bar{V}_t^k(s)$ and (ii) $\bar{V}_t^k(s) \leq \bar{V}_t^{k-1}(s)$.*

*Proof.* Both claims are proven using induction.

*(i)* By the initialization, $\forall s, t, \ V_t^*(s) \leq V_t^0(s)$. Assume the claim holds for $k-1$ episodes. Let $s_t^k$ be the state the algorithm is at in the $t^{th}$ time-step of the $k^{th}$ episode. By the value update of Algorithm 1,

$$\bar{V}_t^k(s_t^k) = \max_a \ r(s_t^k, a) + \sum_{s'} p(s' \mid s_t^k, a)\bar{V}_{t+1}^{k-1}(s')$$

$$\geq \max_a \ r(s_t^k, a) + \sum_{s'} p(s' \mid s_t^k, a)\bar{V}_{t+1}^*(s') = V^*(s_t^k).$$

The second relation holds by the induction hypothesis and the monotonicity of the optimal Bellman operator [Bertsekas and Tsitsiklis, 1996]. The third relation holds by the recursion satisfied by the optimal value function (see Section 2). Thus, the induction step is proven for the first claim.

*(ii)* To prove the base case of the second claim we use the optimistic initialization. Let $s_t^1$ be the state the algorithm is at in the $t^{th}$ time-step of the first episode. By the update rule,

$$\bar{V}_t^1(s_t^1) = \max_a \ r(s_t^1, a) + \sum_{s'} p(s' \mid s_t^1, a)\bar{V}_{t+1}^0(s')$$

$$\overset{(1)}{=} \max_a \ r(s_t^1, a) + H - t$$

$$\overset{(2)}{\leq} 1 + H - t = H - (t-1) \overset{(3)}{=} \bar{V}_t^0(s_t^1).$$

Relation (1) holds by the initialization of the values, (2) holds since $r(s, a) \in [0, 1]$ and (3) is by the initialization. States that were not visited on the first episode were not update, and thus the inequality trivially holds.

Assume the second claim holds for $k-1$ episodes. Let $s_t^k$ be the state that the algorithm is at in the $t^{th}$ time-step of the $k^{th}$ episode. By the value update of Algorithm 1, we have

$$\bar{V}_t^k(s_t^k) = \max_a \ r(s_t^k, a) + \sum_{s'} p(s' \mid s_t^k, a)\bar{V}_{t+1}^{k-1}(s').$$

If $s_t^k$ was previously updated, let $\bar{k}$ be the previous episode in which the update occured. By the induction hypothesis, we have that $\forall s, t, \ \bar{V}_t^{\bar{k}}(s) \geq \bar{V}_t^{k-1}(s)$. Using the monotonicity of the Bellman operator [Bertsekas and Tsitsiklis, 1996], we may write

$$\max_a \ r(s_t^k, a) + \sum_{s'} p(s' \mid s_t^k, a)\bar{V}_{t+1}^{k-1}(s')$$

$$\leq \max_a \ r(s_t^k, a) + \sum_{s'} p(s' \mid s_t^k, a)\bar{V}_{t+1}^{\bar{k}-1}(s') = \bar{V}^{k-1}(s_t^k).$$

Thus, $\bar{V}_t^k(s_t^k) \leq \bar{V}^{k-1}(s_t^k)$ and the induction step is proved. If $s_t^k$ was not previously updated, then $\bar{V}_t^{k-1}(s_t^k) = \bar{V}_t^0(s_t^k)$. In this case, the induction hypothesis implies that $\forall s', \bar{V}_{t+1}^{k-1}(s') \leq \bar{V}_{t+1}^0(s')$ and the result can be proven similarly to the base case. □

**Lemma 2** (Value Update for Exact Model). *The expected cumulative value update of RTDP at the $k^{th}$ episode satisfies*

$$\bar{V}_1^{k-1}(s_1^k) - V_1^{\pi_k}(s_1^k) = \sum_{t=1}^{H} \mathbb{E}[\bar{V}_t^{k-1}(s_t^k) - \bar{V}_t^k(s_t^k) \mid \mathcal{F}_{k-1}].$$

*Proof.* By the definition of $a_t^k$ and the update rule, the following holds:

$$\mathbb{E}[\bar{V}_t^k(s_t^k) \mid \mathcal{F}_{k-1}] = \mathbb{E}[r(s_t^k, a_t^k) + p(\cdot \mid s_t^k, a_t^k)^T \bar{V}_{t+1}^{k-1} \mid \mathcal{F}_{k-1}]$$

$$= \mathbb{E}[r(s_t^k, a_t^k) \mid \mathcal{F}_{k-1}] + \mathbb{E}\left[\sum_{\bar{s}_{t+1}} p(\bar{s}_{t+1} \mid s_t, \pi_k)\bar{V}_{t+1}^{k-1}(\bar{s}_{t+1}) \mid \mathcal{F}_{k-1}\right].$$

Furthermore,

$$\mathbb{E}\left[\sum_{\bar{s}_{t+1}} p(\bar{s}_{t+1} \mid s_t^k, \pi_k)\bar{V}_{t+1}^{k-1}(\bar{s}_{t+1}) \mid \mathcal{F}_{k-1}\right]$$

$$= \sum_{s_t^k} \Pr(s_t^k \mid s_1^k, \pi_k) \sum_{\bar{s}_{t+1} \in \mathcal{S}} p(\bar{s}_{t+1} \mid s_t^k, \pi_k)\bar{V}_{t+1}^{k-1}(\bar{s}_{t+1})$$

$$= \sum_{s_{t+1}^k \in \mathcal{S}} \Pr(s_{t+1}^k \mid s_1^k, \pi_k)\bar{V}_{t+1}^{k-1}(s_{t+1}^k) = \mathbb{E}[\bar{V}_{t+1}^{k-1}(s_{t+1}^k) \mid \mathcal{F}_{k-1}]. \tag{8}$$

The first relation holds by definition and the second one holds by the Markovian property of the dynamics. Substituting back and summing both side from $t = 1, \ldots, H$, we obtain

$$\mathbb{E}\left[\sum_{t=1}^{H} \bar{V}_t^k(s_t^k) \mid \mathcal{F}_{k-1}\right] = \mathbb{E}\left[\sum_{t=1}^{H} r(s_t^k, a_t^k) \mid \mathcal{F}_{k-1}\right] + \mathbb{E}\left[\sum_{t=1}^{H} \bar{V}_{t+1}^{k-1}(s_{t+1}^k) \mid \mathcal{F}_{k-1}\right]$$

$$= \mathbb{E}\left[\sum_{t=1}^{H} r(s_t^k, a_t^k) \mid \mathcal{F}_{k-1}\right] + \mathbb{E}\left[\sum_{t=1}^{H} \bar{V}_t^{k-1}(s_t^k) \mid \mathcal{F}_{k-1}\right] - \bar{V}_1^{k-1}(s_1^k)$$

$$= V_1^{\pi_k}(s_1^k) + \mathbb{E}\left[\sum_{t=1}^{H} \bar{V}_t^{k-1}(s_t^k) \mid \mathcal{F}_{k-1}\right] - \bar{V}_1^{k-1}(s_1^k)$$

The second line hold by shifting the index of the sum and using the fact that $\forall s, \bar{V}_{H+1}^k(s) = 0$. The third line holds by the definition of the value function,

$$\sum_{t=1}^{H} \mathbb{E}[r(s_t^k, a_t^k) \mid \mathcal{F}_{k-1}] = \mathbb{E}[\sum_{t=1}^{H} r(s_t^k, a_t^k) \mid s_1 = s_1^k] = V_1^{\pi_k}(s_1^k).$$

Reorganizing the equation yields the desired result. $\qquad\qquad\square$

**Theorem 4** (Regret Bounds for RTDP). *The following regret bounds hold for RTDP:*

1. $\mathbb{E}[\text{Regret}(K)] \leq SH^2$.

2. *For any $\delta > 0$, with probability $1 - \delta$, for all $K > 0$, $\text{Regret}(K) \leq 9SH^2 \ln(3SH/\delta)$.*

*Proof.* The following bounds on the regret hold.

$$\text{Regret}(K) := \sum_{k=1}^{K} V^*(s_1^k) - V^{\pi_k}(s_1^k) \leq \sum_{k=1}^{K} \bar{V}_1^{k-1}(s_1^k) - V^{\pi_k}(s_1^k)$$

$$\leq \sum_{k=1}^{K} \sum_{t=1}^{H} \mathbb{E}[\bar{V}_t^{k-1}(s_t^k) - \bar{V}_t^k(s_t^k) \mid \mathcal{F}_{k-1}]. \qquad (9)$$

The second relation is by the optimism of the value function (Lemma 1), and the third relation is by Lemma 2.

To prove the bound on the expected regret, we take expectation on both sides of (9). Thus,

$$\mathbb{E}[\text{Regret}(K)] \leq \sum_{k=1}^{K} \mathbb{E}[\mathbb{E}[\sum_{t=1}^{H} \bar{V}_t^{k-1}(s_t^k) - \bar{V}_t^k(s_t^k) \mid \mathcal{F}_{k-1}]] = \mathbb{E}[\sum_{k=1}^{K} \sum_{t=1}^{H} \bar{V}_t^{k-1}(s_t^k) - \bar{V}_t^k(s_t^k)].$$

Where the second relation holds by the tower property and linearity of expectation. Finally, for any run of RTDP, we have that

$$\sum_{k=1}^{K} \sum_{t=1}^{H} \bar{V}_t^{k-1}(s_t^k) - \bar{V}_t^k(s_t^k) = \sum_{s} \sum_{t=1}^{H} \bar{V}_t^0(s) - \bar{V}_t^K(s) \leq \sum_{s} \sum_{t=1}^{H} \bar{V}_t^0(s) - V_t^*(s) \leq SH^2.$$

The first relation holds since per $s$, the sum is telescopic, thus, only the first and last term exist in the sum. Due to the update rule, on the first time a state appears, its value will be $\bar{V}_t^0(s)$. From the last time it appears, its value will not be updated and thus the last value of a state is $\bar{V}_t^K(s)$. The second relation holds by Lemma 1. The third relation holds since $\forall s, t$, $\bar{V}_t^0(s) - V_t^*(s) \in [0, H]$, summing on $SH$ such terms yields the result.

To prove the high-probability bound we apply Lemma 34 by which,

$$(9) = \sum_{t=1}^{H} \sum_{s} \sum_{k=1}^{K} \bar{V}_t^{k-1}(s) - \mathbb{E}[\bar{V}_t^k(s) \mid \mathcal{F}_{k-1}].$$

For a fixed $s, t$, $\{\bar{V}_t^k(s)\}_{k \geq 0}$ is a Decreasing Bounded Process by Lemma 1, and its initial value is less than $H$. Thus, (9) is a sum of $SH$ Decreasing Bounded Processes. We apply Lemma 11 which provides a high-probability bound on a sum of Decreasing Bounded Processes to conclude the proof. $\qquad \square$

**Corollary 5** (RTDP is Uniform PAC). *Let $\delta > 0$ and $N_\epsilon$ be the number of episodes in which RTDP outputs a policy with $V_1^*(s_1^k) - V_1^{\pi_k}(s_1^k) > \epsilon$. Then,*

$$\Pr\left\{\exists \epsilon > 0 : N_\epsilon \geq \frac{9SH^2 \ln(3SH/\delta)}{\epsilon}\right\} \leq \delta.$$

*Proof.* Let $K_{N_\epsilon}$ be an episode index such that there are $N_\epsilon$ previous episodes $k \leq K_{N_\epsilon}$ in which RTDP outputs a policy with $V_1^*(s_1^k) - V_1^{\pi_k}(s_1^k) > \epsilon$. The following relation holds,

$$\forall \epsilon > 0 : kkvbvuvdh firinhblbkudchurbknbulr N_\epsilon \epsilon \leq \text{Regret}(K_{N_\epsilon}).$$

Thus,

$$\left\{\exists \epsilon > 0 : N_\epsilon \epsilon \geq 9SH^2 \ln \frac{3SH}{\delta}\right\} \subseteq \left\{\text{Regret}(K_{N_\epsilon}) \geq 9SH^2 \ln \frac{3SH}{\delta}\right\}$$

$$\subseteq \left\{\exists K > 0 : \text{Regret}(K) \geq 9SH^2 \ln \frac{3SH}{\delta}\right\}.$$

Which results in

$$\Pr\left\{\exists \epsilon > 0 : N_\epsilon \epsilon \geq 9SH^2 \ln \frac{3SH}{\delta}\right\} \leq \Pr\left\{\exists K > 0 : \text{Regret}(K) \geq 9SH^2 \ln \frac{3SH}{\delta}\right\} \leq \delta.$$

where the third relation holds by Theorem 4. $\qquad\square$

**Corollary 6** (RTDP is $(0, \delta)$ PAC). *Let $\delta > 0$ and $N$ be the number of episodes in which RTDP outputs a non optimal policy. Define the (unknown) gap of the MDP, $\Delta(\mathcal{M}) = \min_s \min_{\pi : V_1^\pi(s) \neq V_1^*(s)} V_1^*(s) - V_1^\pi(s) > 0$. Then,*

$$\Pr\left\{N \geq \frac{9SH^2 \ln(3SH/\delta)}{\Delta(\mathcal{M})}\right\} \leq \delta.$$

*Proof.* We have that $N = N_{\Delta(\mathcal{M})}$ since $\Delta(\mathcal{M})$ is the minimal gap; in all rest of episodes in which the gap is smaller than $\Delta(\mathcal{M})$, the policy $\pi_k$ is necessarily the optimal one. Based on Corollary 5 we conclude that,

$$\Pr\left\{N \geq \frac{9SH^2 \ln \frac{3SH}{\delta}}{\Delta(\mathcal{M})}\right\} = \Pr\left\{N_{\Delta(\mathcal{M})} \geq \frac{9SH^2 \ln \frac{3SH}{\delta}}{\Delta(\mathcal{M})}\right\} \leq \delta.$$

$\qquad\square$

## C Proofs of Section 4

**Lemma 7** (Value Update for Optimistic Model). *The expected cumulative value update of Algorithm 2 in the $k^{th}$ episode is bounded by*

$$\bar{V}_1^{k-1}(s_1^k) - V_1^{\pi_k}(s_1^k) \leq \sum_{t=1}^{H} \mathbb{E}\big[\bar{V}_t^{k-1}(s_t^k) - \bar{V}_t^k(s_t^k) \mid \mathcal{F}_{k-1}\big]$$

$$+ \sum_{t=1}^{H} \mathbb{E}\big[(\tilde{r}_{k-1} - r)(s_t^k, a_t^k) + (\tilde{p}_{k-1} - p)(\cdot \mid s_t^k, a_t^k)^T \bar{V}_{t+1}^{k-1} \mid \mathcal{F}_{k-1}\big] \ .$$

We prove a more general, Lemma 12, of which Lemma 7 is a direct corollay (by setting $t = 1$).

**Lemma 12.** *The expected value update of Algorithm 2 in the $k^{th}$ episode at the state $t^{th}$ is bounded by*

$$\bar{V}_t^{k-1}(s_t^k) - V_t^{\pi_k}(s_t^k) \leq \sum_{t'=t}^{H} \mathbb{E}\big[\bar{V}_{t'}^{k-1}(s_{t'}^k) - \bar{V}_{t'}^k(s_{t'}^k) \mid \mathcal{F}_{k-1}, s_t^k\big]$$

$$+ \sum_{t'=t}^{H} \mathbb{E}\big[(\tilde{r}_{k-1} - r)(s_{t'}^k, a_{t'}^k) + (\tilde{p}_{k-1} - p)(\cdot \mid s_{t'}^k, a_{t'}^k)^T \bar{V}_{t'+1}^{k-1} \mid \mathcal{F}_{k-1}, s_t^k\big] \ .$$

*Proof.* We closely follow the proof of Lemma 2. By the definition of $a_t^k$ and the update rule, for $t' \geq t$, the following holds.

$$\mathbb{E}\big[\bar{V}_{t'}^k(s_{t'}^k) \mid \mathcal{F}_{k-1}, s_t^k\big]$$

$$\overset{(1)}{\leq} \mathbb{E}\Bigg[\tilde{r}_{k-1}(s_{t'}^k, a_{t'}^k) + \sum_{\bar{s}_{t'+1} \in \mathcal{S}} \tilde{p}_{k-1}(\bar{s}_{t'+1} \mid s_{t'}^k, a_{t'}) \bar{V}_{t'+1}^{k-1}(\bar{s}_{t'+1}) \mid \mathcal{F}_{k-1}, s_t^k\Bigg]$$

$$\overset{(2)}{=} \mathbb{E}\big[r(s_{t'}^k, a_{t'}^k) \mid \mathcal{F}_{k-1}, s_{t'}^k\big] + \mathbb{E}\Bigg[\sum_{\bar{s}_{t'+1} \in \mathcal{S}} p(\bar{s}_{t'+1} \mid s_{t'}^k, a_{t'}^k) \bar{V}_{t'+1}^{k-1}(\bar{s}_{t'+1}) \mid \mathcal{F}_{k-1}, s_t^k\Bigg]$$

$$+ \mathbb{E}\Bigg[(\tilde{r}_{k-1} - r)(s_{t'}^k, a_{t'}^k) + \sum_{\bar{s}_{t'+1} \in \mathcal{S}} (\tilde{p}_{k-1} - p)(\bar{s}_{t'+1} \mid s_{t'}^k, a_{t'}^k) \bar{V}_{t'+1}^{k-1}(\bar{s}_{t'+1}) \mid \mathcal{F}_{k-1}, s_t^k\Bigg]$$

$$\overset{(3)}{=} \mathbb{E}\big[r(s_{t'}^k, a_{t'}^k) \mid \mathcal{F}_{k-1}, , s_t^k\big] + \mathbb{E}\big[\bar{V}_{t'+1}^{k-1}(s_{t'+1}^k) \mid \mathcal{F}_{k-1}, s_t^k\big]$$

$$+ \mathbb{E}\Bigg[(\tilde{r}_{k-1} - r)(s_{t'}^k, a_{t'}^k) + \sum_{\bar{s}_{t'+1} \in \mathcal{S}} (\tilde{p}_{k-1} - p)(\bar{s}_{t'+1} \mid s_{t'}^k, a_{t'}^k) \bar{V}_{t'+1}^{k-1}(\bar{s}_{t'+1}) \mid \mathcal{F}_{k-1}, s_t^k\Bigg]$$

Relation (1) holds by the update rule for $\bar{V}_t^k$. Next, (2) holds by adding an subtracting the real reward and dynamics and using linearity of expectation. In (3), we used the same reasoning as in Equation (8).

Summing both side from $t' = t, \ldots, H$, we obtain:

$$\mathbb{E}\left[\sum_{t'=t}^{H} \bar{V}_{t'}^{k}(s_{t'}^{k}) \mid \mathcal{F}_{k-1}, s_t^k\right]$$

$$\leq \mathbb{E}\left[\sum_{t'=t}^{H} r(s_{t'}^k, a_{t'}^k) \mid \mathcal{F}_{k-1}, s_t^k\right] + \mathbb{E}\left[\sum_{t'=t}^{H} \bar{V}_{t'+1}^{k-1}(s_{t'+1}^k) \mid \mathcal{F}_{k-1}, s_t^k\right]$$

$$+ \sum_{t'=t}^{H} \mathbb{E}\left[(\tilde{r}_{k-1} - r)(s_{t'}^k, a_{t'}^k) + \sum_{\bar{s}_{t'+1}\in\mathcal{S}} (\tilde{p}_{k-1} - p)(\bar{s}_{t'+1} \mid s_{t'}^k, a_{t'}^k) \bar{V}_{t'+1}^{k-1}(\bar{s}_{t'+1}) \mid \mathcal{F}_{k-1}, s_t^k\right]$$

$$\overset{(1)}{=} V_t^{\pi_k}(s_t^k) + E\left[\sum_{t'=t}^{H} \bar{V}_{t'+1}^{k-1}(s_{t'+1}^k) \mid \mathcal{F}_{k-1}, s_t^k\right]$$

$$+ \sum_{t'=t}^{H} \mathbb{E}\left[(\tilde{r}_{k-1} - r)(s_{t'}^k, a_{t'}^k) + \sum_{\bar{s}_{t'+1}\in\mathcal{S}} (\tilde{p}_{k-1} - p)(\bar{s}_{t'+1} \mid s_{t'}^k, a_{t'}^k) \bar{V}_{t'+1}^{k-1}(\bar{s}_{t'+1}) \mid \mathcal{F}_{k-1}, s_t^k\right]$$

$$\overset{(2)}{=} V_t^{\pi_k}(s_t^k) + \mathbb{E}\left[\sum_{t'=t}^{H} \bar{V}_{t'}^{k-1}(s_{t'}^k) \mid \mathcal{F}_{k-1}, s_t^k\right] - \bar{V}_t^{k-1}(s_t^k)$$

$$+ \sum_{t'=t}^{H} \mathbb{E}\left[(\tilde{r}_{k-1} - r)(s_{t'}^k, a_{t'}^k) + \sum_{\bar{s}_{t'+1}\in\mathcal{S}} (\tilde{p}_{k-1} - p)(\bar{s}_{t'+1} \mid s_{t'}^k, a_{t'}^k) \bar{V}_{t'+1}^{k-1}(\bar{s}_{t'+1}) \mid \mathcal{F}_{k-1}, s_t^k\right]$$

In (1) we used the fact that $V_t^{\pi_k}(s_t^k) = \mathbb{E}[\sum_{t'=t}^{H} r(s_{t'}^k, \pi_k(s_{t'}^k)) \mid \mathcal{F}_{k-1}, s_t^k]$. Relation (2) holds by shifting the index of the sum and using $\forall s, k, \bar{V}_{H+1}^{k-1}(s) = 0$. Reorganizing the equation yields the desired result. $\qquad\square$

# D Proof of Theorem 8

---

**Algorithm 4** UCRL2 with Greedy Policies

---

1: Initialize: $\delta, \delta' = \frac{\delta}{4} \forall s \in \mathcal{S}, t \in [H], \bar{V}_t^0(s) = H - (t-1)$.
2: **for** $k = 1, 2, ..$ **do**
3:    Initialize $s_1^k$
4:    **for** $t = 1, .., H$ **do**
5:        #Update Upper Bound on $V^*$
6:        **for** $a \in \mathcal{A}$ **do**
7:            $\tilde{r}_{k-1}(s_t^k, a) = \hat{r}_{k-1}(s_t^k, a) + \sqrt{\frac{2 \ln \frac{2SAT}{\delta'}}{n_{k-1}(s_t^k, a) \vee 1}}$
8:            $CI(s_t^k, a) = \left\{ P' \in \mathcal{P}(\mathcal{S}) : \|P'(\cdot) - \hat{p}_{k-1}(\cdot \mid s_t^k, a)\|_1 \leq \sqrt{\frac{4S \ln \frac{3SAT}{\delta'}}{n_{k-1}(s_t^k, a) \vee 1}} \right\}$
9:            $\tilde{p}_{k-1}(\cdot \mid s_t^k, a) = \arg\max_{P' \in CI(s_t^k, a)} P'(\cdot \mid s_t^k, a)^T \bar{V}_{t+1}^{k-1}$
10:           $\bar{Q}(s_t^k, a) = \tilde{r}_{k-1}(s_t^k, a) + \tilde{p}_{k-1}(\cdot \mid s_t^k, a)^T \bar{V}_{t+1}^{k-1}$
11:       **end for**
12:       $a_t^k \in \arg\max_a \bar{Q}(s_t^k, a)$
13:       $\bar{V}_t^k(s_t^k) = \min\{\bar{V}_t^{k-1}(s_1^k), \bar{Q}(s_t^k, a_t^k)\}$
14:       #Act by the Greedy Policy
15:       Apply $a_t^k$ and observe $s_{t+1}^k$.
16:    **end for**
17:    Update $\hat{r}_k, \hat{p}_k, n_k$ with all experience gathered in the episode.
18: **end for**

---

We provide the full proof of Theorem 8 which establishes a regret bound for UCRL2 with Greedy Policies (UCRL2-GP) in finite horizon MDPs. In the following, we present the structure of this section.

We define the failure events for UCRL2-GP in Section D.1. Most of the events are standard low-probability failure events, derived using, e.g., Hoeffding's inequality. We add to the standard set of events a failure event which holds when a sum of Decreasing Bounded Processes is large in its value. Using uniform bounds, the failure events are shown to hold jointly. When all failure events do not occur for all time-steps we say the algorithm is outside the failure event. In Section D.2 we establish that UCRL2-GP is optimistic, and, more specifically, that for all $s, t, k$ $\bar{V}_t^k(s) \geq V_t^*(s)$, outside the failure event. Lastly, in Section D.3 we give the full proof of Theorem 8, based on a new regret decomposition using on Lemma 7, the new results on Decreasing Bounded Processes (see Appendix A), and existing techniques (e.g., [Dann et al., 2017, Zanette and Brunskill, 2019]).

## D.1 Failure Events for UCRL2 with Greedy Policies

Define the following failure events.

$$F_k^r = \left\{ \exists s, a : |r(s, a) - \hat{r}_{k-1}(s, a)| \geq \sqrt{\frac{2 \ln \frac{2SAT}{\delta'}}{n_{k-1}(s, a) \vee 1}} \right\}$$

$$F_k^p = \left\{ \exists s, a : \|p(\cdot \mid s, a) - \hat{p}_{k-1}(\cdot \mid s, a)\|_1 \geq \sqrt{\frac{4S \ln \frac{3SAT}{\delta'}}{n_{k-1}(s, a) \vee 1}} \right\}$$

$$F_k^N = \left\{ \exists s, a : n_{k-1}(s, a) \leq \frac{1}{2} \sum_{j < k} w_j(s, a) - H \ln \frac{SAH}{\delta'} \right\}.$$

$$F^{DBP} = \left\{ \exists k > 0 : \sum_{k=1}^{K} \sum_{t=1}^{H} \sum_s \bar{V}_t^{k-1}(s) - \mathbb{E}[\bar{V}_t^k(s) \mid \mathcal{F}_{k-1}] \geq 9SH^2 \ln \frac{3SH}{\delta'} \right\}$$

Furthermore, the following relations hold.

- Let $F^r = \bigcup_{k=1}^K F_k^r$. Then $\Pr\{F^r\} \le \delta'$, by Hoeffding's inequality, and using a union bound argument on all $s, a$, possible values of $n_k(s, a)$ and $k$. Furthermore, for $n(s, a) = 0$ the bound holds trivially since $R \in [0, 1]$.

- Let $F^P = \bigcup_{k=1}^K F_k^p$. Then $\Pr\{F^p\} \le \delta'$, holds by [Weissman et al., 2003] while applying union bound on all $s, a, n_{k-1}(s, a)$ and possible values of $k$ (e.g., Azar et al. 2017, Zanette and Brunskill 2019). Furthermore, for $n(s, a) = 0$ the bound holds trivially.

- Let $F^N = \bigcup_{k=1}^K F_k^N$. Then, $\Pr\{F^N\} \le \delta'$. The proof is given in [Dann et al., 2017] Corollary E.4 (and is used in Zanette and Brunskill 2019 Appendix D.4).

- By construction of Algorithm 2, $\forall s, t, \bar{V}_t^k(s)$ is a decreasing function of $k$, with $\bar{V}_t^0(s) = H$. Furthermore, since $\hat{r}_{k-1}(s, a)$ and $\hat{p}_{k-1}(\cdot \mid s, a)$ are non-negative, and $\bar{V}_t^0(s) > 0$, a simple induction allows us to conclude that $\forall s, t, \bar{V}_t^k(s) \ge 0$. Thus, by applying Lemma 11, $\Pr\{F^{DBP}\} \le \delta'$.

**Lemma 13.** *Setting $\delta' = \frac{\delta}{4}$ then $\Pr\{F^r \bigcup F^p \bigcup F^N \bigcup F^{DBP}\} \le \delta$. When the failure events does not hold we say the algorithm is outside the failure event.*

### D.2 UCRL2 with Greedy Policies is Optimistic

**Lemma 14.** *Outside the failure event UCRL2-GP is Optimistic,*

$$\forall s, t, k \; \bar{V}_t^k(s) \ge V_t^*(s).$$

*Proof.* We prove by induction. The base case holds by the initialization of the algorithm, $\bar{V}_t^0(s) = H - (t-1) \ge V_t^*(s)$. Assume the induction hypothesis holds for $k-1$ episodes. At the $k^{th}$ episode, states there were not visited at step $t$ will not be updated, and thus by the induction hypothesis, the result hold for these states. For states that were visited, if the minimum at the update stage equals to $\bar{V}_t^{k-1}(s)$, then the result similarly holds. Let $s_t^k$ be a state that was updated according to the optimistic model, and let

$$\tilde{a}^* \in \arg\max_a \tilde{r}_{k-1}(s_t^k, a) + \tilde{p}_{k-1}(\cdot \mid s_t^k, a) v_{t+1}^{k-1}$$
$$a^* \in \arg\max_a r(s_t^k, a) + p(\cdot \mid s_t^k, a) V_{t+1}^*.$$

Then,

$$
\begin{aligned}
\bar{V}_t^k(s_t^k) &= \max_a \tilde{r}_{k-1}(s_t^k, a) + \tilde{p}_{k-1}(\cdot \mid s_t^k, a) \bar{V}_{t+1}^{k-1} \\
&\overset{(1)}{=} \tilde{r}_{k-1}(s_t^k, \tilde{a}^*) + \tilde{p}_{k-1}(\cdot \mid s_t^k, \tilde{a}^*) \bar{V}_{t+1}^{k-1} \\
&\overset{(2)}{\ge} \tilde{r}_{k-1}(s_t^k, a^*) + \tilde{p}_{k-1}(\cdot \mid s_t^k, a^*) \bar{V}_{t+1}^{k-1} \\
&\overset{(3)}{\ge} r(s_t^k, a^*) + p(\cdot \mid s_t^k, a^*) \bar{V}_{t+1}^{k-1} \\
&\overset{(4)}{\ge} r(s_t^k, a^*) + p(\cdot \mid s_t^k, a^*) V_{t+1}^* \\
&\overset{(5)}{=} V_t^*(s_t^k).
\end{aligned}
$$

Relations (1) and (2) are by the definition and optimality of $\tilde{a}^*$, respectively. (3) holds since outside failure event $F_k^r$, $\tilde{r}_{k-1}(s_t^k, a^*) \ge r(s_t^k, a^*)$. Furthermore, outside failure event $F_k^p$, the real transition probabilities $p(\cdot \mid s, a^*) \in CI(s_t^k, a^*)$, and thus

$$\max_{P' \in CI(s_t^k, a^*)} P'(\cdot \mid s_t^k, a^*) \bar{V}_{t+1}^{k-1} = \tilde{p}_{k-1}(\cdot \mid s_t^k, a^*) \bar{V}_{t+1}^{k-1} \ge p_{k-1}(\cdot \mid s_t^k, a^*) \bar{V}_{t+1}^{k-1}.$$

Finally, (4) holds by the induction hypothesis $\forall s, a, t, \bar{V}_t^{k-1}(s) \ge V_t^*(s)$ and (5) holds by the Bellman recursion. $\qquad\square$

## D.3 Proof of Theorem 8

**Theorem 8** (Regret Bound of UCRL2-GP). *For any time $T \leq KH$, with probability at least $1 - \delta$, the regret of UCRL2-GP is bounded by $\tilde{\mathcal{O}}\left(HS\sqrt{AT} + H^2\sqrt{S}SA\right)$.*

*Proof.* By the optimism of the value (Lemma 14), we have that

$$
\sum_{k=1}^{K} V_1^*(s_1^k) - V_1^{\pi_k}(s_1^k) \leq \sum_{k=1}^{K} \bar{V}_1^{k-1}(s_1^k) - V_1^{\pi_k}(s_1^k)
$$

$$
\leq \underbrace{\sum_{k=1}^{K}\sum_{t=1}^{H} \mathbb{E}[\bar{V}_t^{k-1}(s_t^k) - \bar{V}_t^k(s_t^k) \mid \mathcal{F}_{k-1}]}_{(A)}
$$

$$
+ \underbrace{\sum_{k=1}^{K}\sum_{t=1}^{H} \mathbb{E}\left[(\tilde{r}_{k-1} - r)(s_t^k, a_t^k) + (\tilde{p}_{k-1} - p)(\cdot \mid s_t^k, a_t^k)^T \bar{V}_{t+1}^{k-1} \mid \mathcal{F}_{k-1}\right]}_{(B)} . \quad (10)
$$

The first relation is by the optimism of the value, and the second relation is by Lemma 7. We now bound the two terms outside the failure event.

**Bounding (A).** By Lemma 34 (Appendix F),

$$
(A) = \sum_{k=1}^{K}\sum_{t=1}^{H}\sum_{s} \bar{V}_t^{k-1}(s) - \mathbb{E}[\bar{V}_t^k(s) \mid \mathcal{F}_{k-1}].
$$

Outside the failure event, the sum is bounded by $9SH^2 \ln \frac{3SH}{\delta'}$ (see event $F^{DBP}$). Thus,

$$
(A) \leq \tilde{\mathcal{O}}(SH^2).
$$

**Bounding (B).** Outside failure event $F_k^r$ the following inequality holds:

$$
\sum_{k=1}^{K}\sum_{t=1}^{H} \mathbb{E}\left[(\tilde{r}_{k-1} - r)(s_t^k, \pi_k(s_t^k) \mid \mathcal{F}_{k-1}\right]
$$

$$
\lesssim \sum_{k=1}^{K}\sum_{t=1}^{H} \mathbb{E}\left[\sqrt{\frac{1}{n_{k-1}(s_t^k, \pi_k(s_t^k)) \vee 1}} \mid \mathcal{F}_{k-1}\right] \lesssim \tilde{\mathcal{O}}(\sqrt{SAT} + SAH), \quad (11)
$$

where the second inequality is by Lemma 38. It is worth noting that Lemma 38 is proven by defining $L_k$, the set of 'good' state-action pairs, that contains pairs that were visited sufficiently often in the past [Dann et al., 2017, Zanette and Brunskill, 2019]. The term we bound is then analyzed separately for state-action pairs inside and outside $L_k$. The definition of $L_k$ can be found in Definition 2, and its properties (including Lemma) are analyzed in Appendix F.1.

Furthermore, outside the failure event,

$$
\sum_{k=1}^{K} \sum_{t=1}^{H} \mathbb{E}\left[(\tilde{p}_{k-1} - p)(\cdot \mid s_t^k, a_t^k)^T \bar{V}_{t+1}^{k-1} \mid \mathcal{F}_{k-1}\right]
$$

$$
= \sum_{k=1}^{K} \sum_{t=1}^{H} \mathbb{E}\left[(\tilde{p}_{k-1} - \hat{p}_{k-1})(\cdot \mid s_t^k, a_t^k)^T \bar{V}_{t+1}^{k-1} \mid \mathcal{F}_{k-1}\right]
$$

$$
+ \mathbb{E}\left[(\hat{p}_{k-1} - p)(\cdot \mid s_t^k, a_t^k)^T \bar{V}_{t+1}^{k-1} \mid \mathcal{F}_{k-1}\right]
$$

$$
\overset{(1)}{\leq} \sum_{k=1}^{K} \sum_{t=1}^{H} \mathbb{E}\left[\|(\tilde{p}_{k-1} - \hat{p}_{k-1})(\cdot \mid s_t^k, a_t^k)\|_1 \|\bar{V}_{t+1}^{k-1}\|_\infty \mid \mathcal{F}_{k-1}\right]
$$

$$
+ \mathbb{E}\left[\|(\hat{p}_{k-1} - p)(\cdot \mid s_t^k, a_t^k)\| \|\bar{V}_{t+1}^{k-1}\|_\infty \mid \mathcal{F}_{k-1}\right]
$$

$$
\overset{(2)}{\leq} H \sum_{k=1}^{K} \sum_{t=1}^{H} \mathbb{E}\left[\|(\hat{p}_{k-1} - p)(\cdot \mid s_t^k, a_t^k)\|_1 + \|(\tilde{p}_{k-1} - \hat{p}_{k-1})(\cdot \mid s_t^k, a_t^k)\|_1 \mid \mathcal{F}_{k-1}\right]
$$

$$
\overset{(3)}{\lesssim} H\sqrt{S} \sum_{k=1}^{K} \sum_{t=1}^{H} \mathbb{E}\left[\sqrt{\frac{1}{n_{k-1}(s_t^k, a_t^k) \vee 1}} \mid \mathcal{F}_{k-1}\right]
$$

$$
\overset{(4)}{\lesssim} \tilde{\mathcal{O}}(HS\sqrt{AT} + H^2\sqrt{S}SA). \tag{12}
$$

Relation (1) holds by Hölder's inequality. Next, (2) holds since $\forall s, t, k, \ 0 \leq \bar{V}_t^k(s) \leq H$. The lower bounds holds by Lemma 14 and since $V_t^* \geq 0$. The upper bound is since the value can only decrease by Algorithm 2 and the inequality holds for the initialized value. Finally, (3) holds outside failure event $F^p$ (Lemma 13), and (4) holds by Lemma 38.

Combining (11), (12) we conclude that,

$$
(B) \leq \tilde{\mathcal{O}}(HS\sqrt{AT} + H^2\sqrt{S}SA)
$$

Combining the bounds on (A) and (B) in (10) concludes the proof. □

# E   Proof of Theorem 9

---

**Algorithm 5** EULER with Greedy Policies

---

1: Initialize: $\delta, \delta' = \frac{\delta}{9}, \forall s \in \mathcal{S}, t \in [H], \bar{V}_t^0(s) = H - (t-1), \underline{V}_t^0(s) = 0,$

2:
$$\phi(s,a) = \sqrt{\frac{2\hat{\mathrm{Var}}_{\hat{p}_{k-1}(s,a)}(\bar{V}_{t+1}^{k-1})\ln\frac{4SAT}{\delta'}}{n_{k-1}(s,a)}} + \frac{2H\ln\frac{4SAT}{\delta'}}{3n_{k-1}(s,a)}, L = 2\sqrt{\ln\frac{4SAT}{\delta'}},$$

3:
$$J = \frac{2H\ln\frac{4SAT}{\delta'}}{3}, B_v = \sqrt{2\ln\frac{4SAT}{\delta'}}, B_p = H\sqrt{2\ln\frac{4SAT}{\delta'}}.$$

4: **for** $k = 1, 2, ..$ **do**

5:     Initialize $s_1^k$

6:     **for** $t = 1, .., H$ **do**

7:         #Update Upper Bound on $V^*$

8:         **for** $a \in \mathcal{A}$ **do**

9:
$$b_k^r(s_t^k, a) = \sqrt{\frac{2\hat{\mathrm{Var}}(R(s_t^k,a))\ln\frac{4SAT}{\delta'}}{n_{k-1}(s_t^k,a)\vee 1}} + \frac{14\ln\frac{4SAT}{\delta'}}{3n_{k-1}(s_t^k,a)\vee 1}$$

10:
$$b_k^{pv}(s_t^k, a) = \phi(\hat{p}_{k-1}(\cdot \mid s_t^k, a), \bar{V}_{t+1}^{k-1}) + \frac{4J+B_p}{n_{k-1}(s_t^k,a)\vee 1} + \frac{B_v\|\bar{V}_{t+1}^{k-1} - \underline{V}_{t+1}^{k-1}\|_{2,\hat{p}}}{\sqrt{n_{k-1}(s_t^k,a)\vee 1}}$$

11:
$$\bar{Q}(s_t^k, a) = \hat{r}_{k-1}(s_t^k, a) + b_k^r(s_t^k, a) + \hat{p}_{k-1}(\cdot \mid s_t^k, a)^T \bar{V}_{t+1}^{k-1} + b_k^{pv}(s_t^k, a)$$

12:         **end for**

13:         $a_t^k \in \arg\max_a \bar{Q}(s_t^k, a)$

14:         $\bar{V}_t^k(s_t^k) = \min\{\bar{V}_t^{k-1}(s_t^k), \bar{Q}(s_t^k, a_t^k)\}$

15:         #Update Lower Bound on $V^*$

16:
$$b_k^{pv}(s_t^k, a_t^k) = \phi(\hat{p}_{k-1}(\cdot \mid s_t^k, a_t^k), \underline{V}_{t+1}^{k-1}) + \frac{4J+B_p}{n_{k-1}(s_t^k,a_t^k)\vee 1} + \frac{B_v\|\bar{V}_{t+1}^{k-1} - \underline{V}_{t+1}^{k-1}\|_{2,\hat{p}}}{\sqrt{n_{k-1}(s_t^k,a_t^k)\vee 1}}$$

17:
$$\underline{Q}(s_t^k, a_t^k) = \hat{r}_{k-1}(s_t^k, a_t^k) - b_k^r(s_t^k, a_t^k) + \hat{p}_{k-1}(\cdot \mid s_t^k, a)^T \underline{V}_{t+1}^{k-1} - b_k^{pv}(s_t^k, a_t^k)$$

18:
$$\underline{V}_t^k(s_t^k) = \max\{\underline{V}_t^{k-1}(s_t^k), \underline{Q}(s_t^k, a_t^k)\}$$

19:         #Act by the Greedy Policy

20:         Apply $a_t^k$ and observe $s_{t+1}^k$.

21:     **end for**

22:     Update $\hat{r}_k, \hat{p}_k, n_k$ with all experience gathered in episode.

23: **end for**

---

**Remark 1.** *Note that the algorithm does not explicitly define $\tilde{r}_{k-1}(s,a)$ and $\tilde{p}_{k-1}(\cdot \mid s, a)$. While we can directly set $\tilde{r}_{k-1}(s,a) = \hat{r}_{k-1}(s,a) + b_k^r(s_t^k, a)$, the optimistic transition kernel is only implicitly defined as*

$$\tilde{p}_{k-1}(\cdot \mid s, a)^T \bar{V}_t^{k-1} = \hat{p}_{k-1}(\cdot \mid s, a)^T \bar{V}_{t+1}^{k-1} + b_k^{pv}(s,a)$$

*Nevertheless, throughout the proofs we are only interested in the above quantity, and thus, except for some abuse of notation, all of the proofs hold. We use this notation since it is common in previous works (e.g., Zanette and Brunskill 2019, Dann et al. 2017) and for brevity.*

In this section, we provide the full proof of Theorem 9 which establishes a regret bound for EULER with Greedy Policies (EULER-GP). In Zanette and Brunskill [2019] the authors prove their results using a general confidence interval, which they refer as *admissible confidence interval*. In Section E.1 we state there definition and state some useful properties. In Section E.2 we define the set of failure events and show that with high-probability the failure events do not occur. The set of failure events includes high-probability events derived using empirical Bernstein inequalities [Maurer and Pontil, 2009], as well as high probability events on Decreasing Bounded Process, as we establish in Appendix A. In Section E.3 we analyze the optimism EULER-GP and prove it satisfies the same optimism and pessimism as in Zanette and Brunskill [2019], outside the failure event for all $s, t, k$
$\underline{V}_t^k(s) \leq V_t^*(s) \leq \bar{V}_t^k(s)$.

In Section E.4, using the above, we give the full proof of Theorem 9. As for the proof of UCRL2-GP, we apply the new suggested regret decomposition, based on Lemma 7, and use the new results on Decreasing Bounded Processes. In section E.5 we modify some results of [Zanette and Brunskill, 2019] to our setting, and utilize the new results to bound each term in the regret decomposition in section E.6.

### E.1 Properties of Confidence Intervals

In this section, we cite the important properties of the confidence intervals of EULER, as was stated in [Zanette and Brunskill, 2019]. We start from their definition of an admissible confidence interval:

**Definition 1.** *A confidence interval $\phi$ is called admissible for EULER if the following properties hold:*

1. *$\phi(p, V)$ takes the following functional form:*

$$\phi(p, V) = \frac{g(p, V)}{\sqrt{n_{k-1}(s, a) \vee 1}} + \frac{j(p, V)}{n_{k-1}(s, a) \vee 1} \quad ,$$

   *for some functions $j(p, V) \leq J \in \mathbb{R}$, and*

$$|g(p, V_1) - g(p, V_2)| \leq B_v \|V_1 - V_2\|_{2, p}$$

   *If the value function is uniform then:*

$$g(p, \alpha \mathbb{1}) = 0, \quad \forall \alpha \in \mathbb{R}.$$

2. *With probability at least $1 - \delta'$ it holds that:*

$$|(\hat{p}_{k-1}(\cdot \mid s, a) - p(\cdot \mid s, a))^T V_{t+1}^*| \leq \phi(p(\cdot \mid s, a), V_{t+1}^*)$$

   *jointly for all timesteps $t$, episodes $k$, states $s$ and actions $a$.*

3. *With probability at least $1 - \delta'$ it holds that:*

$$\left| g(\hat{p}_{k-1}(\cdot \mid s, a), V_{t+1}^*) - g(p(\cdot \mid s, a), V_{t+1}^*) \right| \leq \frac{B_p}{\sqrt{n_{k-1}(s, a) \vee 1}}$$

   *jointly for all episodes $k$, timesteps $t$, states $s$, actions $a$ and some constant $B_p$ that does not depend on $\sqrt{n_{k-1}(s, a) \vee 1}$.*

An admissible confidence interval enjoys many properties, which are summarized in the following lemma:

**Lemma 15.** *If $\phi$ is admissible for EULER, and under the events that properties 2,3 of Definition 1 hold, then:*

1. *For any $V \in \mathbb{R}^S$ with $\|V\|_\infty \leq H$, it holds that $|g(p, V)| \leq B_v H$.*

2. *For any $V \in \mathbb{R}^S$,*

$$|\phi(\hat{p}_{k-1}(\cdot \mid s, a), V) - \phi(p(\cdot \mid s, a), V_{t+1}^*)| \leq \frac{B_v \|V - V_{t+1}^*\|_{2, \hat{p}}}{\sqrt{n_{k-1}(s, a) \vee 1}} + \frac{B_p + 4J}{n_{k-1}(s, a) \vee 1}$$

3. *Let $b_k^{pv}$ be the transition bonus, which is defined as*

$$b_k^{pv}(\hat{p}_{k-1}(\cdot \mid s, a), V_1, V_2) = \phi(\hat{p}_{k-1}(\cdot \mid s, a), V_1) + \frac{B_p + 4J}{n_{k-1}(s, a) \vee 1} + \frac{B_v \|V_2 - V_1\|_{2, \hat{p}}}{\sqrt{n_{k-1}(s, a) \vee 1}} \quad .$$

   *For any $V_1, V_2 \in \mathbb{R}^S$ such that $V_1 \leq V^* \leq V_2$ pointwise, it holds that*

$$b_k^{pv}(\hat{p}_{k-1}(\cdot \mid s, a), V_1, V_2) \geq \phi(p(\cdot \mid s, a), V^*)$$
$$b_k^{pv}(\hat{p}_{k-1}(\cdot \mid s, a), V_2, V_1) \geq \phi(p(\cdot \mid s, a), V^*)$$

*Proof.* The first property is due to Corollary 1.3 of [Zanette and Brunskill, 2019], and the second one is Lemma 4 of their paper. The third property is equivalent to Proposition 3 in [Zanette and Brunskill, 2019], but since we allow general value functions $V_1, V_2$, we write the full proof for completeness.

we start by proving that if $V_1 \leq V^* \leq V_2$, then for any transition probability vector $p$,

$$\|V_2 - V^*\|_{2,p} \leq \|V_2 - V_1\|_{2,p}$$
$$\|V_1 - V^*\|_{2,p} \leq \|V_2 - V_1\|_{2,p} \tag{13}$$

To this end, notice that $\forall s$

$$0 \leq V_2(s) - V^*(s) \leq V_2(s) - V_1(s) \ ,$$

and since all of the quantities are non-negative, it also holds that

$$0 \leq (V_2(s) - V^*(s))^2 \leq (V_2(s) - V_1(s))^2 \ .$$

The inequality holds pointwise, and therefore holds for any linear combination with non-negative constants:

$$0 \leq \sum_s p(s)(V_2(s) - V^*(s))^2 \leq \sum_s p(s)(V_2(s) - V_1(s))^2 \ .$$

Taking the root of this inequality yields Inequality (13). Substituting in the definition of $b_k^{pv}(\hat{p}_{k-1}(\cdot \mid s, a), V_2, V_1)$ yields:

$$b_k^{pv}(\hat{p}_{k-1}(\cdot \mid s, a), V_2, V_1) = \phi(\hat{p}_{k-1}(\cdot \mid s, a), V_2) + \frac{B_p + 4J}{n_{k-1}(s,a) \vee 1} + \frac{B_v \|V_2 - V_1\|_{2,\hat{p}}}{\sqrt{n_{k-1}(s,a) \vee 1}}$$

$$\geq \phi(\hat{p}_{k-1}(\cdot \mid s, a), V_2) + \frac{B_p + 4J}{n_{k-1}(s,a) \vee 1} + \frac{B_v \|V_2 - V^*\|_{2,\hat{p}}}{\sqrt{n_{k-1}(s,a) \vee 1}}$$

$$\geq \phi(p(\cdot \mid s, a), V^*)$$

The first inequality is due to (13), and the second is due to the second part of the Lemma. The result for $b_k^{pv}(\hat{p}_{k-1}(\cdot \mid s, a), V_1, V_2)$ can be proven similarly, and thus omitted.

$\square$

Another property that will be useful throughout the proof is the following upper bound on $b_k^{pv}(\hat{p}_{k-1}(\cdot \mid s, a), V_1, V_2)$

**Lemma 16.** *For any $V_1, V_2$ such that for all $s$, $V_1(s), V_2(s) \in [0, H]$*

$$b_k^{pv}(\hat{p}_{k-1}(\cdot \mid s, a), V_2, V_1) \leq \frac{2B_v H + 5J + B_p}{\sqrt{n_{k-1}(s,a) \vee 1}},$$

*Proof.* We bound $b_k^{pv}(\hat{p}_{k-1}(\cdot \mid s, a), V_2, V_1)$ as follows:

$$b_k^{pv}(\hat{p}_{k-1}(\cdot \mid s, a), V_2, V_1) = \phi(\hat{p}_{k-1}(\cdot \mid s, a), V_2) + \frac{B_p + 4J}{n_{k-1}(s,a) \vee 1} + \frac{B_v \|V_2 - V_1\|_{2,\hat{p}}}{\sqrt{n_{k-1}(s,a) \vee 1}}$$

$$\overset{(1)}{\leq} \frac{g(p, V)}{\sqrt{n_{k-1}(s,a) \vee 1}} + \frac{j(p, V)}{n_{k-1}(s,a) \vee 1} + \frac{B_p + 4J}{n_{k-1}(s,a) \vee 1} + \frac{B_v H}{\sqrt{n_{k-1}(s,a) \vee 1}}$$

$$\overset{(2)}{\leq} \frac{B_v H}{\sqrt{n_{k-1}(s,a) \vee 1}} + \frac{J}{n_{k-1}(s,a) \vee 1} + \frac{B_p + 4J}{n_{k-1}(s,a) \vee 1} + \frac{B_v H}{\sqrt{n_{k-1}(s,a) \vee 1}}$$

$$\overset{(3)}{\leq} \frac{2B_v H + 5J + B_p}{\sqrt{n_{k-1}(s,a) \vee 1}}$$

In (1), we substituted $\phi$ and bounded $\|V_2 - V_1\|_{2,\hat{p}} \leq H$. (2) is by Lemma 15 and Definition 1, and (3) is by noting that $n \geq \sqrt{n}$ for $n \geq 1$. $\square$

We end this section by stating that Bernstein's inequality induces an admissible $\phi$. The proof can be found in [Zanette and Brunskill, 2019], Proposition 2.

**Lemma 17.** *Bernstein inequality induces an admissible confidence interval with* $g(p,V) = \sqrt{2\mathrm{Var}_{s'\sim p(\cdot|s,a)}V\ln\frac{2SAT}{\delta'}}$ *and* $j(p,V) = 2H\ln\frac{2SAT}{\delta'}$, *or explicitly:*

$$\phi(p(\cdot\mid s,a),V) = \sqrt{\frac{2\mathrm{Var}_{s'\sim p(\cdot|s,a)}V\ln\frac{2SAT}{\delta'}}{n_{k-1}(s,a)\vee 1}} + \frac{2H\ln\frac{2SAT}{\delta'}}{3n_{k-1}(s,a)\vee 1}$$

*with the constants* $J = \frac{2H\ln\frac{2SAT}{\delta'}}{3} = \tilde{\mathcal{O}}(H), B_v = \sqrt{2\ln\frac{2SAT}{\delta'}} = \tilde{\mathcal{O}}(1)$ *and* $B_p = H\sqrt{2\ln\frac{2SAT}{\delta'}} = \tilde{\mathcal{O}}(H)$. *Using lemma 16, it also implies that for any* $V_1, V_2$ *such that for all* $s$, $V_1(s), V_2(s) \in [0,H]$, *it holds that* $b_k^{pv}(\hat{p}_{k-1}(\cdot\mid s,a),V_2,V_1) \lesssim \tilde{\mathcal{O}}(H)$.

## E.2 Failure Events

### E.2.1 Failure Events of EULER

We start by recalling the failure events as stated in [Zanette and Brunskill, 2019], Appendix D. These events are high probability bounds that are based on the Empirical Bernstein Inequality [Maurer and Pontil, 2009] and leads to the bonus terms of the algorithm. Importantly, these events depend on the state-action visitation counter, and, thus, are indifferent to the greedy exploration scheme which we consider.

Define the following failure events.

$$F^r = \left\{\exists s,a,k:\ |r(s,a)-\hat{r}_{k-1}(s,a)|\geq\sqrt{\frac{2\hat{\mathrm{Var}}_{k-1}R(s,a)\ln\frac{4SAT}{\delta'}}{n_{k-1}(s,a)\vee 1}}+\frac{14\ln\frac{4SAT}{\delta'}}{3(n_{k-1}(s,a)\vee 1)}\right\}$$

$$F^{vr} = \left\{\exists s,a,k:\ \left|\sqrt{\hat{\mathrm{Var}}_{k-1}R(s,a)}-\sqrt{\mathrm{Var}\,R(s,a)}\right|\geq\sqrt{\frac{4\ln\frac{2SAT}{\delta'}}{n_{k-1}(s,a)\vee 1}}\right\}$$

$$F^{pv} = \left\{\exists s,a,t,k:\ \left|(\hat{p}_{k-1}(\cdot\mid s,a)-p(\cdot\mid s,a))^T V_{t+1}^*\right|\geq\sqrt{\frac{2\mathrm{Var}_{s'\sim p(\cdot|s,a)}V_{t+1}^*\ln\frac{4SAT}{\delta'}}{n_{k-1}(s,a)\vee 1}}+\frac{2H\ln\frac{2SAT}{\delta'}}{3(n_{k-1}(s,a)\vee 1)}\right\}$$

$$F^{pv2} = \left\{\exists s,a,t,k:\ |\|V_t^*\|_{2,\hat{p}}-\|V_t^*\|_{2,p}|\geq H\sqrt{\frac{4\ln\frac{2SAT}{\delta'}}{n_{k-1}(s,a)\vee 1}}\right\}$$

$$F^{ps} = \left\{\exists s,s',a,k:\ |\hat{p}_{k-1}(s'\mid s,a)-p_{k-1}(s'\mid s,a)|\geq\sqrt{\frac{p(s'\mid s,a)(1-p(s'\mid s,a))\ln\frac{2TS^2A}{\delta'}}{n_{k-1}(s,a)\vee 1}}+\frac{2\ln\frac{2TS^2A}{\delta'}}{3(n_{k-1}(s,a)\vee 1)}\right\}$$

$$F^{pn1} = \left\{\exists s,a,k:\ \|\hat{p}_{k-1}(\cdot\mid s,a)-p(\cdot\mid s,a)\|_1\geq\sqrt{\frac{4S\ln\frac{3SAT}{\delta'}}{n_{k-1}(s,a)\vee 1}}\right\}$$

$$F_k^N = \left\{\exists s,a,k:n_{k-1}(s,a)\leq\frac{1}{2}\sum_{j<k}w_j(s,a)-H\ln\frac{SAH}{\delta'}\right\}.$$

where $w_j(s,a):=\sum_{t=1}^{H}w_{tj}(s,a)$. In [Zanette and Brunskill, 2019], Appendix D, it is shown these events hold individually with probability at most $\delta'$.

### E.2.2 Failure Events of Decreasing Bounded Processes

In this section, we add another failure events to the total set of failure events. This set of failure event is not present in previous analysis of regret in optimistic RL algorithms (e.g., in Azar et al. 2017, Dann et al. 2017, 2018, Zanette and Brunskill 2019).

We define the following failure events.

$$F^{vDP} = \left\{ \exists K \geq 0 : \sum_{k=1}^{K} \sum_{t=1}^{H} \sum_{s} \bar{V}_t^{k-1}(s) - \mathbb{E}[\bar{V}_t^{k}(s) \mid \mathcal{F}_{k-1}] \geq 9SH^2 \ln \frac{3SH}{\delta'} \right\}$$

$$F^{vsDP} = \left\{ \exists K \geq 0 : \sum_{k=1}^{K} \sum_{t=1}^{H} \sum_{s} (\bar{V}_t^{k-1}(s) - \underline{V}_t^{k-1}(s))^2 - \mathbb{E}[(\bar{V}_t^{k}(s) - \underline{V}_t^{k}(s))^2 \mid \mathcal{F}_{k-1}] \geq 9SH^3 \ln \frac{3SH}{\delta'} \right\}$$

In this section, we prove that both of these failure events occur with low probability $\delta'$.

We start by proving that $\{\bar{V}_t^k(s)\}$ is a decreasing processes, independently to the previously defined failure events. We continue and prove that $\left\{\bar{V}_t^k(s) - \underline{V}_t^k(s)\right\}^2$ starts as a decreasing process and then becomes and increasing process.

**Lemma 18.** *The following claims hold.*

1. *For every $s, t$, $\left\{\bar{V}_t^k(s)\right\}_k$ is a decreasing process and is bounded by $[0, H - (t-1)]$.*

2. *For every $s, t$, $\left\{\underline{V}_t^k(s)\right\}_k$ is an increasing process and is bounded by $[0, H - (t-1)]$.*

3. *For every $s, t$, $\left\{\left(\bar{V}_t^k(s) - \underline{V}_t^k(s)\right)^2\right\}_k$ starts as a decreasing process bounded by $[0, (H - (t-1)^2)]$ and then, possibly, becomes an increasing process.*

*Proof.* We start by proving the first claim. The following holds. By the initialization of the algorithm $\forall s, t, \bar{V}_t^0(s) = H - (t-1)$. By construction of the update rule $\bar{V}_t^k(s)$ can only decrease (see Line 14).

We now prove that for every $s, t, k$, $\left\{\bar{V}_t^k(s)\right\}_k$ is bounded from below by $0$. By assumption $r(s, a) \in [0, 1]$, and thus $\hat{r}_{k-1}(s, a) \geq 0$ a.s. . By induction, this implies $\bar{V}_t^{k-1} \geq 0$. The base case holds by initialization, and the induction step by the fact $\hat{r}_{k-1} \geq 0$ and that the bonus terms are positive.

Proving the second claim is done with similar argument, while using $\hat{r}_{k-1}(s, a) \leq 1$ a.s.. By the update rule (see Line 18), $\left\{\underline{V}_t^k(s)\right\}_k$ is an Increasing Bounded Process in $[0, H - (t-1)]$ (similar definition as in 1 with opposite inequality).

To prove the third claim we combine the two claims. Thus, $\left\{\left(\bar{V}_t^k(s) - \underline{V}_t^k(s)\right)^2\right\}_k$ starts as a decreasing process. Then, if the upper and lower value function crosses one another, the process becomes an increasing process. □

**Remark 2.** *Notice that the upper bound and lower bound of the optimal value crosses one another only inside the failure events defined in Section E.2.1). Yet, the analysis in the following will be indifferent to whether the failure event takes place or not.*

**Lemma 19.** $\Pr\left\{F^{vDP}\right\} \leq \delta'$.

*Proof.* We wish to bound

$$\Pr\{\exists K \geq 0 : \sum_{k=1}^{K} \sum_{t=1}^{H} \sum_{s} \bar{V}_h^{k-1}(s) - \mathbb{E}[\bar{V}_h^{k}(s) \mid \mathcal{F}_{k-1}] \geq 9SH^2 \ln \frac{3SH}{\delta'}\}.$$

According to Lemma 18, for every $s, t$, $\left\{\bar{V}_t^k(s)\right\}_{k \geq 1}$ is a decreasing process. Applying Lemma 11 (Appendix A) which bounds the sum of Decreasing Bounded Processes we conclude the proof. □

**Lemma 20.** $\Pr\{F^{vsDP}\} \le \delta'$.

*Proof.* We wish to bound

$$\Pr\left\{\exists K \ge 0 : \sum_{k=1}^{K}\sum_{t=1}^{H}\sum_{s}(\bar{V}_t^{k-1}(s) - \underline{V}_t^{k-1}(s))^2 - \mathbb{E}[(\bar{V}_t^k(s) - \underline{V}_t^k(s))^2 \mid \mathcal{F}_{k-1}] \ge 9SH^3 \ln\frac{3SH}{\delta'}\right\}.$$

Consider a fixed $s, t$. Furthermore, define the following event

$$\mathbb{A}_{k-1} = \left\{\bar{V}_t^{k-1}(s) > \underline{V}_t^{k-1}(s)\right\}.$$

We have that

$$\sum_{k=1}^{K}(\bar{V}_t^{k-1}(s) - \underline{V}_t^{k-1}(s))^2 - \mathbb{E}[(\bar{V}_t^k(s) - \underline{V}_t^k(s))^2 \mid \mathcal{F}_{k-1}]$$

$$\le \sum_{k=1}^{K}\left((\bar{V}_t^{k-1}(s) - \underline{V}_t^{k-1}(s))^2 - \mathbb{E}[(\bar{V}_t^k(s) - \underline{V}_t^k(s))^2 \mid \mathcal{F}_{k-1}]\right)\mathbb{1}\{\mathbb{A}_{k-1}\}$$

$$= \sum_{k=1}^{K}(\bar{V}_t^{k-1}(s) - \underline{V}_t^{k-1}(s))^2\mathbb{1}\{\mathbb{A}_{k-1}\} - \mathbb{E}[(\bar{V}_t^k(s) - \underline{V}_t^k(s))^2\mathbb{1}\{\mathbb{A}_{k-1}\} \mid \mathcal{F}_{k-1}]$$

$$= \sum_{k=1}^{K}(\bar{V}_t^{k-1}(s) - \underline{V}_t^{k-1}(s))^2\mathbb{1}\{\mathbb{A}_{k-1}\} - \mathbb{E}[(\bar{V}_t^k(s) - \underline{V}_t^k(s))^2\mathbb{1}\{\mathbb{A}_k\} \mid \mathcal{F}_{k-1}]$$

$$- \sum_{k=1}^{K}\mathbb{E}[(\bar{V}_t^k(s) - \underline{V}_t^k(s))^2(\mathbb{1}\{\mathbb{A}_{k-1}\} - \mathbb{1}\{\mathbb{A}_k\}) \mid \mathcal{F}_{k-1}]$$

$$\le \sum_{k=1}^{K}(\bar{V}_t^{k-1}(s) - \underline{V}_t^{k-1}(s))^2\mathbb{1}\{\mathbb{A}_{k-1}\} - \mathbb{E}[(\bar{V}_t^k(s) - \underline{V}_t^k(s))^2\mathbb{1}\{\mathbb{A}_k\} \mid \mathcal{F}_{k-1}].$$

The first relation holds by definition, if the event $\mathbb{A}_{k-1}$ is false then the term is negative, since the process becomes increasing, and only decreases the sum. The second relation holds since $\mathbb{1}\{\mathbb{A}_{k-1}\}$ is $\mathcal{F}_{K-1}$ measureable. The forth relation holds since $(\bar{V}_t^{k-1}(s) - \underline{V}_t^{k-1}(s))^2 \ge 0$ and $(\mathbb{1}\{\mathbb{A}_{k-1}\} - \mathbb{1}\{\mathbb{A}_k\}) \ge 0$. Where the latter holds since $\mathbb{1}\{\mathbb{A}_k\} = 1 \to \mathbb{1}\{\mathbb{A}_{k-1}\} = 1$, i.e.,

$$\bar{V}_t^k(s) > \underline{V}_t^k(s) \; \to \bar{V}_t^{k-1}(s) > \underline{V}_t^{k-1}(s).$$

Differently put, if at the $k^{th}$ episode $\bar{V}_t^k(s) > \underline{V}_t^k(s)$ then it also holds for the $k-1^{th}$ episode, $\bar{V}_t^{k-1}(s) > \underline{V}_t^k(s-1)$, as the process $\{\bar{V}_t^k(s)\}_{k\ge 0}$ is increasing and $\left\{\underline{V}_t^k(s)\right\}_{k\ge 0}$ is decreasing by Lemma 18.

Furthermore, by Lemma 18, $\left\{(\bar{V}_t^k(s) - \underline{V}_t^k(s))^2\mathbb{1}\{\mathbb{A}_k\}\right\}_k$ is a Decreasing Bounded Process in $[0, H^2]$. Initially, it decreases since $\mathbb{1}\{\mathbb{A}_k\}_k = 1$ and $\left\{(\bar{V}_t^k(s) - \underline{V}_t^k(s))^2\right\}$ is initially decreasing. Furthermore, when $\mathbb{1}\{\mathbb{A}_k\} = 0$ it cannot increase. Lastly, $(\bar{V}_t^0(s) - \underline{V}_t^0(s))^2\mathbb{1}\{\mathbb{A}_0\} \le H^2$.

Applying Theorem 3 we get that for a fixed $s, t$, with probability $\frac{\delta'}{SH}$

$$\sum_{k=1}^{K}(\bar{V}_t^{k-1}(s) - \underline{V}_t^k(s))^2 - \mathbb{E}[(\bar{V}_t^k(s) - \underline{V}_t^{k-1}(s))^2 \mid \mathcal{F}_{k-1}] \ge 9SH^3 \ln\frac{3SH}{\delta'}.$$

By applying Lemma 11 (Appendix A), which extends this bound to the sum on $s, t$ we conclude the proof. $\qquad\square$

**Lemma 21.** *(All Failure Events) If $\delta' = \frac{\delta}{9}$, then*

$$F := F^r \bigcup F^{vr} \bigcup F^{pr} \bigcup F^{pv} \bigcup F^{pv2} \bigcup F^{ps} \bigcup F^{pn1} \bigcup F^{vDP} \bigcup F^{vsDP}$$

*holds with probability at most $\delta$. If the event $F$ does not hold we say the algorithm is outside the failure event.*

*Proof.* Applying a union bound on all events, which hold individually with probability at most $\delta'$ yield the result. □

### E.3 EULER with Greedy Policies is Optimistic

Our algorithm modifies the exploration bonus of [Zanette and Brunskill, 2019] by using $\bar{V}_{k-1}, \underline{V}_{k-1}$ instead of $\bar{V}_k, \underline{V}_k$, and uses the following bonus (with some abuse of notation):

$$b_k^{pv}(s,a) = b_k^{pv}\big(\hat{p}_{k-1}(\cdot \mid s,a), \bar{V}_{k-1}, \underline{V}_{k-1}\big) \ .$$

We now show that the modified bonus retains the optimism of the algorithm:

**Lemma 22.** *Outside the failure event of the estimation (see Lemma 21), if the confidence interval is admissible, then the relation*

$$\underline{V}_t^{k-1} \le V_t^* \le \bar{V}_t^{k-1}$$

*holds pointwise for all timesteps $t$ and episodes $k$.*

*Proof.* We follow the proof of [Zanette and Brunskill, 2019], Proposition 4, and prove by induction. We first prove that for all $k$, $V_t^* \le \bar{V}_t^k$.

The claim trivially holds for $k = 0$, due to the initialization of the value. Suppose that the result holds for any state $s$ and timestep $t$ in the $k-1^{\text{th}}$ episode. If

$$\hat{r}_{k-1}(s_t^k, a_t^k) + b_{k-1}^r(s_t^k, a_t^k) + \hat{p}_{k-1}(\cdot \mid s, a_t^k)^T \bar{V}_{t+1}^{k-1} + b_{k-1}^{pv}(s_t^k, a_t^k) \ge \bar{V}_t^{k-1}(s_t) \ ,$$

then by the induction's assumption we are done. Otherwise, denote the optimal action in the real MDP at state $s_t^k$ by $a_t^*$. The value is updated as follows:

$$\begin{aligned}
\bar{V}_k(s_t^k) &= \hat{r}_{k-1}(s_t^k, a_t^k) + b_{k-1}^r(s_t^k, a_t^k) + \hat{p}_{k-1}(\cdot \mid s, a_t^k)^T \bar{V}_{t+1}^{k-1} + b_{k-1}^{pv}(s_t^k, a_t^k) \\
&\ge \hat{r}_{k-1}(s_t^k a_t^*) + b_{k-1}^r(s_t^k, a_t^*) + \hat{p}_{k-1}(\cdot \mid s, a_t^*)^T \bar{V}_{t+1}^{k-1} + b_{k-1}^{pv}(s_t^k, a_t^*) \\
&\ge r(s_t^k, a_t^*) + \hat{p}_{k-1}(\cdot \mid s, a_t^*)^T \bar{V}_{t+1}^{k-1} + b_{k-1}^{pv}(s_t^k, a_t^*)
\end{aligned}$$

The first inequality is since $a_t^k$ is the action that maximizes the greedy value and the second inequality is due to the optimism of the reward when the reward bonus is added, outside the failure events (Lemma 21). Next, using the inductive hypothesis ($V_{t+1}^* \le \bar{V}_{t+1}^{k-1}$ element-wise), we get

$$\bar{V}_t^k(s_t^k) \ge r(s_t^k, a_t^*) + \hat{p}_{k-1}(\cdot \mid s, a_t^*)^T V_{t+1}^* + b_{k-1}^{pv}(s_t^k, a_t^*)$$

We now apply Lemma 15, which implies that

$$b_{k-1}^{pv}(s_t^k, a_t^*) \ge \phi(p(\cdot \mid s_t^k, a_t^*), V^*) \ ,$$

and thus

$$\bar{V}_t^k(s_t^k) \ge r(s_t^k, a_t^*) + \hat{p}_{k-1}(\cdot \mid s, a_t^*)^T V_{t+1}^* + \phi(p(\cdot \mid s_t^k, a_t^*), v^*)$$

Finally, since $\phi$ is admissible, we get the desired result from property (2) of Definition 1:

$$\bar{V}_t^k(s_t^k) \ge r(s_t^k, a_t^*) + p(\cdot \mid s, a_t^*)^T V_{t+1}^* = V_{t+1}^*(s_t^k)$$

The proof for $\underline{V}_t^{k-1} \le V_t^*$ is almost identical, and thus omitted from this paper. □

### E.4   Proof of Theorem 9

*Proof.* Throughout the proof, we assume that we are outside the failure events that were defined in Section E.2, which happens with probability of at least $1 - \delta$ (Lemma 21). Specifically, it implies that the value function is optimistic, namely $V_1^*(s) \leq V_1^k(s)$ (Lemma 22), and we can bound the regret by,

$$\text{Regret}(K) = \sum_{k=1}^{K} V_1^*(s_1^k) - V_1^{\pi_k}(s_1^k) \leq \sum_{k=1}^{K} \bar{V}_1^{k-1}(s_1^k) - V_1^{\pi_k}(s_1^k).$$

Next, by applying Lemma 7 the following bound holds,

$$\leq \underbrace{\sum_{k=1}^{K} \sum_{t=1}^{H} \mathbb{E}[\bar{V}_t^{k-1}(s_t^k) - \bar{V}_t^k(s_t^k) \mid \mathcal{F}_{k-1}]}_{(A)}$$

$$+ \underbrace{\sum_{k=1}^{K} \sum_{t=1}^{H} \mathbb{E}\big[(\tilde{r}_{k-1} - r)(s_t^k, a_t^k) + (\tilde{p}_{k-1} - p)(\cdot \mid s_t^k, a_t^k)^T \bar{V}_{t+1}^{k-1} \mid \mathcal{F}_{k-1}\big]}_{(B)}. \qquad (14)$$

The regret is thus upper bounded by two terms. The first term $(A)$ also appears in the analysis of RTDP (Theorem 4). Specifically, by Lemma 34 (Appendix F), we can express this term as a sum of $SH$ Decreasing Bounded Process in $[0, H]$:

$$(A) = \sum_{s} \sum_{t=1}^{H} \sum_{k=1}^{K} \bar{V}_t^{k-1}(s) - \mathbb{E}[\bar{V}_t^k(s) \mid \mathcal{F}_{k-1}].$$

**Bounding (A).** Outside failure event $F^{vDP}$, this term is bounded by $9SH^2 \ln \frac{3SH}{\delta'}$. Thus,

$$(A) \lesssim \tilde{\mathcal{O}}(SH^2)$$

**Bounding (B).** The term (B) is almost the same term that is bounded in [Zanette and Brunskill, 2019], and its presence is common in recent literature on exploration in RL (e.g., Dann et al. 2017, 2018, Zanette and Brunskill 2019). The only difference between (B) and the term bounded in [Zanette and Brunskill, 2019] is the presence of $\bar{V}^{k-1}$, the value *before* the update, instead of $\bar{V}^k$, the value after applying the update rule. This is since existing algorithms perform planning from the end of an episode and backwards. Thus, when choosing an action at some timestep $t$, these algorithms have access to the updated value of step $t + 1$. In contrast, we avoid the planning stage, and therefore must rely on the previous value $\bar{V}^{k-1}$. We will later see that we can overcome this without affecting the regret.

Next, let $L_k$ be the set of 'good' state-action pairs, which is defined in Definition 2 and analyzed thoroughly in Appendix F.1. We now decompose the sum of $(B)$ to state actions in and outside $L_k$. We also note that except for the $s_t^k, a_t^k$, all of the variables in $(B)$ are $\mathcal{F}_{k-1}$ measurable, which allows us to explicitly write the conditional expectation using $w_{tk}(s, a)$, as follows:

$$(B) = \sum_{k=1}^{K} \sum_{t=1}^{H} w_{tk}(s,a)\Big((\tilde{r}_{k-1} - r)(s,a) + (\tilde{p}_{k-1} - p)(\cdot \mid s,a)^T \bar{V}_{t+1}^{k-1}\Big)$$

$$= \sum_{k=1}^{K} \sum_{t=1}^{H} \sum_{(s,a) \in L_k} w_{tk}(s,a)\Big((\tilde{r}_{k-1} - r)(s,a) + (\tilde{p}_{k-1} - p)(\cdot \mid s,a)^T \bar{V}_{t+1}^{k-1}\Big)$$

$$+ \sum_{k=1}^{K} \sum_{t=1}^{H} \sum_{(s,a) \notin L_k} w_{tk}(s,a)\Big((\tilde{r}_{k-1} - r)(s,a) + (\tilde{p}_{k-1} - p)(\cdot \mid s,a)^T \bar{V}_{t+1}^{k-1}\Big)$$

$$\overset{(1)}{\lesssim} \sum_{k=1}^{K} \sum_{t=1}^{H} \sum_{(s,a) \in L_k} w_{tk}(s,a)\Big((\tilde{r}_{k-1} - r)(s,a) + (\tilde{p}_{k-1} - p)(\cdot \mid s,a)^T \bar{V}_{t+1}^{k-1}\Big)$$

$$+ H \sum_{k=1}^{K} \sum_{t=1}^{H} \sum_{(s,a) \notin L_k} w_{tk}(s,a)$$

$$\overset{(2)}{\lesssim} \sum_{k=1}^{K} \sum_{t=1}^{H} \sum_{(s,a) \in L_k} w_{tk}(s,a)\Big((\tilde{r}_{k-1} - r)(s,a) + (\tilde{p}_{k-1} - p)(\cdot \mid s,a)^T \bar{V}_{t+1}^{k-1}\Big) + \tilde{\mathcal{O}}(SAH^2)$$

For (1), we bound $(\tilde{r}_{k-1} - r)(s,a) \leq \tilde{r}_{k-1}(s,a)$ and $(\tilde{p}_{k-1} - p)(\cdot \mid s,a)^T \bar{V}_{t+1}^{k-1} \leq \tilde{p}_{k-1}(\cdot \mid s,a)^T \bar{V}_{t+1}^{k-1}$. The estimated reward is in $[0,1]$ and it's bonus is at most $\tilde{\mathcal{O}}(1)$, and thus the optimistic reward $\tilde{r}_{k-1}(s,a)$ is $\tilde{\mathcal{O}}(1)$. Due to Lemma 22, the optimistic value $\bar{V}_{t+1}^{k-1} \leq H$, and thus $\hat{p}_{k-1}(\cdot \mid s,a)^T \leq H$. The transition bonus is $b_k^{pv}(s,a) = \tilde{\mathcal{O}}(H)$ due to Lemma 17, which implies that the second term is $\tilde{\mathcal{O}}(H)$. Together, both terms are $\tilde{\mathcal{O}}(H)$. (2) is due to Lemma 36 of Appendix F.1.

As in [Zanette and Brunskill, 2019], we continue the decomposition of the remaining term by adding and subtracting cross-terms that depends on $\hat{p}_{k-1}(\cdot \mid s,a)$

$$\sum_{k=1}^{K} \sum_{t=1}^{H} \sum_{(s,a) \in L_k} w_{tk}(s,a)\big((\tilde{r}_{k-1} - r)(s,a) + (\tilde{p}_{k-1} - p)(\cdot \mid s,a)^T \bar{V}_{t+1}^{k-1}\big)$$

$$= \sum_{k=1}^{K} \sum_{t=1}^{H} \sum_{(s,a) \in L_k} \underbrace{w_{tk}(s,a)(\tilde{r}_{k-1} - r)(s,a)}_{(1)} + \underbrace{w_{tk}(s,a)(\tilde{p}_{k-1} - \hat{p}_{k-1})^T(\cdot \mid s,a)\bar{V}_{t+1}^{k-1}}_{(2)}$$

$$+ \underbrace{w_{tk}(s,a)(\hat{p}_{k-1} - p)(\cdot \mid s,a)^T V_{t+1}^*}_{(3)} + \underbrace{w_{tk}(s,a)(\hat{p}_{k-1} - p)(\cdot \mid s,a)^T (\bar{V}_{t+1}^{k-1} - V_{t+1}^*)}_{(4)}. \quad (15)$$

Recall that we use Bernstein's inequality as the admissible confidence interval. Thus, by Lemma 26, it holds that $J = \frac{2H \ln \frac{2SAT}{\delta'}}{3} = \tilde{\mathcal{O}}(H)$, $B_v = \sqrt{2 \ln \frac{2SAT}{\delta'}} = \tilde{\mathcal{O}}(1)$ and $B_p = H\sqrt{2 \ln \frac{2SAT}{\delta'}} = \tilde{\mathcal{O}}(H)$. Also let $F, D$ be the constants defined in Lemma 23, and specifically

$$F := 2L + LH\sqrt{S} + 6B_v H = \tilde{\mathcal{O}}\big(H\sqrt{S}\big)$$

$$D := 18J + 4B_p + 4L^2 = \tilde{\mathcal{O}}(H)$$

Substituting these constants, terms $(1) - (4)$ are bounded in Lemmas 30, 33, 31 and 32 respectively as follows:

$$(1) \lesssim \sqrt{\mathbb{C}_r^* SAT} + SA$$

$$(2) \lesssim \min\Big\{\sqrt{\mathbb{C}^* SAT} + S\sqrt{S}AH^2 + SAH^{\frac{5}{2}}, \sqrt{\mathbb{C}^\pi SAT} + S\sqrt{S}AH^{\frac{5}{2}}\Big\}$$

$$(3) \lesssim \min\Big\{\sqrt{\mathbb{C}^* SAT} + SAH, \sqrt{\mathbb{C}^\pi SAT} + S\sqrt{S}AH^{\frac{5}{2}}\Big\}$$

$$(4) \lesssim S^2 AH^2 + S\sqrt{S}AH^{\frac{5}{2}}$$

Thus, term $(B)$ of the regret is bounded by:

$$(B) \lesssim \min\left\{\sqrt{\mathbb{C}^* SAT}, \sqrt{\mathbb{C}^\pi SAT}\right\} + \sqrt{\mathbb{C}_r^* SAT} + S^2 AH^2 + S\sqrt{S}AH^{\frac{5}{2}}$$

$$\lesssim \sqrt{\min\{\mathbb{C}^* + \mathbb{C}_r^*, \mathbb{C}^\pi + \mathbb{C}_r^*\}SAT} + S\sqrt{S}AH^2\left(\sqrt{S} + \sqrt{H}\right)$$

Finally, using Lemma 28, we can bound this term by

$$(B) \lesssim \sqrt{\min\left\{\mathbb{Q}^*, \frac{\mathcal{G}^2}{H}\right\}SAT} + S\sqrt{S}AH^2\left(\sqrt{S} + \sqrt{H}\right)$$

and noticing that $(A)$ is negligible compared to $(B)$, we get

$$\text{Regret}(K) \lesssim \sqrt{\min\left\{\mathbb{Q}^*, \frac{\mathcal{G}^2}{H}\right\}SAT} + S\sqrt{S}AH^2\left(\sqrt{S} + \sqrt{H}\right)$$

To derive the problem independent bound, we use the fact that the maximal reward in a trajectory is bounded by $\mathcal{G} \leq H$, which yields

$$\text{Regret}(K) \lesssim \sqrt{HSAT} + S\sqrt{S}AH^2\left(\sqrt{S} + \sqrt{H}\right)$$

$\square$

### E.5 Cumulative Squared Value Difference

In this section, we aim to bound the expected cumulative squared value difference. Specifically, we are interested in a bound for the following quantities:

$$\sum_{k=1}^{K}\sum_{t=1}^{H}\sum_{s,a} w_{tk}(s,a)p(\cdot \mid s,a)^T\left(\bar{V}_{t+1}^{k-1} - \underline{V}_{t+1}^{k-1}\right)^2. \tag{16}$$

$$\sum_{k=1}^{K}\sum_{t=1}^{H}\sum_{s,a} w_{tk}(s,a)p(\cdot \mid s,a)^T\left(\bar{V}_{t+1}^{k-1} - V_{t+1}^{\pi_k}\right)^2 \tag{17}$$

The first quantity allows us to replace Lemma 12 [Zanette and Brunskill, 2019], and the second allows us to prove Lemma 14 of the same paper. Together, they enable us to use the same analysis of [Zanette and Brunskill, 2019]. The final results are stated in Lemmas 26 and 27 by the end of this section. Most of the section will focus on bounding (16), which requires a much more delicate analysis than the bound of [Zanette and Brunskill, 2019].

In order to bound (16), we start by bounding $\left(\bar{V}_{t+1}^{k-1} - \underline{V}_{t+1}^{k-1}\right)^2$ in the following lemma, which corresponds to Proposition 5 of [Zanette and Brunskill, 2019]:

**Lemma 23.** *Outside the failure event, the following holds:*

$$\bar{V}_t^k(s_t^k) - \underline{V}_t^k(s_t^k) \leq \mathbb{E}[\bar{V}_{t+1}^{k-1}(s_{t+1}^k) - \underline{V}_{t+1}^{k-1}(s_{t+1}^k) \mid \mathcal{F}_{k-1}, s_t^k] + \min\left\{\frac{F+D}{\sqrt{n_{k-1}(s_t^k, a_t^k) \vee 1}}, H\right\},$$

*where $F := 2L + LH\sqrt{S} + 6B_v H$, $D := 18J + 4B_p + 4L^2$, the constants $J, B_v, B_p$ are defined in Definition 1 and $L := 2\sqrt{\ln\frac{4SAT}{\delta'}}$.*

*Proof.* The proof is similar to [Zanette and Brunskill, 2019] Proposition 5, which is presented here with the needed adaptation.

If the state $s_t^k$ is encountered in the $k^{th}$ episode at the $t^{th}$ time-step, then $\bar{V}_t^k(s_t^k), \underline{V}_t^k(s_t^k)$ will be updated according to the update rule. Thus,

$$\bar{V}_t^k(s_t^k) \leq \hat{r}_{k-1}(s_t^k, a_t^k) + b_{k-1}^r(s_t^k, a_t^k) + \hat{p}_{k-1}(\cdot \mid s_t^k, a_t^k)^T \bar{V}_{t+1}^{k-1} + b_k^{pv}(\hat{p}_{k-1}(\cdot \mid s_t^k, a_t^k), \bar{V}_{t+1}^{k-1}, \underline{V}_{t+1}^{k-1})$$

$$\underline{V}_t^k(s_t^k) \geq \hat{r}_{k-1}(s_t^k, a_t^k) - b_{k-1}^r(s_t^k, a_t^k) + \hat{p}_{k-1}(\cdot \mid s_t^k, a_t^k)^T \underline{V}_{t+1}^{k-1} - b_k^{pv}(\hat{p}_{k-1}(\cdot \mid s_t^k, a_t^k), \underline{V}_{t+1}^{k-1}, \bar{V}_{t+1}^{k-1}).$$

Subtraction yields:

$$\bar{V}_t^k(s_t^k) - \underline{V}_t^k(s_t^k) \leq 2b_{k-1}^r(s_t^k, a_t^k) + \hat{p}_{k-1}(\cdot \mid s_t^k, a_t^k)^T (\bar{V}_{t+1}^{k-1} - \underline{V}_{t+1}^{k-1})$$
$$+ b_k^{pv}(\hat{p}_{k-1}(\cdot \mid s_t^k, a_t^k), \bar{V}_{t+1}^{k-1}, \underline{V}_{t+1}^{k-1}) + b_k^{pv}(\hat{p}_{k-1}(\cdot \mid s_t^k, a_t^k), \underline{V}_{t+1}^{k-1}, \bar{V}_{t+1}^{k-1}).$$

Next, we substitute the definition of the confidence bonus, which yields

$$\bar{V}_t^k(s_t^k) - \underline{V}_t^k(s_t^k) \leq 2b_{k-1}^r(s_t^k, a_t^k) + \hat{p}_{k-1}(\cdot \mid s_t^k, a)^T (\bar{V}_{t+1}^{k-1} - \underline{V}_{t+1}^{k-1})$$
$$+ \phi(\hat{p}_{k-1}(\cdot \mid s_t^k, a_t^k), \bar{V}_{t+1}^{k-1}) + \frac{4J + B_p}{n_{k-1}(s_t^k, a_t^k) \vee 1} + \frac{B_v \|\bar{V}_{t+1}^{k-1} - \underline{V}_{t+1}^{k-1}\|_{2, \hat{p}}}{\sqrt{n_{k-1}(s_t^k, a_t^k) \vee 1}}$$
$$+ \phi(\hat{p}_{k-1}(\cdot \mid s_t^k, a_t^k), \underline{V}_{t+1}^{k-1}) + \frac{4J + B_p}{n_{k-1}(s_t^k, a_t^k) \vee 1} + \frac{B_v \|\bar{V}_{t+1}^{k-1} - \underline{V}_{t+1}^{k-1}\|_{2, \hat{p}}}{\sqrt{n_{k-1}(s_t^k, a_t^k) \vee 1}}.$$

Using Lemma 15, property (2), and Inequalities (13), we get,

$$\bar{V}_t^k(s_t^k) - \underline{V}_t^k(s_t^k) \leq 2b_{k-1}^r(s_t^k, a_t^k) + \hat{p}_{k-1}(\cdot \mid s_t^k, a)^T (\bar{V}_{t+1}^{k-1} - \underline{V}_{t+1}^{k-1})$$
$$+ 2\phi(p(\cdot \mid s_t^k, a_t^k), \underline{V}_{t+1}^*) + 4\left( \frac{4J + B_p}{n_{k-1}(s_t^k, a_t^k) \vee 1} + \frac{B_v \|\bar{V}_{t+1}^{k-1} - \underline{V}_{t+1}^{k-1}\|_{2, \hat{p}}}{\sqrt{n_{k-1}(s_t^k, a_t^k) \vee 1}} \right)$$
$$= 2b_{k-1}^r(s_t^k, a_t^k) + p(\cdot \mid s_t^k, a)^T (\bar{V}_{t+1}^{k-1} - \underline{V}_{t+1}^{k-1})$$
$$+ (\hat{p}_{k-1}(\cdot \mid s_t^k, a_t^k) - p(\cdot \mid s_t^k, a_t^k))^T (\bar{V}_{t+1}^{k-1} - \underline{V}_{t+1}^{k-1})$$
$$+ 2\frac{g(p(\cdot \mid s_t^k, a_t^k), V_{t+1}^*)}{\sqrt{n_{k-1}(s_t^k, a_t^k) \vee 1}} + 2\frac{J}{n_{k-1}(s_t^k, a_t^k) \vee 1}$$
$$+ 4\left( \frac{4J + B_p}{n_{k-1}(s_t^k, a_t^k) \vee 1} + \frac{B_v \|\bar{V}_{t+1}^{k-1} - \underline{V}_{t+1}^{k-1}\|_{2, \hat{p}}}{\sqrt{n_{k-1}(s_t^k, a_t^k) \vee 1}} \right),$$

where in the last relation we substituted $\phi$ and added and subtracted $p(\cdot \mid s_t^k, a)^T (\bar{V}_{t+1}^{k-1} - \underline{V}_{t+1}^{k-1})$.

By Lemma 18, we know that $\bar{V}_{t+1}^{k-1}, \underline{V}_{t+1}^{k-1} \in [0, H]$. Thus, $\left\| \bar{V}_{t+1}^{k-1} - \underline{V}_{t+1}^{k-1} \right\| \leq H$, which also implies that $\|\bar{V}_{t+1}^{k-1} - \underline{V}_{t+1}^{k-1}\|_{2, \hat{p}} \leq H$. In addition, using Hölder's inequality, and outside failure event $F^{pn1}$, we can bound

$$(\hat{p}_{k-1}(\cdot \mid s_t^k, a_t^k) - p(\cdot \mid s_t^k, a_t^k))^T (\bar{V}_{t+1}^{k-1} - \underline{V}_{t+1}^{k-1})$$
$$\leq \left\| \hat{p}_{k-1}(\cdot \mid s_t^k, a_t^k) - p(\cdot \mid s_t^k, a_t^k) \right\|_1 \left\| \bar{V}_{t+1}^{k-1} - \underline{V}_{t+1}^{k-1} \right\|_\infty$$
$$\leq H \sqrt{\frac{4S \ln \frac{2SAT}{\delta'}}{n_{k-1}(s, a) \vee 1}} = LH \sqrt{\frac{S}{n_{k-1}(s_t^k, a_t^k) \vee 1}}$$

Substituting both of these bounds, we get

$$\bar{V}_t^k(s_t^k) - \underline{V}_t^k(s_t^k) \leq 2b_{k-1}^r(s_t^k, a_t^k) + p(\cdot \mid s_t^k, a)^T(\bar{V}_{t+1}^{k-1} - \underline{V}_{t+1}^{k-1}) + LH\sqrt{\frac{S}{n_{k-1}(s_t^k, a_t^k) \vee 1}}$$

$$+ 2\frac{g(p(\cdot \mid s_t^k, a_t^k), V_{t+1}^*)}{\sqrt{n_{k-1}(s_t^k, a_t^k) \vee 1}} + 2\frac{J}{n_{k-1}(s_t^k, a_t^k) \vee 1}$$

$$+ 4\left(\frac{4J + B_p}{n_{k-1}(s_t^k, a_t^k) \vee 1} + \frac{B_v H}{\sqrt{n_{k-1}(s_t^k, a_t^k) \vee 1}}\right) \tag{18}$$

We now bound the remaining terms. First, using Lemma 15, property (1), we can bound $g(p, V_{t+1}^*) \leq B_v H$. Second, notice that

$$p(\cdot \mid s_t^k, a))^T(\bar{V}_{t+1}^{k-1} - \underline{V}_{t+1}^{k-1}) = \sum_{s_{t+1}^k} p(s_{t+1}^k \mid s_t^k, a))^T(\bar{V}_{t+1}^{k-1}(s_{t+1}^k) - \underline{V}_{t+1}^{k-1}(s_{t+1}^k))$$

$$= \mathbb{E}\left[\bar{V}_{t+1}^{k-1}(s_{t+1}^k) - \underline{V}_{t+1}^{k-1}(s_{t+1}^k) \mid \mathcal{F}_{k-1}, s_t^k\right].$$

Finally, outside failure event $F^r$, the reward bonus can be bounded by

$$b_k^r(s_t^k, a_t^k) = \sqrt{\frac{2\hat{\mathrm{Var}}(R(s_t^k, a_t^k)) \ln \frac{4SAT}{\delta'}}{n_{k-1}(s_t^k, a_t^k) \vee 1}} + \frac{14 \ln \frac{4SAT}{\delta'}}{3n_{k-1}(s_t^k, a_t^k) \vee 1}$$

$$\leq \frac{L}{\sqrt{n_{k-1}(s_t^k, a_t^k) \vee 1}} + \frac{2L^2}{n_{k-1}(s_t^k, a_t^k) \vee 1}$$

where we used the fact that for variables in $[0, 1]$, $\hat{\mathrm{Var}}(R(s_t^k, a_t^k)) \leq 1$.

Putting it all together in (18), we get

$$\bar{V}_t^k(s_t^k) - \underline{V}_t^k(s_t^k) \leq \mathbb{E}\left[\bar{V}_{t+1}^{k-1}(s_{t+1}^k) - \underline{V}_{t+1}^{k-1}(s_{t+1}^k) \mid \mathcal{F}_{k-1}, s_t^k\right]$$

$$+ \frac{2L + LH\sqrt{S} + 6B_v H}{\sqrt{n_{k-1}(s_t^k, a_t^k) \vee 1}} + \frac{18J + 4B_p + 4L^2}{n_{k-1}(s_t^k, a_t^k) \vee 1}$$

$$\leq \mathbb{E}\left[\bar{V}_{t+1}^{k-1}(s_{t+1}^k) - \underline{V}_{t+1}^{k-1}(s_{t+1}^k) \mid \mathcal{F}_{k-1}, s_t^k\right] + \frac{F + D}{n_{k-1}(s_t^k, a_t^k) \vee 1}$$

where in the last relation we substituted $F$ and $D$ and used $\sqrt{n} \leq n$ for $n \geq 1$.

To finalize the proof note that outside the failure event, $\underline{V}_t^k(s) \leq \bar{V}_t^k(s)$, and the first term is therefore positive. combined with $\bar{V}_t^k(s_t^k) - \underline{V}_t^k(s_t^k) \leq H$ yields

$$\bar{V}_t^k(s_t^k) - \underline{V}_t^k(s_t^k) \leq \min\left\{\mathbb{E}\left[\bar{V}_{t+1}^{k-1}(s_{t+1}^k) - \underline{V}_{t+1}^{k-1}(s_{t+1}^k) \mid \mathcal{F}_{k-1}, s_t^k\right] + \frac{F + D}{\sqrt{n_{k-1}(s_t^k, a_t^k) \vee 1}}, H\right\}$$

$$\leq \mathbb{E}\left[\bar{V}_{t+1}^{k-1}(s_{t+1}^k) - \underline{V}_{t+1}^{k-1}(s_{t+1}^k) \mid \mathcal{F}_{k-1}, s_t^k\right] + \min\left\{\frac{F + D}{\sqrt{n_{k-1}(s_t^k, a_t^k) \vee 1}}, H\right\}.$$

$\square$

**Remark 3.** *See that the first term in Equation Lemma 23 does not appear in the analysis of [Zanette and Brunskill, 2019]. Its existence is a direct consequence of the fact we use 1-step greedy policies, and not solving the approximate model at the beginning of each episode. Remarkably, we will later see that this term is comparable to the other previously existing terms.*

We now move to bounding the expected squared value difference, as formally stated in as follows:

**Lemma 24.** *Let* $\Delta_t^k := \left(\bar{V}_t^{k-1}(s_t^k) - \underline{V}_t^{k-1}(s_t^k)\right) - \left(\bar{V}_t^k(s_t^k) - \underline{V}_t^k(s_t^k)\right)$. *Then, outside the failure event,*

$$\mathbb{E}\left[\left(\bar{V}_t^{k-1}(s_t^k) - \underline{V}_t^{k-1}(s_t^k)\right)^2 \mid \mathcal{F}_{k-1}\right]$$

$$\leq 2H \sum_{t'=t}^{H-1} \mathbb{E}\left[\Delta_{t'}^k(s_{t'}^k)^2 + \min\left\{\frac{(F+D)^2}{n_{k-1}(s_{t'}^k, a_{t'}^k) \vee 1}, H^2\right\} \mid \mathcal{F}_{k-1}\right],$$

*where* $F + D$ *is defined in Lemma 23.*

*Proof.* Before proving the bound, we express the bound of Lemma 23 in terms of $\Delta_t^k$. For brevity, we denote $Y_k(s, a) := \min\left\{\frac{F+D}{\sqrt{n_k(s,a)\vee 1}}, H\right\}$, which is $\mathcal{F}_k$ measurable.

Assume the state $s_t^k$ is visited in the $k^{th}$ episode at the $t^{th}$ time-step. Then, by Lemma 23,

$$\bar{V}_t^{k-1}(s_t^k) - \underline{V}_6^{k-1}(s_t^k) = \Delta_t^k + \bar{V}_t^k(s_t^k) - \underline{V}_t^k(s_t^k)$$

$$\leq \Delta_t^k + Y_{k-1}(s_t^k, a_t^k) + \underbrace{\mathbb{E}\left[\bar{V}_{t+1}^{k-1}(s_{t+1}^k) - \underline{V}_{t+1}^{k-1}(s_{t+1}^k) \mid \mathcal{F}_{k-1}, s_t^k\right]}_{(*)}. \qquad (19)$$

Next, by substituting Equation (19) in (*), we get

$$(*) \leq \mathbb{E}\left[\Delta_{t+1}^k + Y_{k-1}(s_{t+1}^k, a_{t+1}^k) + \mathbb{E}\left[\bar{V}_{t+2}^{k-1}(s_{t+2}^k) - \underline{V}_{t+2}^{k-1}(s_{t+2}^k) \mid \mathcal{F}_{k-1}, s_{t+1}^k\right] \mid \mathcal{F}_{k-1}, s_t^k\right]$$

$$= \mathbb{E}\left[\Delta_{t+1}^k + Y_{k-1}(s_{t+1}^k, a_{t+1}^k) + \bar{V}_{t+2}^{k-1}(s_{t+2}^k) - \underline{V}_{t+2}^{k-1}(s_{t+2}^k) \mid \mathcal{F}_{k-1}, s_t^k\right],$$

where the last relation holds by the tower property.

Iterating using this technique until $t = H$, and using $\bar{V}_{H+1} = \underline{V}_{H+1} = 0$, we conclude the following bound:

$$\bar{V}_t^{k-1}(s_t^k) - \underline{V}_t^{k-1}(s_t^k) \leq \sum_{t'=t}^{H} \mathbb{E}[\Delta_{t'}^k(s_{t'}^k) + Y_{k-1}(s_{t'}^k, a_{t'}^k) \mid \mathcal{F}_{k-1}, s_t^k],$$

With this bound at hand, we can derive the desired result as follows:

$$\left(\bar{V}_t^{k-1}(s_t^k) - \underline{V}_t^{k-1}(s_t^k)\right)^2 \leq \left(\sum_{t'=t}^{H} \mathbb{E}\left[\Delta_{t'}^k(s_{t'}^k) + Y_{k-1}(s_{t'}^k, a_{t'}^k) \mid \mathcal{F}_{k-1}, s_t^k\right]\right)^2$$

$$\stackrel{(CS)}{\leq} (H - t + 1) \sum_{t'=t}^{H} \mathbb{E}\left[\Delta_{t'}^k(s_{t'}^k) + Y_{k-1}(s_{t'}^k, a_{t'}^k) \mid \mathcal{F}_{k-1}, s_t^k\right]^2$$

$$\stackrel{(J)}{\leq} (H - t + 1) \sum_{t'=t}^{H} \mathbb{E}\left[\left(\Delta_{t'}^k(s_{t'}^k) + Y_{k-1}(s_{t'}^k, a_{t'}^k)\right)^2 \mid \mathcal{F}_{k-1}, s_t^k\right]$$

$$\stackrel{(CS)}{\leq} 2(H - t + 1) \sum_{t'=t}^{H} \mathbb{E}\left[\Delta_{t'}^k(s_{t'}^k)^2 + Y_{k-1}^2(s_{t'}^k, a_{t'}^k) \mid \mathcal{F}_{k-1}, s_t^k\right]$$

$$\leq 2H \sum_{t'=t}^{H} \mathbb{E}\left[\Delta_{t'}^k(s_{t'}^k)^2 + Y_{k-1}^2(s_{t'}^k, a_{t'}^k) \mid \mathcal{F}_{k-1}, s_t^k\right]$$

$(CS)$ denotes Cauchy-Schwarz inequality, and specifically $\left(\sum_{i=1}^n a_i\right)^2 \leq n \sum_{i=1}^n a_i^2$. $(J)$ is Jensen's inequality. Taking the conditional expectation $\mathbb{E}[\cdot \mid \mathcal{F}_{k-1}]$, using the tower property and substituting $Y_k(s, a)$ gives the desired result. $\qquad \square$

After bounding the expected squared value difference in a single state, we now move to bounding its sum over different time-steps and episode. The main difficulty is in bounding the sum over the first term, which we bound in the following lemma:

**Lemma 25.** *Outside the failure event,*

$$\sum_{k=1}^{K}\sum_{t=1}^{H}\sum_{t'=t}^{H-1}\mathbb{E}[\Delta_{t'}^{k}(s_{t'}^{k})^{2} \mid \mathcal{F}_{k-1}] \leq \tilde{\mathcal{O}}(SH^{4}),$$

*where $\Delta_{t}^{k}(s_{t}^{k})$ is defined in Lemma 24.*

*Proof.* We have that

$$\sum_{t=1}^{H}\sum_{t'=t}^{H}\mathbb{E}[\Delta_{t'}^{k}(s_{t'}^{k})^{2} \mid \mathcal{F}_{k-1}] = \sum_{t=1}^{H}t\mathbb{E}[\Delta_{t}^{k}(s_{t}^{k})^{2} \mid \mathcal{F}_{k-1}] \leq H\sum_{t=1}^{H}\mathbb{E}[\Delta_{t}^{k}(s_{t}^{k})^{2} \mid \mathcal{F}_{k-1}].$$

Furthermore,

$$
\begin{aligned}
\left(\Delta_{t}^{k}(s_{t}^{k})\right)^{2} &= \left(\left(\bar{V}_{t}^{k-1}(s_{t}^{k}) - \underline{V}_{t}^{k-1}(s_{t}^{k})\right) - \left(\bar{V}_{t}^{k}(s_{t}^{k}) - \underline{V}_{t}^{k}(s_{t}^{k})\right)\right)^{2}\\
&= \left(\bar{V}_{t}^{k-1}(s_{t}^{k}) - \underline{V}_{t}^{k-1}(s_{t}^{k})\right)^{2} + \left(\bar{V}_{t}^{k}(s_{t}^{k}) - \underline{V}_{t}^{k}(s_{t}^{k})\right)^{2}\\
&\quad - 2\left(\bar{V}_{t}^{k-1}(s_{t}^{k}) - \underline{V}_{t}^{k-1}(s_{t}^{k})\right)\left(\bar{V}_{t}^{k}(s_{t}^{k}) - \underline{V}_{t}^{k}(s_{t}^{k})\right)\\
&\leq \left(\bar{V}_{t}^{k-1}(s_{t}^{k}) - \underline{V}_{t}^{k-1}(s_{t}^{k})\right)^{2} + \left(\bar{V}_{t}^{k}(s_{t}^{k}) - \underline{V}_{t}^{k}(s_{t}^{k})\right)^{2} - 2\left(\bar{V}_{t}^{k}(s_{t}^{k}) - \underline{V}_{t}^{k}(s_{t}^{k})\right)^{2}\\
&= \left(\bar{V}_{t}^{k-1}(s_{t}^{k}) - \underline{V}_{t}^{k-1}(s_{t}^{k})\right)^{2} - \left(\bar{V}_{t}^{k}(s_{t}^{k}) - \underline{V}_{t}^{k}(s_{t}^{k})\right)^{2},
\end{aligned}
$$

where the third relation holds since $\bar{V}^{k}(s), \underline{V}^{k}(s)$ decreases and increases, respectively, by Lemma 18, and since outside of the failure event $\bar{V}^{k}(s) \geq \underline{V}^{k}(s), \forall k$ (Lemma 22). Another implication these properties is that

$$\left(\bar{V}_{t}^{k-1}(s_{t}^{k}) - \underline{V}_{t}^{k-1}(s_{t}^{k})\right)^{2} \geq \left(\bar{V}_{t}^{k}(s_{t}^{k}) - \underline{V}_{t}^{k}(s_{t}^{k})\right)^{2},$$

Thus,

$$
\begin{aligned}
H&\sum_{k=1}^{K}\sum_{t=1}^{H}\mathbb{E}[\Delta_{t}^{k}(s_{t}^{k})^{2} \mid \mathcal{F}_{k-1}]\\
&\leq H\sum_{k=1}^{K}\sum_{t=1}^{H}\mathbb{E}[\left(\bar{V}_{t}^{k-1}(s_{t}^{k}) - \underline{V}_{t}^{k-1}(s_{t}^{k})\right)^{2} - \left(\bar{V}_{t}^{k}(s_{t}^{k}) - \underline{V}_{t}^{k}(s_{t}^{k})\right)^{2} \mid \mathcal{F}_{k-1}] \ . \quad (20)
\end{aligned}
$$

For brevity, we define $\Delta V_t^k(s) = \bar{V}_t^k(s) - \underline{V}_t^k(s)$. Similarly to the technique used in Lemma 34 (Appendix F),

$$\sum_{k=1}^{K}\sum_{t=1}^{H}\mathbb{E}[\left(\bar{V}_t^{k-1}(s_t^k) - \underline{V}_t^{k-1}(s_t^k)\right)^2 - \left(\bar{V}_t^k(s_t^k) - \underline{V}_t^k(s_t^k)\right)^2 \mid \mathcal{F}_{k-1}]$$

$$= \sum_{k=1}^{K}\sum_{t=1}^{H}\mathbb{E}[\Delta V_t^{k-1}(s_t^k)^2 - \Delta V_t^k(s_t^k)^2 \mid \mathcal{F}_{k-1}]$$

$$\overset{(1)}{=} \sum_{k=1}^{K}\sum_{t=1}^{H}\sum_{s}\mathbb{E}[\mathbb{1}\{s_t^k = s\}\Delta V_t^{k-1}(s)^2 - \mathbb{1}\{s_t^k = s\}\Delta V_t^k(s)^2 \mid \mathcal{F}_{k-1}]$$

$$\overset{(2)}{=} \sum_{k=1}^{K}\sum_{t=1}^{H}\sum_{s}\mathbb{E}[\mathbb{1}\{s_t^k = s\}\Delta V_t^{k-1}(s)^2 + \mathbb{1}\{s_t^k \neq s\}\Delta V_t^{k-1}(s)^2 \mid \mathcal{F}_{k-1}]$$

$$- \mathbb{E}[\mathbb{1}\{s_t^k = s\}\Delta V_t^k(s)^2 + \mathbb{1}\{s_t^k \neq s\}\Delta V_t^{k-1}(s)^2 \mid \mathcal{F}_{k-1}]$$

$$\overset{(3)}{=} \sum_{k=1}^{K}\sum_{t=1}^{H}\sum_{s}\Delta V_t^{k-1}(s)^2 - \mathbb{E}[\Delta V_t^k(s)^2 \mid \mathcal{F}_{k-1}]$$

Relation (1) holds by adding and subtracting $\mathbb{1}\{s \neq s_t^k\}\bar{V}_t^{k-1}(s)$ while using the linearity of expectation. (2) holds since for any event $\mathbb{1}\{A\} + \mathbb{1}\{A^c\} = 1$ and since $\Delta V_t^{k-1}$ is $\mathcal{F}_{k-1}$ measurable. (3) holds by the definition of the update rule. If state $s$ is visited in the $k^{th}$ episode at time-step $t$, then both $\bar{V}_t^k(s), \underline{V}_t^k(s)$ are updated. If not, their value remains as in the $k-1$ iteration.

Lastly,

$$\sum_{k=1}^{K}\sum_{t=1}^{H}\sum_{s}\Delta V_t^{k-1}(s)^2 - \mathbb{E}[\Delta V_t^k(s)^2 \mid \mathcal{F}_{k-1}]$$

$$= \sum_{k=1}^{K}\sum_{t=1}^{H}\sum_{s}\left(\bar{V}_t^{k-1}(s) - \underline{V}_t^{k-1}(s)\right)^2 - \mathbb{E}[\left(\bar{V}_t^k(s) - \underline{V}_t^k(s)\right)^2 \mid \mathcal{F}_{k-1}] \leq \tilde{\mathcal{O}}(SH^3),$$

where the inequality holds outside the failure event $F^{vsDP}$, which is defined in Appendix E.2.2. Plugging this into (20) concludes the proof. □

We are now ready to prove the main results of this section and bound (16) and (17):

**Lemma 26.** *Outside the failure event.*

$$\sum_{k=1}^{K}\sum_{t=1}^{H}\sum_{s,a} w_{tk}(s,a)p(\cdot \mid s,a)^T \left(\bar{V}_{t+1}^{k-1} - \underline{V}_{t+1}^{k-1}\right)^2 \leq \tilde{\mathcal{O}}(SAH^2(F+D)^2 + SAH^5).$$

*where $F + D$ is defined in Lemma 23*

*Proof.* Recall that $w_{tk}(s,a) = \Pr(s_t^k \mid s_1^k, \pi_k)$ is the probability when following $\pi^k$ in the true MDP the state-action in the $k^{th}$ episode at the $t^{th}$ time-step is $(s_t^k, a_t^k) = (s,a)$. Thus, the following relation holds.

$$\sum_{s,a} w_{tk}(s,a)p(\cdot \mid s,a)^T (\bar{V}_{t+1}^{k-1} - \underline{V}_{t+1}^{k-1})^2$$

$$= \sum_{s_t} \Pr(s_t^k \mid s_1^k, \pi_k) \sum_{s_{t+1}} p(s_{t+1}^k \mid s_t^k, a_t^k)(\bar{V}_{t+1}^{k-1}(s_{t+1}^k) - \underline{V}_{t+1}^{k-1}(s_{t+1}^k))^2$$

$$= \sum_{s_{t+1}} \Pr(s_{t+1}^k \mid s_1^k, \pi_k)(\bar{V}_{t+1}^{k-1}(s_{t+1}^k) - \underline{V}_{t+1}^{k-1}(s_{t+1}^k))^2$$

$$= \mathbb{E}[(\bar{V}_{t+1}^{k-1}(s_{t+1}) - \underline{V}_{t+1}^{k-1}(s_{t+1}))^2 \mid \mathcal{F}_{k-1}].$$

Since $\bar{V}_{H+1}^{k-1}(s_{t+1}) = \underline{V}_{H+1}^{k-1}(s_{t+1} = 0$, we obtain,

$$\sum_{k=1}^{K}\sum_{t=1}^{H}\sum_{s,a} w_{tk}(s,a)p(\cdot \mid s,a)(\bar{V}_{t+1}^{k-1} - \underline{V}_{t+1}^{k-1})^2$$

$$= \sum_{k=1}^{K}\sum_{t=1}^{H} \mathbb{E}[(\bar{V}_{t+1}^{k-1}(s_t^k) - \underline{V}_{t+1}^{k-1}(s_t^k)^2 \mid \mathcal{F}_{k-1}]$$

$$\leq \sum_{k=1}^{K}\sum_{t=1}^{H} \mathbb{E}[(\bar{V}_{t}^{k-1}(s_t^k) - \underline{V}_{t}^{k-1}(s_t^k)^2 \mid \mathcal{F}_{k-1}].$$

Thus,

$$\sum_{k=1}^{K}\sum_{t=1}^{H}\sum_{s,a} w_{tk}(s,a)p(\cdot \mid s,a)^T \left(\bar{V}_{t+1}^{k-1} - \underline{V}_{t+1}^{k-1}\right)^2$$

$$\leq \sum_{k=1}^{K}\sum_{t=1}^{H} \mathbb{E}[\left(\bar{V}_{t+1}^{k-1}(s_t^k) - \underline{V}_{t+1}^{k-1}(s_t^k)\right)^2 \mid \mathcal{F}_{k-1}]$$

$$\overset{(*)}{\leq} 2H \sum_{k=1}^{K}\sum_{t=1}^{H}\sum_{t'=t}^{H} \mathbb{E}[\Delta_{t'}^k(s_{t'}^k)^2 \mid \mathcal{F}_{k-1}] + 2H \sum_{k=1}^{K}\sum_{t=1}^{H}\sum_{t'=t}^{H} \mathbb{E}\left[\min\left\{\frac{(F+D)^2}{n_{k-1}(s_{t'}^k, a_{t'}^k) \vee 1}, H^2\right\} \mid \mathcal{F}_{k-1}\right]$$

$$= 2H \sum_{k=1}^{K}\sum_{t=1}^{H} t\mathbb{E}[\Delta_t^k(s_t^k)^2 \mid \mathcal{F}_{k-1}] + 2H \sum_{k=1}^{K}\sum_{t=1}^{H} t\mathbb{E}\left[\min\left\{\frac{(F+D)^2}{n_{k-1}(s_t^k, a_t^k) \vee 1}, H^2\right\} \mid \mathcal{F}_{k-1}\right]$$

$$\leq 2H^2 \sum_{k=1}^{K}\sum_{t=1}^{H} \mathbb{E}[\Delta_t^k(s_t^k)^2 \mid \mathcal{F}_{k-1}] + 2H^2 \sum_{k=1}^{K}\sum_{t=1}^{H} \mathbb{E}\left[\min\left\{\frac{(F+D)^2}{n_{k-1}(s_t^k, a_t^k) \vee 1}, H^2\right\} \mid \mathcal{F}_{k-1}\right] \quad (21)$$

where $(*)$ last relation holds by Lemma 24, in which $\Delta_t^k(s_t^k)$ is defined. The first term is bounded in Lemma 25 by $\tilde{\mathcal{O}}(SH^5)$. The second term is bounded outside the failure event Using the 'Good Set' $L_k$, which is defined and analyzed in Appendix F.1. The bound for this term can be found in Lemma 39. Combining both of the results and substituting into (21) yields

$$\sum_{k=1}^{K}\sum_{t=1}^{H}\sum_{s,a} w_{tk}(s,a)p(\cdot \mid s,a)^T \left(\bar{V}_{t+1}^{k-1} - \underline{V}_{t+1}^{k-1}\right)^2 \leq \tilde{\mathcal{O}}(SH^5) + \tilde{\mathcal{O}}(SAH^2(F+D)^2 + SAH^5)$$

$$= \tilde{\mathcal{O}}(SAH^2(F+D)^2 + SAH^5)$$

$\square$

**Lemma 27.** *Outside the failure event.*

$$\sum_{k=1}^{K}\sum_{t=1}^{H}\sum_{s,a} w_{tk}(s,a)p(\cdot\mid s,a)^{T}\left(\bar{V}_{t+1}^{k-1}-V_{t+1}^{\pi_{k}}\right)^{2} \leq \tilde{\mathcal{O}}(SAH^{3}(F+D)^{2}+SAH^{5})$$

*where $F+D$ is defined in Lemma 23*

*Proof.* Similarly to Lemma 26, we have that

$$\sum_{k=1}^{K}\sum_{t=1}^{H}\sum_{s,a} w_{tk}(s,a)p(\cdot\mid s,a)^{T}\left(\bar{V}_{t+1}^{k-1}-V_{t+1}^{\pi_{k}}\right)^{2}$$

$$\leq \sum_{k=1}^{K}\sum_{t=1}^{H}\mathbb{E}\left[\left(\bar{V}_{t+1}^{k-1}(s_{t}^{k})-V_{t+1}^{\pi_{k}}(s_{t}^{k})\right)^{2}\mid \mathcal{F}_{k-1}\right]$$

$$\leq \sum_{k=1}^{K}\sum_{t=1}^{H}\mathbb{E}\left[\left(\bar{V}_{t}^{k-1}(s_{t}^{k})-V_{t}^{\pi_{k}}(s_{t}^{k})\right)^{2}\mid \mathcal{F}_{k-1}\right]. \tag{22}$$

where the last inequality is since $\bar{V}_{H+1}^{k-1}(s_{t+1})=V_{H+1}^{\pi_{k}}(s_{t+1}=0$. Applying Lemma 7, we get,

$$\mathbb{E}[\left(\bar{V}_{t}^{k-1}(s_{t}^{k})-V_{t}^{\pi_{k}}(s_{t}^{k})\right)^{2}\mid \mathcal{F}_{k-1}]$$

$$\overset{(1)}{\leq}\mathbb{E}\left[\left(\sum_{t'=t}^{H}\mathbb{E}\left[\bar{V}^{k-1}(s_{t'}^{k})-\bar{V}^{k}(s_{t'}^{k})+(\tilde{r}_{k-1}-r)(s_{t'}^{k},a_{t'}^{k})+(\tilde{p}_{k-1}-p)(s_{t'}^{k},a_{t'}^{k})\bar{V}_{t+1}^{k-1}\mid \mathcal{F}_{k-1},s_{t}^{k}\right]\right)^{2}\mid \mathcal{F}_{k-1}\right]$$

$$\overset{(2)}{\leq}3H\mathbb{E}\left[\sum_{t'=t}^{H}\mathbb{E}\left[\left(\bar{V}^{k-1}(s_{t'}^{k})-\bar{V}^{k}(s_{t'}^{k})\right)^{2}\mid \mathcal{F}_{k-1},s_{t}^{k}\right]\mid \mathcal{F}_{k-1}\right]$$

$$+3H\mathbb{E}\left[\sum_{t'=t}^{H}\mathbb{E}\left[\left((\tilde{r}_{k-1}-r)(s_{t'}^{k},a_{t'}^{k})\right)^{2}+\left((\tilde{p}_{k-1}-p)(s_{t'}^{k},a_{t'}^{k})\bar{V}_{t+1}^{k-1}\right)^{2}\mid \mathcal{F}_{k-1},s_{t}^{k}\right]\mid \mathcal{F}_{k-1}\right]$$

$$\overset{(3)}{=}3H\sum_{t'=t}^{H}\mathbb{E}\left[\left(\bar{V}^{k-1}(s_{t'}^{k})-\bar{V}^{k}(s_{t'}^{k})\right)^{2}\mid \mathcal{F}_{k-1}\right]$$

$$+3H\sum_{t'=t}^{H}\mathbb{E}\left[\left((\tilde{r}_{k-1}-r)(s_{t'}^{k},a_{t'}^{k})\right)^{2}+\left((\tilde{p}_{k-1}-p)(s_{t'}^{k},a_{t'}^{k})\bar{V}_{t+1}^{k-1}\right)^{2}\mid \mathcal{F}_{k-1}\right].$$

Inequality (1) is by Lemma 12. (2) is due to Jensen's inequality, and using the inequality $\left(\sum_{i=1}^{n}a_{i}\right)^{2}\leq n\sum_{i=1}^{n}a_{i}^{2}$, and (3) is by the tower property.

Plugging this back into (22),

$$(22)\leq 3H\sum_{k=1}^{K}\sum_{t=1}^{H}\sum_{t'=t}^{H}\mathbb{E}\left[\left(\bar{V}^{k-1}(s_{t'}^{k})-\bar{V}^{k}(s_{t'}^{k})\right)^{2}\mid \mathcal{F}_{k-1}\right]$$

$$+3H\sum_{k=1}^{K}\sum_{t=1}^{H}\sum_{t'=t}^{H}\mathbb{E}\left[\left((\tilde{r}_{k-1}-r)(s_{t'}^{k},a_{t'}^{k})\right)^{2}+\left((\tilde{p}_{k-1}-p)(s_{t'}^{k},a_{t'}^{k})\bar{V}_{t+1}^{k-1}\right)^{2}\mid \mathcal{F}_{k-1}\right]$$

$$\leq 3H^{2}\sum_{k=1}^{K}\sum_{t=1}^{H}\underbrace{\mathbb{E}\left[\left(\bar{V}^{k-1}(s_{t}^{k})-\bar{V}^{k}(s_{t}^{k})\right)^{2}\mid \mathcal{F}_{k-1}\right]}_{(*)}$$

$$+3H^{2}\sum_{k=1}^{K}\sum_{t=1}^{H}\mathbb{E}\left[\underbrace{\left((\tilde{r}_{k-1}-r)(s_{t}^{k},a_{t}^{k})\right)^{2}}_{(**)}+\underbrace{\left((\tilde{p}_{k-1}-p)(s_{t}^{k},a_{t}^{k})\bar{V}_{t+1}^{k-1}\right)^{2}}_{(***)}\mid \mathcal{F}_{k-1}\right]. \tag{23}$$

We now bound each term of the above. First, we have that

$$
(*) = \sum_{t=1}^{H}\sum_{t=1}^{H}\mathbb{E}\Big[\big(\bar{V}^{k-1}(s_t^k) - \bar{V}^k(s_t^k)\big)^2 \mid \mathcal{F}_{k-1}\Big]
$$

$$
= \sum_{t=1}^{H}\sum_{t=1}^{H}\mathbb{E}\Big[\big(\bar{V}^{k-1}(s_t^k)\big)^2 + \big(\bar{V}^k(s_t^k)\big)^2 - 2\bar{V}^k(s_t^k)\bar{V}^{k-1}(s_t^k) \mid \mathcal{F}_{k-1}\Big]
$$

$$
\overset{(1)}{\leq} \sum_{t=1}^{H}\sum_{t=1}^{H}\mathbb{E}\Big[\big(\bar{V}^{k-1}(s_t^k)\big)^2 + \big(\bar{V}^k(s_t^k)\big)^2 - 2\big(\bar{V}^k(s_t^k)\big)^2 \mid \mathcal{F}_{k-1}\Big]
$$

$$
= \sum_{t=1}^{H}\sum_{t=1}^{H}\mathbb{E}\Big[\big(\bar{V}^{k-1}(s_t^k)\big)^2 - \big(\bar{V}^k(s_t^k)\big)^2 \mid \mathcal{F}_{k-1}\Big]
$$

$$
\overset{(2)}{=} \sum_{t=1}^{H}\sum_{t=1}^{H}\sum_{s}\big(\bar{V}^{k-1}(s)\big)^2 - \mathbb{E}\Big[\big(\bar{V}^k(s)\big)^2 \mid \mathcal{F}_{k-1}\Big].
$$

Relation (1) holds since $0 \leq \bar{V}^k \leq \bar{V}^{k-1}$ (see Lemma 18). (2) is proven similarly to Lemma 34 (Appendix F), as follows

$$
\sum_{k=1}^{K}\sum_{t=1}^{H}\mathbb{E}\Big[\big(\bar{V}^{k-1}(s_t^k)\big)^2 - \big(\bar{V}^k(s_t^k)\big)^2 \mid \mathcal{F}_{k-1}\Big]
$$

$$
\overset{(1)}{=} \sum_{k=1}^{K}\sum_{t=1}^{H}\sum_{s}\mathbb{E}[\mathbb{1}\{s_t^k = s\}\big(\bar{V}^{k-1}(s)\big)^2 - \mathbb{1}\{s_t^k = s\}\big(\bar{V}^k(s)\big)^2 \mid \mathcal{F}_{k-1}]
$$

$$
\overset{(2)}{=} \sum_{k=1}^{K}\sum_{t=1}^{H}\sum_{s}\mathbb{E}[\mathbb{1}\{s_t^k = s\}\big(\bar{V}^{k-1}(s)\big)^2 + \mathbb{1}\{s_t^k \neq s\}\big(\bar{V}^{k-1}(s)\big)^2 \mid \mathcal{F}_{k-1}]
$$

$$
- \mathbb{E}[\mathbb{1}\{s_t^k = s\}\big(\bar{V}^k(s)\big)^2 + \mathbb{1}\{s_t^k \neq s\}\big(\bar{V}^{k-1}(s)\big)^2 \mid \mathcal{F}_{k-1}]
$$

$$
\overset{(3)}{=} \sum_{k=1}^{K}\sum_{t=1}^{H}\sum_{s}\big(\bar{V}^{k-1}(s)\big)^2 - \mathbb{E}[\big(\bar{V}^k(s)\big)^2 \mid \mathcal{F}_{k-1}]
$$

(1) holds by adding and subtracting $\mathbb{1}\{s \neq s_t^k\}\bar{V}_t^{k-1}(s)$ while using the linearity of expectation. (2) holds since for any event $\mathbb{1}\{A\} + \mathbb{1}\{A^c\} = 1$ and since $\Delta V_t^{k-1}$ is $\mathcal{F}_{k-1}$ measurable. (3) holds by the definition of the update rule. If state $s$ is visited in the $k^{th}$ episode at time-step $t$, then both $\bar{V}_t^k(s), \underline{V}_t^k(s)$ are updated. If not, their value remains as in the $k-1$ iteration.

Next, by Lemma 18 for a fixed $s, t$, $\big\{\bar{V}_t^k(s)\big\}_{k\geq 0}$ is a Decreasing Bounded Process in $[0, H^2]$. Applying Lemma 11 we conclude that

$$
(*) \leq \sum_{k=1}^{K}\sum_{t=1}^{H}\sum_{s}\big(\bar{V}^{k-1}(s)\big)^2 - \mathbb{E}[\big(\bar{V}^k(s)\big)^2 \mid \mathcal{F}_{k-1}] \lesssim \tilde{\mathcal{O}}(H^3 S).
$$

We now turn to bound $(**)$.

$$\sum_{k=1}^{K}\sum_{t=1}^{H}\mathbb{E}\left[\left((\tilde{r}_{k-1}-r)(s_t^k,a_t^k)\right)^2 \mid \mathcal{F}_{k-1}\right]$$

$$\overset{(1)}{\leq} 2\sum_{k=1}^{K}\sum_{t=1}^{H}\mathbb{E}\left[\left((\hat{r}_{k-1}-r)(s_t^k,a_t^k)\right)^2 \mid \mathcal{F}_{k-1}\right] + 2\sum_{k=1}^{K}\sum_{t=1}^{H}\mathbb{E}\left[\left(b_k^r(s_t^k,a_t^k)\right)^2 \mid \mathcal{F}_{k-1}\right]$$

$$\overset{(2)}{\leq} 4\sum_{k=1}^{K}\sum_{t=1}^{H}\mathbb{E}\left[\left(b_k^r(s_t^k,a_t^k)\right)^2 \mid \mathcal{F}_{k-1}\right]$$

$$= 4\sum_{k=1}^{K}\sum_{t=1}^{H}\mathbb{E}\left[\left(\sqrt{\frac{2\hat{\mathrm{Var}}(R(s_t^k,a_t^k))\ln\frac{4SAT}{\delta'}}{n_{k-1}(s_t^k,a_t^k)\vee 1}} + \frac{14\ln\frac{4SAT}{\delta'}}{3n_{k-1}(s_t^k,a_t^k)\vee 1}\right)^2 \mid \mathcal{F}_{k-1}\right]$$

$$\overset{(3)}{\lesssim} \sum_{k=1}^{K}\sum_{t=1}^{H}\mathbb{E}\left[\frac{1}{n_{k-1}(s_t^k,a_t^k)\vee 1} \mid \mathcal{F}_{k-1}\right]$$

$$\overset{(4)}{\lesssim} \tilde{\mathcal{O}}(SAH).$$

In (1), we used the definition of $\tilde{r}_{k-1}$ and the inequality $(a+b)^2 \leq 2a^2 + 2b^2$. (2) is since outside the failure event $F^r$, $(\hat{r}_{k-1}-r)(s,a) \leq b_k^r(s,a)$. (3) uses the fact that $R(s,a) \in [0,1]$, and thus $\hat{\mathrm{Var}}(R(s,a) \leq 1$, and $\sqrt{n} \leq n$ for $n \geq 1$. Finally, (4) is due to Lemma 39.

Lastly, we bound $(***)$.

$$\sum_{k=1}^{K}\sum_{t=1}^{H}\mathbb{E}\left[\left((\tilde{p}_{k-1}-p)(s_t^k,a_t^k)^T\bar{V}_{t+1}^{k-1}\right)^2 \mid \mathcal{F}_{k-1}\right]$$

$$= \sum_{k=1}^{K}\sum_{t=1}^{H}\mathbb{E}\left[\left((\hat{p}_{k-1}-p)(s_t^k,a_t^k)^T\bar{V}_{t+1}^{k-1} + b_k^{pv}(s_t^k,a_t^k)\right)^2 \mid \mathcal{F}_{k-1}\right]$$

$$\overset{(1)}{\leq} 2\sum_{k=1}^{K}\sum_{t=1}^{H}\mathbb{E}\left[\left((\hat{p}_{k-1}-p)(s_t^k,a_t^k)^T\bar{V}_{t+1}^{k-1}\right)^2 + \left(b_k^{pv}(s_t^k,a_t^k)\right)^2 \mid \mathcal{F}_{k-1}\right]$$

$$\overset{(2)}{\leq} 2\sum_{k=1}^{K}\sum_{t=1}^{H}\mathbb{E}\left[\left(\|\hat{p}_{k-1}-p\|_1\|\bar{V}_{t+1}^{k-1}\|_\infty\right)^2 + \left(b_k^{pv}(s_t^k,a_t^k)\right)^2 \mid \mathcal{F}_{k-1}\right]$$

$$\overset{(3)}{\leq} 2\sum_{k=1}^{K}\sum_{t=1}^{H}\mathbb{E}\left[H^2\|\hat{p}_{k-1}-p\|_1^2 + \left(b_k^{pv}(s_t^k,a_t^k)\right)^2 \mid \mathcal{F}_{k-1}\right]$$

$$\overset{(4)}{\lesssim} \sum_{k=1}^{K}\sum_{t=1}^{H}\mathbb{E}\left[\frac{H^2 S}{n_{k-1}(s_t^k,a_t^k)} + \frac{(2B_v H + 5J + B_p)^2}{n_{k-1}(s_t^k,a_t^k)} \mid \mathcal{F}_{k-1}\right].$$

$$\overset{(5)}{\lesssim} \tilde{\mathcal{O}}(SAH(F+D)^2)$$

Similarly to the bound on the reward, (1) uses the inequality $(a+b)^2 \leq 2a^2 + 2b^2$. Inequality (2) is due to Hölder's inequality, and (3) bounds $\|\bar{V}_{t+1}^{k-1}\|_\infty \leq H$, which is due to Lemma 18. Next, (4) bounds the transition error outside to failure event $F^{pn1}$ and $b_k^{pv}$ according to Lemma 16. Finally, (5) is by Lemma 39 and noting that $H^2 S + (2B_v H + 5J + B_p)^2 \lesssim (F+D)^2$.

Substituting all of the results into (23), and remembering the $H^2$ factor in this equation, gives the desired result.

$\square$

## E.6 Bounding Different Terms in the Regret Decomposition

In this section, we bound each of the individual terms of the regret decomposition (Equation 15), relaying on results from [Zanette and Brunskill, 2019], as well as on the new lemmas derived in Section E.5, Lemma 26 and Lemma 27. First, we present the problem dependent constants of [Zanette and Brunskill, 2019] for general admissible confidence intervals, and their relation to problem dependent constants with Bernstein's inequality

**Lemma 28.** *Let $\mathbb{C}^*$ and $\mathbb{C}^\pi$ be upper dependent bounds on the following qualities:*

$$\mathbb{C}^* \geq \frac{1}{T} \sum_{k=1}^{K} \sum_{t=1}^{H} \sum_{s,a} w_{tk}(s,a) g(p, V_{t+1}^*)^2$$

$$\mathbb{C}^\pi \geq \frac{1}{T} \sum_{k=1}^{K} \sum_{t=1}^{H} \sum_{s,a} w_{tk}(s,a) g(p, V_{t+1}^{\pi_k})^2 \ ,$$

*with $g(p,V) = \sqrt{2\mathrm{Var}_{s' \sim p(\cdot|s,a)} V(s') \ln \frac{2SAT}{\delta'}}$, and let*

$$\mathbb{C}_r^* = \frac{1}{T} \left( \sum_{k=1}^{K} \sum_{t=1}^{H} \sum_{(s,a) \in L_k} w_{tk}(s,a) \mathrm{Var} R(s,a) \right) \ ,$$

*where $L_k$ is defined in Definition 2. Finally, let $\mathbb{Q}^* := \max_{s,a,t} \big( \mathrm{Var} R(s,a) + \mathrm{Var}_{s' \sim p(\cdot|s,a)} V_{t+1}^*(s') \big)$. Then,*

$$\mathbb{C}_r^* + \mathbb{C}^* \lesssim \mathbb{Q}^*$$

$$\mathbb{C}^\pi \lesssim \frac{\mathcal{G}^2}{H}$$

$$\mathbb{C}_r^* \leq \frac{\mathcal{G}^2}{H}$$

*Proof.* We follow proposition 6 of [Zanette and Brunskill, 2019], and start by substituting $g(p,V)$ into $\mathbb{C}_r^* + \mathbb{C}^*$

$$\mathbb{C}_r^* + \mathbb{C}^* \lesssim \frac{1}{T} \left( \sum_{k=1}^{K} \sum_{t=1}^{H} \sum_{(s,a) \in L_k} w_{tk}(s,a) \mathrm{Var} R(s,a) \right)$$

$$+ \frac{1}{T} \sum_{k=1}^{K} \sum_{t=1}^{H} \sum_{s,a} w_{tk}(s,a) \mathrm{Var}_{s' \sim p(\cdot|s,a)} V_{t+1}^*(s')$$

$$\leq \frac{1}{T} \left( \sum_{k=1}^{K} \sum_{t=1}^{H} \sum_{(s,a)} w_{tk}(s,a) (\mathrm{Var} R(s,a)) + \mathrm{Var}_{s' \sim p(\cdot|s,a)} V_{t+1}^*(s') \right)$$

$$\leq \frac{1}{T} \left( \sum_{k=1}^{K} \sum_{t=1}^{H} \sum_{(s,a)} w_{tk}(s,a) \max_{s,a,t}\{\mathrm{Var} R(s,a)\} + \mathrm{Var}_{s' \sim p(\cdot|s,a)} V_{t+1}^*(s') \right)$$

$$= \frac{\mathbb{Q}^*}{T} \left( \sum_{k=1}^{K} \sum_{t=1}^{H} \sum_{(s,a)} w_{tk}(s,a) \right)$$

$$= \mathbb{Q}^*$$

where the last equality is since $\sum_{(s,a)} w_{tk}(s,a) = 1$ and $T = HK$.

Next, we bound $\mathbb{C}^\pi$:

$$\mathbb{C}^\pi \lesssim \frac{1}{T}\sum_{k=1}^{K}\sum_{t=1}^{H}\sum_{s,a} w_{tk}(s,a)\mathrm{Var}_{s'\sim p(\cdot|s,a)}V_{t+1}^{\pi_k}(s')$$

$$\overset{(1)}{=} \frac{1}{T}\sum_{k=1}^{K}\mathbb{E}\left[\left(\sum_{t=1}^{H} r(s_t^k,a_t^k) - V_1^{\pi_k}(s_1^k)\right)^2 \mid \mathcal{F}_{k-1}\right]$$

$$\leq \frac{1}{T}\sum_{k=1}^{K}\mathbb{E}\left[\left(\sum_{t=1}^{H} r(s_t^k,a_t^k)\right)^2 \mid \mathcal{F}_{k-1}\right]$$

$$\overset{(2)}{\leq} \frac{1}{T}K\mathcal{G}^2 = \frac{\mathcal{G}^2}{H} \ ,$$

where $(1)$ is due to the Law of Total Variance (LTV), which was used in [Azar et al., 2017], and was stated formally in Lemma 15 of [Zanette and Brunskill, 2019]. In $(2)$, we bound the reward in an episode by $\mathcal{G}$.

Finally, the bound on $\mathbb{C}_r^*$ is proven in Lemma 8 of [Zanette and Brunskill, 2019], which concludes this proof.

$\square$

We also prove the following lemma that helps translating bounds that depend on $\mathbb{C}^*$ to bounds that depends on $\mathbb{C}^\pi$. This lemma is equivalent to lemma 14 of [Zanette and Brunskill, 2019], but the prove requires Lemma 27, that was not proved in their paper. This is since they rely on the inequality $\underline{V}_t^{k-1} \leq V^{\pi_k}$, which does not seem to hold.

**Lemma 29** (Bound Translation Lemma). *Outside the failure event, it holds that*

$$\sum_{k=1}^{K}\sum_{t=1}^{H}\sum_{(s,a)\in L_k} w_{tk}(s,a)\frac{g(p,V_{t+1}^*)}{\sqrt{n_{k-1}(s,a)\vee 1}} - \sum_{k=1}^{K}\sum_{t=1}^{H}\sum_{(s,a)\in L_k} w_{tk}(s,a)\frac{g(p,V_{t+1}^{\pi_k})}{\sqrt{n_{k-1}(s,a)\vee 1}}$$

$$= \tilde{\mathcal{O}}\left(B_v SAH^{\frac{3}{2}}(F+D) + B_v SAH^{\frac{5}{2}}\right)$$

*where $F,D$ are defined in Lemma 23.*

*Proof.* We start as in the original Lemma 14 of [Zanette and Brunskill, 2019]:

$$\sum_{k=1}^{K}\sum_{t=1}^{H}\sum_{(s,a)\in L_k} w_{tk}(s,a)\frac{g(p,V_{t+1}^*)}{\sqrt{n_{k-1}(s,a)\vee 1}} - \sum_{k=1}^{K}\sum_{t=1}^{H}\sum_{(s,a)\in L_k} w_{tk}(s,a)\frac{g(p,V_{t+1}^{\pi_k})}{\sqrt{n_{k-1}(s,a)\vee 1}}$$

$$\overset{(1)}{\leq} B_v \sum_{k=1}^{K}\sum_{t=1}^{H}\sum_{(s,a)\in L_k} w_{tk}(s,a)\frac{\left\|V_{t+1}^* - V_{t+1}^{\pi_k}\right\|_{2,p}}{\sqrt{n_{k-1}(s,a)\vee 1}}$$

$$\overset{(2)}{\leq} B_v \sqrt{\sum_{k=1}^{K}\sum_{t=1}^{H}\sum_{(s,a)\in L_k} \frac{w_{tk}(s,a)}{n_{k-1}(s,a)\vee 1}}\sqrt{\sum_{k=1}^{K}\sum_{t=1}^{H}\sum_{(s,a)\in L_k} w_{tk}(s,a)\left\|V_{t+1}^* - V_{t+1}^{\pi_k}\right\|_{2,p}^2}$$

$$\overset{(3)}{\lesssim} B_v \sqrt{SA}\sqrt{\sum_{k=1}^{K}\sum_{t=1}^{H}\sum_{(s,a)} w_{tk}(s,a)p(\cdot\mid s,a)^T\left(\bar{V}_{t+1}^{k-1} - V_{t+1}^{\pi_k}\right)^2}$$

where in $(1)$ we use property 1 of Definition 1, and $(2)$ is due to Cauchy-Schwarz inequality. In $(3)$ we used Lemma 37. Next, we apply Lemma 27 to bound the remaining term by $\tilde{\mathcal{O}}\left(\sqrt{SAH^3(F+D)^2 + SAH^5}\right)$ and bound $\sqrt{SAH^3(F+D)^2 + SAH^5} \leq \sqrt{SAH^3(F+D)^2} + \sqrt{SAH^5}$, which yields the desired result $\square$

We are now ready to bound each of the terms of the regret. To bound the first term, we cite Lemma 8 of [Zanette and Brunskill, 2019]:

**Lemma 30** (Optimistic Reward Bound). *Outside the failure event, it holds that*

$$\sum_{k=1}^{K}\sum_{t=1}^{H}\sum_{(s,a)\in L_k} w_{tk}(s,a)(\tilde{r}_{k-1}-r)(s_t^k,a_t^k) = \tilde{\mathcal{O}}\Big(\sqrt{\mathbb{C}_r^* SAT} + SA\Big)$$

The next three lemmas correspond to the remaining terms, and follow Lemmas 9,10 and 11 of [Zanette and Brunskill, 2019], with slight modifications:

**Lemma 31** (Empirical Transition Bound). *Outside the failure event, it holds that*

$$\sum_{k=1}^{K}\sum_{t=1}^{H}\sum_{(s,a)\in L_k} w_{tk}(s,a)(\hat{p}_{k-1}-p_{k-1})(\cdot \mid s,a)^T V_{t+1}^* = \tilde{\mathcal{O}}\Big(\sqrt{\mathbb{C}^* SAT} + JSA\Big)$$

*The following bound also holds:*

$$\sum_{k=1}^{K}\sum_{t=1}^{H}\sum_{(s,a)\in L_k} w_{tk}(s,a)(\hat{p}_{k-1}-p_{k-1})(\cdot \mid s,a)^T V_{t+1}^*$$
$$= \tilde{\mathcal{O}}\Big(\sqrt{\mathbb{C}^\pi SAT} + JSA + B_v SAH^{\frac{3}{2}}(F+D) + B_v SAH^{\frac{5}{2}}\Big)$$

*where $F, D$ are defined in Lemma 23.*

*Proof.* Similarly to Lemma 9 of [Zanette and Brunskill, 2019], by the definition of $\phi$ (Definition 1), and outside failure event $F^{pv}$,

$$\sum_{k=1}^{K}\sum_{t=1}^{H}\sum_{(s,a)\in L_k} w_{tk}(s,a)(\hat{p}_{k-1}-p_{k-1})(\cdot \mid s,a)^T V_{t+1}^*$$
$$\leq \sum_{k=1}^{K}\sum_{t=1}^{H}\sum_{(s,a)\in L_k} w_{tk}(s,a)\left(\frac{g(p,V_{t+1}^*)}{\sqrt{n_{k-1}(s,a)\vee 1}} + \frac{J}{n_{k-1}(s,a)\vee 1}\right) \tag{24}$$
$$\overset{(*)}{\leq} \sqrt{\sum_{k=1}^{K}\sum_{t=1}^{H}\sum_{(s,a)\in L_k} w_{tk}(s,a)g(p,V_{t+1}^*)^2}\sqrt{\sum_{k=1}^{K}\sum_{t=1}^{H}\sum_{(s,a)\in L_k} \frac{w_{tk}(s,a)}{n_{k-1}(s,a)\vee 1}}$$
$$+ J\sum_{k=1}^{K}\sum_{t=1}^{H}\sum_{(s,a)\in L_k} \frac{w_{tk}(s,a)}{n_{k-1}(s,a)\vee 1}$$

where the last inequality is by Cauchy-Schwarz Inequality. Substituting the definition of $\mathbb{C}^*$, and using Lemma 37, we get

$$\lesssim \sqrt{T\mathbb{C}^*}\sqrt{SA} + JSA ,$$

which concludes the first statement of the lemma. To get the second statement, we apply Lemma 29 before inequality $(*)$ and only then use Cauchy-Schwarz Inequality. This creates the additional constant term of $\tilde{\mathcal{O}}\Big(B_v SAH^{\frac{3}{2}}(F+D) + B_v SAH^{\frac{5}{2}}\Big)$. Then, by applying Lemma 37, we get the bound with $\mathbb{C}^\pi$. $\qquad\square$

**Lemma 32** (Lower Order Term). *Let $F, D$ be the constants defined in Lemma 23. Outside the failure event, it holds that*

$$\sum_{k=1}^{K}\sum_{t=1}^{H}\sum_{(s,a)\in L_k} w_{tk}(s,a)\big|(\hat{p}_{k-1}-p)(\cdot\mid s,a)^T(\bar{V}_{t+1}^{k-1}-V_{t+1}^*)\big|$$
$$=\tilde{\mathcal{O}}\Big(S^{\frac{3}{2}}AH(F+D+H^{\frac{3}{2}})+S^2AH\Big)$$

*Proof.* Similarly to Lemma 11 of [Zanette and Brunskill, 2019], by the definition of $\phi$ (Definition 1), and outside failure event $F^{ps}$,

$$\sum_{k=1}^{K}\sum_{t=1}^{H}\sum_{(s,a)\in L_k} w_{tk}(s,a)\big|(\hat{p}_{k-1}-p)(\cdot\mid s,a)^T(\bar{V}_{t+1}^{k-1}-V_{t+1}^*)\big|$$

$$\lesssim \sum_{k=1}^{K}\sum_{t=1}^{H}\sum_{(s,a)\in L_k} w_{tk}(s,a)\sum_{s'}\sqrt{\frac{p(s'\mid s,a)(1-p(s'\mid s,a))}{n_{k-1}(s,a)\vee 1}}\big|\bar{V}_{t+1}^{k-1}(s')-V_{t+1}^*(s')\big|$$

$$+\sum_{k=1}^{K}\sum_{t=1}^{H}\sum_{(s,a)\in L_k} w_{tk}(s,a)\sum_{s'}\frac{\big|\bar{V}_{t+1}^{k-1}(s')-V_{t+1}^*(s')\big|}{n_{k-1}(s,a)\vee 1}$$

$$\leq \sum_{k=1}^{K}\sum_{t=1}^{H}\sum_{(s,a)\in L_k} w_{tk}(s,a)\sum_{s'}\sqrt{\frac{p(s'\mid s,a)(1-p(s'\mid s,a))}{n_{k-1}(s,a)\vee 1}}\big|\bar{V}_{t+1}^{k-1}(s')-V_{t+1}^*(s')\big|$$

$$+\sum_{k=1}^{K}\sum_{t=1}^{H}\sum_{(s,a)\in L_k} w_{tk}(s,a)\frac{HS}{n_{k-1}(s,a)\vee 1} \ ,$$

where in the last inequality we used the fact that $V_{t+1}^*$ and $\bar{V}_{t+1}^{k-1}$ are in $[0,H]$, by Lemma 18. Next, using the optimism of the value $\underline{V}_{t+1}^{k-1}\leq V_{t+1}^*\leq\bar{V}_{t+1}^{k-1}$ (Lemma 22), and since $(1-p)\leq 1$ for $p\in[0,1]$, we can bound

$$\leq \sum_{k=1}^{K}\sum_{t=1}^{H}\sum_{(s,a)\in L_k} w_{tk}(s,a)\sum_{s'}\sqrt{\frac{p(s'\mid s,a)}{n_{k-1}(s,a)\vee 1}}\big|\bar{V}_{t+1}^{k-1}(s')-\underline{V}_{t+1}^{k-1}(s')\big|$$

$$+HS\sum_{k=1}^{K}\sum_{t=1}^{H}\sum_{(s,a)\in L_k}\frac{w_{tk}(s,a)}{n_{k-1}(s,a)\vee 1}$$

$$\overset{(CS)}{\leq}\sum_{k=1}^{K}\sum_{t=1}^{H}\sum_{(s,a)\in L_k} w_{tk}(s,a)\sqrt{\frac{Sp(\cdot\mid s,a)^T\big(\bar{V}_{t+1}^{k-1}-\underline{V}_{t+1}^{k-1}\big)^2}{n_{k-1}(s,a)\vee 1}}$$

$$+HS\sum_{k=1}^{K}\sum_{t=1}^{H}\sum_{(s,a)\in L_k}\frac{w_{tk}(s,a)}{n_{k-1}(s,a)\vee 1}$$

$$\overset{(CS)}{\leq}\sqrt{S}\sqrt{\sum_{k=1}^{K}\sum_{t=1}^{H}\sum_{(s,a)\in L_k}\frac{w_{tk}(s,a)}{n_{k-1}(s,a)\vee 1}}\sqrt{\sum_{k=1}^{K}\sum_{t=1}^{H}\sum_{(s,a)\in L_k} w_{tk}(s,a)p(\cdot\mid s,a)^T\big(\bar{V}_{t+1}^{k-1}-\underline{V}_{t+1}^{k-1}\big)^2}$$

$$+HS\sum_{k=1}^{K}\sum_{t=1}^{H}\sum_{(s,a)\in L_k}\frac{w_{tk}(s,a)}{n_{k-1}(s,a)\vee 1}$$

$$\overset{(*)}{\lesssim}\sqrt{S}\sqrt{SA}\sqrt{SAH^2(F+D)^2+SAH^5}+SH\cdot SA$$
$$=\tilde{\mathcal{O}}\Big(S^{\frac{3}{2}}AH(F+D+H^{\frac{3}{2}})+S^2AH\Big)$$

$(CS)$ denotes Cauchy-Schwarz. Specifically, the first inequality uses $\sum_{i=1}^{n}a_ib_i\leq\sqrt{n\sum_{i=1}^{n}a_i^2b_i^2}$. In $(*)$, we used Lemmas 37 and 26. □

**Lemma 33** (Optimistic Transition Bound). *Let $F, D$ be the constants defined in Lemma 23. Outside the failure event, it holds that*

$$\sum_{k=1}^{K}\sum_{t=1}^{H}\sum_{(s,a)\in L_k} w_{tk}(s,a)(\tilde{p}_{k-1} - \hat{p}_{k-1})(\cdot \mid s,a)^T \bar{V}_{t+1}^{k-1}$$

$$= \tilde{\mathcal{O}}\left( \sqrt{\mathbb{C}^* SAT} + (J + B_p)SA + B_v SAH\left(F + D + H^{\frac{3}{2}}\right) + B_v SA\sqrt{S^{\frac{1}{2}}H(F + D + H^{\frac{5}{2}}) + SH^2} \right)$$

*The following bound also holds:*

$$\sum_{k=1}^{K}\sum_{t=1}^{H}\sum_{(s,a)\in L_k} w_{tk}(s,a)(\hat{p}_{k-1} - \hat{p}_{k-1})(\cdot \mid s,a)^T V_{t+1}^*$$

$$= \tilde{\mathcal{O}}\left( \sqrt{\mathbb{C}^\pi SAT} + (J + B_p)SA + B_v SAH^{\frac{3}{2}}(F + D + H) + B_v SA\sqrt{S^{\frac{1}{2}}H(F + D + H^{\frac{5}{2}}) + SH^2} \right)$$

*Proof.* Similarly to Lemma 10 of [Zanette and Brunskill, 2019], by the definition of the bonus,

$$\sum_{k=1}^{K}\sum_{t=1}^{H}\sum_{(s,a)\in L_k} w_{tk}(s,a)(\tilde{p}_{k-1} - \hat{p}_{k-1})(\cdot \mid s,a)^T \bar{V}_{t+1}^{k-1}$$

$$= \sum_{k=1}^{K}\sum_{t=1}^{H}\sum_{(s,a)\in L_k} w_{tk}(s,a) b_k^{pv}(s,a)$$

$$= \sum_{k=1}^{K}\sum_{t=1}^{H}\sum_{(s,a)\in L_k} w_{tk}(s,a) \left( \phi(\hat{p}_{k-1}(\cdot \mid s,a), \underline{V}_{t+1}^{k-1}) + \frac{B_v\|\bar{V}_{t+1}^{k-1} - \underline{V}_{t+1}^{k-1}\|_{2,\hat{p}}}{\sqrt{n_{k-1}(s,a) \vee 1}} + \frac{4J + B_p}{n_{k-1}(s,a) \vee 1} \right)$$

$$\leq \sum_{k=1}^{K}\sum_{t=1}^{H}\sum_{(s,a)\in L_k} w_{tk}(s,a) \left( \phi(p(\cdot \mid s,a), V^*) + 2\frac{B_v\|\bar{V}_{t+1}^{k-1} - \underline{V}_{t+1}^{k-1}\|_{2,\hat{p}}}{\sqrt{n_{k-1}(s,a) \vee 1}} + 2\frac{4J + B_p}{n_{k-1}(s,a) \vee 1} \right).$$

In the last inequality, we applied Lemma 15, Property(3), and used Equation (13) together with the optimism of the value function, that is $\underline{V}_{t+1}^{k-1} \leq V_{t+1}^* \leq \bar{V}_{t+1}^{k-1}$ (Lemma 18). Next, we substitute the definition of $\phi$ (Definition 1), and get

$$\lesssim \sum_{k=1}^{K}\sum_{t=1}^{H}\sum_{(s,a)\in L_k} w_{tk}(s,a) \left( \frac{g(p, V_{t+1}^*)}{n_{k-1}(s,a) \vee 1} + \frac{J + B_p}{n_{k-1}(s,a) \vee 1} \right) \tag{25}$$

$$+ \sum_{k=1}^{K}\sum_{t=1}^{H}\sum_{(s,a)\in L_k} w_{tk}(s,a) \frac{B_v\|\bar{V}_{t+1}^{k-1} - \underline{V}_{t+1}^{k-1}\|_{2,\hat{p}}}{\sqrt{n_{k-1}(s,a) \vee 1}}. \tag{26}$$

The term in Equation (25) is almost identical to Equation (24) of Lemma 31, and can be similarly bounded by replacing $J$ with $J + B_p$. This yields a bound of either $\tilde{\mathcal{O}}\left( \sqrt{\mathbb{C}^* SAT} + (J + B_p)SA \right)$ or $\tilde{\mathcal{O}}\left( \sqrt{\mathbb{C}^\pi SAT} + (J + B_p)SA + B_v SAH^{\frac{3}{2}}(F + D) + B_v SAH^{\frac{5}{2}} \right)$. We now move to bounding the second term. Notice that

$$\|\bar{V}_{t+1}^{k-1} - \underline{V}_{t+1}^{k-1}\|_{2,\hat{p}}^2 = \hat{p}_{k-1}(\cdot \mid s,a)^T \left( \bar{V}_{t+1}^{k-1} - \underline{V}_{t+1}^{k-1} \right)^2$$

$$= p(\cdot \mid s,a)^T \left( \bar{V}_{t+1}^{k-1} - \underline{V}_{t+1}^{k-1} \right)^2 + (\hat{p}_{k-1} - p)(\cdot \mid s,a)^T \left( \bar{V}_{t+1}^{k-1} - \underline{V}_{t+1}^{k-1} \right)^2$$

$$= \|\bar{V}_{t+1}^{k-1} - \underline{V}_{t+1}^{k-1}\|_{2,p}^2 + (\hat{p}_{k-1} - p)(\cdot \mid s,a)^T \left( \bar{V}_{t+1}^{k-1} - \underline{V}_{t+1}^{k-1} \right)^2 \tag{27}$$

Next, applying Cauchy-Schwartz Inequality on (26), we get

$$(26) \leq B_v \sqrt{\sum_{k=1}^{K}\sum_{t=1}^{H}\sum_{(s,a)\in L_k} \frac{w_{tk}(s,a)}{n_{k-1}(s,a)\vee 1}} \sqrt{\sum_{k=1}^{K}\sum_{t=1}^{H}\sum_{(s,a)\in L_k} w_{tk}(s,a)\|\bar{V}_{t+1}^{k-1} - \underline{V}_{t+1}^{k-1}\|_{2,\hat{p}}^2}$$

$$\lesssim B_v\sqrt{SA}\sqrt{\sum_{k=1}^{K}\sum_{t=1}^{H}\sum_{(s,a)\in L_k} w_{tk}(s,a)\|\bar{V}_{t+1}^{k-1} - \underline{V}_{t+1}^{k-1}\|_{2,p}^2}$$

$$+ B_v\sqrt{SA}\sqrt{\left|\sum_{k=1}^{K}\sum_{t=1}^{H}\sum_{(s,a)\in L_k} w_{tk}(s,a)\left|(\hat{p}_{k-1}-p)(\cdot \mid s,a)^T\left(\bar{V}_{t+1}^{k-1} - \underline{V}_{t+1}^{k-1}\right)^2\right|\right|}} \; ,$$

where the last inequality is by Lemma 37, substituting (27) and using the inequality $\sqrt{a+b} \leq \sqrt{a} + \sqrt{b}$. The first term can be directly bounded by Lemma 26. The second term can be bounded using Lemma 32 as follows:

$$\sum_{k=1}^{K}\sum_{t=1}^{H}\sum_{(s,a)\in L_k} w_{tk}(s,a)\left|(\hat{p}_{k-1}-p)(\cdot \mid s,a)^T\left(\bar{V}_{t+1}^{k-1} - \underline{V}_{t+1}^{k-1}\right)^2\right|$$

$$\leq H\sum_{k=1}^{K}\sum_{t=1}^{H}\sum_{(s,a)\in L_k} w_{tk}(s,a)\left|(\hat{p}_{k-1}-p)(\cdot \mid s,a)^T\left(\bar{V}_{t+1}^{k-1} - \underline{V}_{t+1}^{k-1}\right)\right|$$

$$= \tilde{\mathcal{O}}\left(S^{\frac{3}{2}}AH(F+D+H^{\frac{5}{2}}) + S^2AH^2\right) \; ,$$

where we trivially bounded the value difference by $H$ at the first inequality (due to Lemma 18) and used Lemma 32 at the second one. Summing both terms yields

$$(26) = \tilde{\mathcal{O}}\left( B_v\sqrt{SA}\sqrt{SAH^2(F+D)^2 + SAH^5} + B_v\sqrt{SA}\sqrt{S^{\frac{3}{2}}AH(F+D+H^{\frac{5}{2}}) + S^2AH^2} \right)$$

$$= \tilde{\mathcal{O}}\left( B_vSAH\left(F+D+H^{\frac{3}{2}}\right) + B_vSA\sqrt{S^{\frac{1}{2}}H(F+D+H^{\frac{5}{2}}) + SH^2} \right)$$

Combining both bounds on (25) and (26) concludes the proof. $\qquad\square$

# F    General Lemmas

**Lemma 34** (On Trajectory Regret to Sum of Decreasing Bounded Processes Regret). *For Algorithm 1 and Algorithm 2 it holds that,*

$$\sum_{k=1}^{K}\sum_{t=1}^{H}\mathbb{E}[\bar{V}_t^{k-1}(s_t^k) - \bar{V}_t^k(s_t^k) \mid \mathcal{F}_{k-1}] = \sum_{t=1}^{H}\sum_{s}\sum_{k=1}^{K}\bar{V}_t^{k-1}(s) - \mathbb{E}[\bar{V}_t^k(s) \mid \mathcal{F}_{k-1}]$$

*Proof.* The following relations hold.

$$\sum_{k=1}^{K}\sum_{t=1}^{H}\mathbb{E}[\bar{V}_t^{k-1}(s_t^k) - \bar{V}_t^k(s_t^k) \mid \mathcal{F}_{k-1}] \tag{28}$$

$$= \sum_{k=1}^{K}\sum_{t=1}^{H}\sum_{s}\mathbb{E}[\mathbb{1}\{s = s_t^k\}\bar{V}_t^{k-1}(s) - \mathbb{1}\{s = s_t^k\}\bar{V}_t^k(s) \mid \mathcal{F}_{k-1}]$$

$$\overset{(1)}{=} \sum_{t=1}^{H}\sum_{s}\sum_{k=1}^{K}\mathbb{E}[\mathbb{1}\{s = s_t^k\}\bar{V}_t^{k-1}(s) + \mathbb{1}\{s \neq s_t^k\}\bar{V}_t^{k-1}(s) \mid \mathcal{F}_{k-1}]$$

$$\qquad\qquad - \mathbb{E}[\mathbb{1}\{s = s_t^k\}\bar{V}_t^k(s) + \mathbb{1}\{s \neq s_t^k\}\bar{V}_t^{k-1}(s) \mid \mathcal{F}_{k-1}]$$

$$\overset{(2)}{=} \sum_{t=1}^{H}\sum_{s}\sum_{k=1}^{K}\bar{V}_t^{k-1}(s) - \mathbb{E}[\mathbb{1}\{s = s_t^k\}\bar{V}_t^k(s) + \mathbb{1}\{s \neq s_t^k\}\bar{V}_t^{k-1}(s) \mid \mathcal{F}_{k-1}]$$

$$\overset{(3)}{=} \sum_{t=1}^{H}\sum_{s}\sum_{k=1}^{K}\bar{V}_t^{k-1}(s) - \mathbb{E}[\bar{V}_t^k(s) \mid \mathcal{F}_{k-1}]. \tag{29}$$

Relation (1) holds by adding and subtracting $\mathbb{1}\{s \neq s_t^k\}\bar{V}_t^{k-1}(s)$ while using the linearity of expectation. (2) holds since for any event $\mathbb{1}\{A\} + \mathbb{1}\{A^c\} = 1$ and since $\Delta V_t^{k-1}$ is $\mathcal{F}_{k-1}$ measurable. (3) holds by the definition of the update rule. If state $s$ is visited in the $k^{th}$ episode at time-step $t$, then both $\bar{V}_t^k(s), \underline{V}_t^k(s)$ are updated. If not, their value remains as in the $k-1$ iteration. $\qquad\square$

## F.1    The Good Set $L_k$ and Few Lemmas

We introduce that set $L_k$. The construction is similar to [Dann et al., 2017] and we follow the one formulated in [Zanette and Brunskill, 2019]. The idea is to partition the state-action space at each episode to two sets, the set of state-action pairs that have been visited sufficiently often, and the ones that were not.

**Definition 2.** *The set $L_k$ is defined as follows.*

$$L_k := \left\{(s, a) \in \mathcal{S} \times \mathcal{A} : \frac{1}{4}\sum_{j < k} w_j(s, a) \geq H \ln \frac{SAH}{\delta'} + H\right\}$$

*where $w_j(s, a) := \sum_{t=1}^{H} w_{tj}(s, a)$*

We now state some useful lemmas. See proofs in [Zanette and Brunskill, 2019], Lemma 6, Lemma 7, Lemma 13.

**Lemma 35.** *Outside the failure event, it holds that if $(s, a) \in L_k$, then*

$$n_{k-1}(s, a) \geq \frac{1}{4}\sum_{j \geq k} w_j(s, a) \ ,$$

*which also implies that $n_{k-1}(s, a) \geq H \ln \frac{SAH}{\delta'} + H \geq 1$*

**Lemma 36.** *Outside the failure event, it holds that*

$$\sum_{k=1}^{K}\sum_{t=1}^{H}\sum_{(s,a)\notin L_k} w_{tk}(s, a) \leq \tilde{\mathcal{O}}(SAH).$$

**Lemma 37.** *Outside the failure event, it holds that*

$$\sum_{k=1}^{K}\sum_{t=1}^{H}\sum_{(s,a)\in L_k} \frac{w_{tk}(s,a)}{n_{k-1}(s,a)} \leq \tilde{\mathcal{O}}(SA).$$

Combining these lemmas we conclude the following one.

**Lemma 38.** *Outside the failure event, it holds that*

$$\sum_{k=1}^{K}\sum_{t=1}^{H}\mathbb{E}\left[\sqrt{\frac{1}{n_{k-1}(s_t^k,\pi_k(s_t^k))\vee 1}} \mid \mathcal{F}_{k-1}\right] \leq \tilde{\mathcal{O}}(\sqrt{SAT}+SAH)$$

*Proof.* The following holds relations hold.

$$\sum_{k=1}^{K}\sum_{t=1}^{H}\mathbb{E}\left[\sqrt{\frac{1}{n_{k-1}(s_t^k,\pi_k(s_t^k))\vee 1}} \mid \mathcal{F}_{k-1}\right]$$

$$= \sum_{k=1}^{K}\sum_{t=1}^{H}\sum_{s,a} w_{tk}(s,a)\sqrt{\frac{1}{n_{k-1}(s,a)\vee 1}}$$

$$\leq \sum_{k=1}^{K}\sum_{t=1}^{H}\sum_{s,a\in L_k} w_{tk}(s,a)\sqrt{\frac{1}{n_{k-1}(s,a)}} + \sum_{k=1}^{K}\sum_{t=1}^{H}\sum_{s,a\notin L_k} w_{tk}(s,a)$$

$$\leq \sum_{k=1}^{K}\sum_{t=1}^{H}\sum_{s,a\in L_k} w_{tk}(s,a)\sqrt{\frac{1}{n_{k-1}(s,a)}} + SAH. \tag{30}$$

The first relation holds by definition. The second relation holds by the following argument. For the first term, if $(s,a)\in L_k$ then by Lemma 35, $n_{k-1}(s,a)\geq 1$, and thus $n_{k-1}(s,a)\vee 1 = n_{k-1}(s,a)$. The second term is bounded by taking the worst case for the fraction, which is $n_{k-1}(s,a)\vee 1 \geq 1$. The third relation holds by Lemma 36.

Consider the first term in (30).

$$\sum_{k=1}^{K}\sum_{t=1}^{H}\sum_{s,a\in L_k} w_{tk}(s,a)\sqrt{\frac{1}{n_{k-1}(s,a)}}$$

$$\leq \sqrt{\sum_{k=1}^{K}\sum_{t=1}^{H}\sum_{s,a\in L_k} w_{tk}(s,a)}\sqrt{\sum_{k=1}^{K}\sum_{t=1}^{H}\sum_{s,a\in L_k} \frac{w_{tk}(s,a)}{n_{k-1}(s,a)}}$$

$$\leq \sqrt{\sum_{k=1}^{K}\sum_{t=1}^{H}\sum_{s,a} w_{tk}(s,a)}\sqrt{\sum_{k=1}^{K}\sum_{t=1}^{H}\sum_{s,a\in L_k} \frac{w_{tk}(s,a)}{n_{k-1}(s,a)}}$$

$$= \sqrt{T}\sqrt{\sum_{k=1}^{K}\sum_{t=1}^{H}\sum_{s,a} \frac{w_{tk}(s,a)}{n_{k-1}(s,a)}} \lesssim \tilde{\mathcal{O}}(\sqrt{SAT}).$$

The first relation holds by Cauchy-Schartz inequality. In the second relation, we replaced the sum in the first term to cover all of the state-action pairs, thus adding positive quantities. The third relation holds since by definition $\sum_{t=1}^{H}\sum_{s,a} w_{tk}(s,a) = H$ and $T = KH$. The last relation holds by Lemma 37.

Combining the result in (30) concludes the proof. $\qquad\qquad\square$

**Lemma 39.** *Let $u, v \geq 0$ be some non-negative constants. Outside the failure event,*

$$\sum_{k=1}^{K} \sum_{t=1}^{H} \mathbb{E}\left[\min\left\{\frac{u}{n_{k-1}(s_{t'}^{k}, a_{t'}^{k}) \vee 1}, v\right\} \mid \mathcal{F}_{k-1}\right] \leq \tilde{\mathcal{O}}(SAu + SAHv) \;,$$

*and specifically,*

$$\sum_{k=1}^{K} \sum_{t=1}^{H} \mathbb{E}\left[\frac{u}{n_{k-1}(s_{t'}^{k}, a_{t'}^{k}) \vee 1} \mid \mathcal{F}_{k-1}\right] \leq \tilde{\mathcal{O}}(SAHu) \;.$$

*Proof.* The proof partially follows [Zanette and Brunskill, 2019], Lemma 12:

$$\sum_{k=1}^{K} \sum_{t=1}^{H} \mathbb{E}\left[\min\left\{\frac{u}{n_{k-1}(s_{t'}^{k}, a_{t'}^{k}) \vee 1}, v\right\} \mid \mathcal{F}_{k-1}\right]$$

$$\stackrel{(1)}{=} \sum_{k=1}^{K} \sum_{t=1}^{H} \sum_{s,a} w_{tk}(s,a) \min\left\{\frac{u}{n_{k-1}(s,a) \vee 1}, v\right\}$$

$$\stackrel{(2)}{\leq} \sum_{k=1}^{K} \sum_{t=1}^{H} \sum_{(s,a) \in L_k} w_{tk}(s,a) \frac{u}{n_{k-1}(s,a)} + v \sum_{k=1}^{K} \sum_{t=1}^{H} \sum_{(s,a) \notin L_k} w_{tk}(s,a)$$

$$\stackrel{(3)}{\lesssim} SAu + SAHv$$

(1) is from the definition of $w_{tk}(s,a)$ and the fact that $n_{k-1}(s,a)$ is $\mathcal{F}_{k-1}$ measurable. In (2) we divided the sum into state-actions in and outside $L_k$. For state-actions in $L_k$, we bounded the minimum by the first term, and otherwise we bounded it by $H^2$. Note that for any $(s,a) \in L_k$, $n_{k-1}(s,a) \geq 1$, from Lemma 35. (3) is due to Lemmas 36 and 37.

The second part of the lemma is a direct result of fixing $v = u$.

$\square$