[Reviews · NeurIPS 2019]

Reviewer 1



Originality: The paper proceeds in a direction somewhat orthogonal to recent literature in the area, by swapping out the full-planning strategy for a one-step greedy strategy. As such, it opens up potential new research approaches along with providing an improvement on the SOTA. Quality: The argument is well-developed, and extensive proofs are provided in the supplementary materials or referenced in existing literature. The greedy approach is directly applied to two existing SOTA full-planning-based algorithms, suggesting it is a generalizable alternative. Clarity: The paper is generally well-organized and clear; the paper gives an intuitive sense of the results, although the bulk of the proofs are confined to the supplementary material. Several scattered clarity issues are described in the detailed comments below. Significance: The paper provides both a direct theoretical contribution to the field of regret analysis. More generally, challenging the assumed supremacy of full-planning-based approaches may have ripple-on effects beyond the specific problem domain (tabular finite-horizon undiscounted MDPs). Detailed comments: Abstract: I think it's important to point out that the results apply to undiscounted MDPs. Table 1: Why is the Regret bound for UCLR2 listed in terms of the horizon H and not the diameter D, as in the original paper? Line 124: It's unclear here what is former and what is latter; I would state "since \bar{V}^{k-1}(s_1^k) >= V*(s_1^k) >= V_1^{\pi_k}(s_1^k)". Line 212 and Algorithm 2: Algorithm 5 in the supplementary materials requires updating the lower bound variables after action selection, and therefore seemingly can't be used as a drop-in replacement for ModelBaseOptimisticQ as suggested. Ideally Algorithm 2 should include an additional line to be fully general to both UCRL2 and EULER, or it should at least be mentioned in a footnote. Line 263: Prioritized sweeping, particularly prioritized sweeping with small backups (van Seijen & Sutton 2013) should be referenced as a highly efficient full-planning approach.

Reviewer 2



The paper presents an original and novel contribution that is of interest to the field. However, the paper is not very accessible and makes little effort to summaries the found results in an intuitive manner and why the results are true. For the NeurIPS audience, the writing of the paper should put far more emphasis on building intuitive examples that explain the theoretical findings, rather than just stating them. Including a proof sketch into the main body of the paper does not help the overall presentation. In Sec. 4, why are the model-based algorithms optimistic (as stated in lines 177)? According to my understanding, Alg. 2 builds an approximate model (rhat_k, phat_k,...). Couldn’t approximation errors bias the model so that the algorithm is not strictly optimistic? Regarding Lemma 1 and line 109: Isn’t lemma 1 a re-statement of the proof presented at this link: https://m.tau.ac.il/~mansour/rl-course/scribe6/node4.html

Reviewer 3



This paper considers the interesting question of whether model-based reinforcement learning can achieve good performing by using a myopic policy instead of doing full planning at each step, and presents an affirmative answer for this. Specifically, it is shown that UCRL2 and EULER can be adapted to use a myopic policy that does 1-step planning, preserving the regret and improving the time complexity by a factor of S. How large are the constants in the complexity notations and how do they compare with their full-planning counterparts? Some experiments comparing the actual runing time of the greedy algorithms with their full-planning counterparts will be helpful. Overall, the paper is well-written and the results are very interesting. * Update after rebuttal The authors have clearly answered my questions about the the constants in the complexity notations and promised to provide experimental comparison between the greedy algorithms and their full-planning counterparts. I still vote for acceptance.

[Author Response · NeurIPS 2019]

We would like to begin by highlighting two contributions of the paper we feel remained unnoticed by R#2 and R#3. In
the final version, we will better emphasize their value as it seems their importance was not properly conveyed.

(i) Theorem 1 bounds with high probability the regret of a Decreasing Bounded Process (DBP) by a constant (with no
dependence on the episode number $K$) under *no assumptions* except the structure of a bounded and decreasing process.
Due to its generality it is a powerful tool and is indeed central in all our analysis. We believe it can be instrumental
analyzing other RL and planning algorithms as well as possibly in general learning theory.

(ii) The analysis of RTDP is important on its own. RTDP is a well known and practical algorithm. Previous analysis of
RTDP required the algorithm designer to set a predefined level of accuracy $\epsilon$. In the analysis of Section 3, we prove
both regret bounds (which did not exist before) and better PAC bounds in comparison to previous analysis. On top of
that, in the analyzed version we need not assume a predefined accuracy level $\epsilon$ as in previous works.

**Reviewer #1:** We thank the reviewer for his/her favorable review. We will make sure that the final version of the paper
takes into account the reviewer's comments. Specifically:

• Abstract/Line 124/Line 263 - will be corrected, thanks!

• Comparison to UCRL2: In the literature on regret analysis of finite-horizon MDPs, it is common to present the
regret bounds as a function of the horizon. Specifically, and to the best of our knowledge, all previous works present
UCRL2's regret bound as in the table (e.g., Azar et al. 2017, Jin et al. 2018). Nevertheless, we will add a remark on
this issue to avoid confusion.

• Line 212: True, the more general version of Algorithm 2 should allow some auxiliary calculations for the optimistic
model – we will modify the final version to reflect this issue.

**Reviewer #2:** We thank the reviewer for the comments. In the final version of the paper, we will take extra care to
make the paper more accessible, as well as add experiments to better demonstrate the ideas (see response to R#3).

• The result demonstrates a possibly intuitive argument: if you only access an approximate model, and thus do not
know the exact outcomes of your actions, it is redundant to estimate long term outcomes with it. Instead, one can
incorporate the long-term outcomes in an optimistic value function while keeping the same performance. That is to
say, it is surprising that the regret bounds on the full-planning and RTDP approaches are exactly the same (up to
numerical constants) and is a major contribution of this paper. We will emphasize this better in the introduction.

• Optimism (Line 177) - We combine our approach with optimistic algorithms, i.e., algorithms that ensure with high
probability that the Q-values are optimistic. Both UCRL2 and EULER do so by using upper confidence bounds on
both rewards and transitions. The Q-values are optimistic in a high-probability sense, i.e., with high probability
$\max_a \bar{Q}(s,a) \geq \max_a Q^*(s,a)$ for any $s$ (Lemma 14, 22). Equivalently, the confidence intervals we use assure that
the probability the Q-values are pessimistic is small. This is a common approach in both RL and bandits literature.

• Lemma 1: The lemma referred by the reviewer is related to Value Iteration (VI) when the value is uniformly updated
on all the states by the Bellman operator. RTDP, unlike VI, only updates *visited states*, and does not perform
uniform updates as VI. While there are some similarities to the monotonicity of the Bellman operator, more careful
arguments are required for this result to hold in case of RTDP.

**Reviewer #3:** We thank the reviewer for the feedback. We will address the following question that was raised: 'How
large are the constants in the complexity notations and how do they compare with their full-planning counterparts?'

Our analysis decomposes the regret into two terms (e.g., Equation 14) (A) and (B):

**(A)** This term is a DBP and does not exist in previous analysis. Nevertheless, it is an additive term which can be
bounded by $9SH^2 \ln(3SH/\delta)$, and thus negligible relatively to rest of the constants.

**(B)** For EULER-GP, this term is almost identical to the regret analyzed in (Zanette et al.), there it is bounded by
$O\left(\sqrt{HSAT} + \sqrt{S}SAH^2(\sqrt{S} + \sqrt{H})\right)$. In our case, the analysis of the second term is a bit more involved and
requires using (again) the result on DBP on top of using results from (Zanette et al.). The $\sqrt{T}$ term is the *exact* same
one as in (Zanette et al.), i.e., the omitted constant terms are similar. Our analysis results in additional additive terms
which are independent of $\sqrt{T}$ and smaller than $O\left(\sqrt{S}SAH^2(\sqrt{S} + \sqrt{H})\right)$. Comparing the value of the multiplicative
constant of this term is problematic as the results in (Zanette et al.) do not state the values of the multiplicative constants.

To summarize, except for a negligible additive constants, there is almost no effect on the constants of the regret bounds.

**Experiments:** we will add some experiments to compare the original and modified algorithms in the final version. We
already performed some initial simulations which indicate that the performance with greedy policies is similar to the
performance with full planning in some simple environments, but further experiments are required.

[Meta-Review · NeurIPS 2019]

All reviews agree that the contribution is novel and strong. The rebuttal gave important answers and we all strongly defend acceptance.